# Robust Conformal Outlier Detection under Contaminated Reference Data

**Meshi Bashari** [1]   **Matteo Sesia** [2 3]   **Yaniv Romano** [1 4]

## Abstract

Conformal prediction is a flexible framework for calibrating machine learning predictions, providing distribution-free statistical guarantees. In outlier detection, this calibration relies on a reference set of labeled inlier data to control the type-I error rate. However, obtaining a perfectly labeled inlier reference set is often unrealistic, and a more practical scenario involves access to a contaminated reference set containing a small fraction of outliers. This paper analyzes the impact of such contamination on the validity of conformal methods. We prove that under realistic, non-adversarial settings, calibration on contaminated data yields conservative type-I error control, shedding light on the inherent robustness of conformal methods. This conservativeness, however, typically results in a loss of power. To alleviate this limitation, we propose a novel, active data-cleaning framework that leverages a limited labeling budget and an outlier detection model to selectively annotate data points in the contaminated reference set that are suspected as outliers. By removing only the annotated outliers in this "suspicious" subset, we can effectively enhance power while mitigating the risk of inflating the type-I error rate, as supported by our theoretical analysis. Experiments on real datasets validate the conservative behavior of conformal methods under contamination and show that the proposed data-cleaning strategy improves power without sacrificing validity.

[1]Department of Electrical and Computer Engineering, Technion IIT, Haifa, Israel [2]Department of Data Sciences and Operations, University of Southern California, Los Angeles, California, USA [3]Department of Computer Science, University of Southern California, Los Angeles, California, USA [4]Department of Computer Science, Technion IIT, Haifa, Israel. Correspondence to: Meshi Bashari <meshi.b@campus.technion.ac.il>.

*Proceedings of the 42^{nd} International Conference on Machine Learning*, Vancouver, Canada. PMLR 267, 2025. Copyright 2025 by the author(s).

## 1. Introduction

### 1.1. Background and Motivation

This paper studies the problem of outlier detection: given a reference dataset (e.g., a collection of legitimate financial transactions) and an unlabeled test point (a new transaction), our goal is to determine whether the test point is an outlier (a fraudulent transaction) by assessing its deviation from the reference data distribution. Naturally, we aim to maximize the detection of outliers by harnessing the capabilities of complex machine learning (ML) models. However, these models typically lack type-I error rate control, potentially resulting in unreliable detections. In our running example, the type-I error is the probability of falsely flagging a legitimate transaction as fraudulent. As such, uncontrolled error rates can lead to costly unnecessary investigations of legitimate transactions and negatively impact customer experience.

The broad need for reliable ML systems has sparked a surge of interest in conformal prediction—a versatile framework that can provide statistical guarantees for any "black-box" predictive model (Vovk et al., 2005). This framework formulates the outlier detection task as a statistical test, where the null hypothesis is that the new data point is not an outlier (Laxhammar & Falkman, 2015; Bates et al., 2023). To derive a decision rule guaranteeing type-I error control, conformal inference relies on a reference (calibration) set of inlier data points. These points are assumed to be sampled i.i.d. from an unknown distribution, independent of the data used to train the outlier detection model.

In practice, however, it is often difficult to obtain a perfectly clean reference dataset that contains no outliers (Park et al., 2021; Zhao et al., 2019; Chalapathy & Chawla, 2019; Jiang et al., 2022). In our example, a more realistic scenario would assume instead access to a slightly *contaminated* reference set, mostly legitimate transactions with a few unnoticed outliers (Zhao et al., 2019). But this setting poses new challenges for conformal prediction methods, potentially invalidating the error control guarantees or, as we shall see, often reducing the power to detect true outliers at test time.

### 1.2. Outline and Contributions

While type-I error control in conformal inference theoretically requires perfectly clean reference data, in practice

contaminated data often makes these methods overly conservative, reducing the power to detect true outliers rather than inflating the type-I error rate. This empirical observation motivates the first question explored in this paper:

**Q1:** *When does conformal outlier detection with contaminated reference data yield valid type-I error control?*

In Section 2.3, we present the first contribution of this paper: a novel theoretical analysis that identifies common conditions under which this conservative behavior arises. Unfortunately, this conservativeness often comes at the cost of reduced detection power, particularly when targeting low type-I error rates. To address this issue, we investigate data-driven cleaning strategies aimed at mitigating the contamination in the reference dataset.

A straightforward approach to cleaning the contaminated set is to remove all data points flagged as likely outliers by the detection model. However, this method is unsatisfactory, as it risks inadvertently removing inliers along with outliers, resulting in an "overly clean" reference set. This, in turn, distorts the inlier distribution and inflates the type-I error rate above the desired nominal level.

This challenge motivates our second and main contribution. In Section 3.3, we introduce an approach to clean the contaminated reference set by leveraging a limited labeling budget (e.g., 50 annotations). The outlier detection model is first used to identify suspected outliers within the contaminated reference set. The limited budget is then strategically allocated to annotate these points, thereby avoiding the unintended removal of inliers. While this is a practical and intuitive approach, it naturally prompts a critical question:

**Q2:** *How does the selective annotation and partial removal of outliers from a contaminated reference set affect the validity of conformal inferences?*

We analyze the validity of this active labeling approach for trimming outliers in the contaminated set. Our theoretical results identify the conditions required to achieve approximate type-I error control, even when the data are selectively annotated and not all outliers are removed. This analysis also highlights key factors that can inflate the error rate, offering practitioners guiding principles to enhance the power of conformal methods in the presence of contaminated data.

Finally, in Section 4, we empirically validate our theory and proposed data-cleaning approach through comprehensive experiments on real-world datasets. The experiments confirm that conformal inference with contaminated data tends to be conservative. Furthermore, they demonstrate that our method significantly boosts power, particularly when the target type-I error rate is low and the number of outliers in the contaminated set is small.

A software that implements the proposed method is available

at https://github.com/Meshiba/robust-conformal-od.

### 1.3. Related Work

Recently, there has been growing interest in studying the statistical properties of conformal inference methods under more realistic scenarios, moving beyond the idealized assumption of perfectly clean and exchangeable observations to account for various types of *distribution shift* (Tibshirani et al., 2019; Einbinder et al., 2024; Sesia et al., 2024; Barber et al., 2023; Gibbs & Candès, 2021; Zaffran et al., 2022; Feldman et al., 2023; Gibbs & Candès, 2024; Podkopaev & Ramdas, 2021; Si et al., 2023; Prinster et al.). This paper draws inspiration from several prior works in this area.

Tibshirani et al. (2019) introduced a weighted conformal prediction approach to address covariate shift between calibration and test data, later extended by Podkopaev & Ramdas (2021) to accommodate label shift. Both settings, however, involve a different form of distribution shift from the one we study here. Barber et al. (2023) extend this line of work by analyzing the effects of general distribution shifts on the validity of conformal methods, focusing however on worst-case scenarios; see also Farinhas et al. (2024).

In contrast, our work moves away from this worst-case perspective. We aim to explain why conformal outlier detection with contaminated reference data often results in a conservative type-I error rate, rather than investigating type-I error inflation, which, while theoretically possible in adversarial settings, appears less common in practice. Furthermore, we focus on developing methods to address this over-conservativeness, boosting detection power.

A more closely related line of work investigates the robustness of conformal prediction to label noise (Einbinder et al., 2024; Sesia et al., 2024; Clarkson et al., 2024; Penso et al., 2025) or other forms of data contamination (Zaffran et al., 2023; 2024; Feldman & Romano, 2024). Specifically, Einbinder et al. (2024) and Sesia et al. (2024) show that, under certain assumptions, conformal prediction for classification with noisy labels often results in conservative type-I error rates. Furthermore, Sesia et al. (2024) proposes a method to address this conservativeness by leveraging an explicit "label noise model" that captures the relationship between the true and contaminated labels in the calibration dataset.

In contrast, this paper avoids relying on an explicit model for the contaminated data, as such models can be difficult to estimate in practice within our context. Instead, we utilize a pre-trained black-box outlier detection model and a limited annotation budget to selectively and reliably trim outliers from the contaminated set. Furthermore, it is important to emphasize that the method proposed by Sesia et al. (2024) is primarily designed for classification tasks with relatively balanced data, whereas outlier detection naturally involves

extreme class imbalance. This distinction underscores the need for solutions specifically tailored to outlier detection.

## 2. Setup and Preliminary Results

### 2.1. Inference with Clean Calibration Data

Conformal inference for outlier detection requires a reference (or *calibration*) set, $\mathcal{D}_{\mathrm{cal}} = [n] := \{1, \ldots, n\}$, containing $n$ data points. The reference set is typically assumed to be *clean*, consisting solely of *inliers*, which are i.i.d. samples from an unknown distribution $\mathbb{P}_0$ (exchangeability may sometimes suffice, but this work assumes i.i.d. inliers). Under this assumption, $\mathcal{D}_{\mathrm{cal}}$ may be referred to as $\mathcal{D}_{\mathrm{inlier}}$.

The goal is to determine whether a new observation, $X_{n+1}$, is an inlier—independently sampled from $\mathbb{P}_0$—or an *outlier*, sampled from a different distribution $\mathbb{P}_1 \neq \mathbb{P}_0$. This can be formulated as a hypothesis testing problem, where the null hypothesis $\mathcal{H}_0$ claims that $X_{n+1}$ is an inlier:

$$X_i \overset{\text{i.i.d.}}{\sim} \mathbb{P}_0, \ \forall i \in \mathcal{D}_{\mathrm{inlier}}, \quad \mathcal{D}_{\mathrm{inlier}} = \mathcal{D}_{\mathrm{cal}} = [n],$$
$$\mathcal{H}_0 : X_{n+1} \overset{\text{ind.}}{\sim} \mathbb{P}_0 \,. \tag{1}$$

The split-conformal method, a simple and computationally efficient approach, uses a pre-trained outlier detection model—potentially any machine learning model—to compute *nonconformity scores* that quantify how different a data point is from the reference distribution. The model, represented by a score function $s$, is trained on a separate dataset $\mathcal{D}_{\mathrm{train}}$, which is similar to but independent of $\mathcal{D}_{\mathrm{cal}}$. Typically, a larger dataset of inliers, assumed to be i.i.d. samples from $\mathbb{P}_0$, is randomly split into $\mathcal{D}_{\mathrm{train}}$ and $\mathcal{D}_{\mathrm{cal}}$.

The model tries to learn a score function $s$ such that larger values of $s(X_{n+1})$ indicate stronger evidence that the test point may be an outlier. Conformal inference rigorously quantifies this evidence, providing a principled decision rule for rejecting $\mathcal{H}_0$ when the evidence is strong enough, while controlling the type-I error rate—the probability of incorrectly rejecting $\mathcal{H}_0$ when $X_{n+1}$ is actually an inlier.

This statistical evidence is quantified by computing a *conformal p-value*, defined as:

$$\hat{p}_{n+1} = \frac{1 + \sum_{i=1}^{n} \mathbb{I}[s(X_i) \geq s(X_{n+1})]}{1 + n}. \tag{2}$$

Thus, larger values of $s(X_{n+1})$ correspond to smaller values of $\hat{p}_{n+1}$, and the test point $X_{n+1}$ can be confidently classified as an outlier (rejecting $\mathcal{H}_0$) when $\hat{p}_{n+1}$ is smaller than a given significance level $\alpha \in (0, 1)$.

**Proposition 2.1** (from Vovk et al. (2005)). *Under* (1)*, if the null hypothesis $\mathcal{H}_0$ is true, then for any $\alpha \in (0, 1)$:* $\mathbb{P}(\hat{p}_{n+1} \leq \alpha) \leq \alpha$. *Further, if $s(X)$ has a continuous distribution under $\mathbb{P}_0$, then $\mathbb{P}(\hat{p}_{n+1} \leq \alpha) \geq \alpha - 1/(n+1)$.*

Proposition 2.1 intuitively states that the conformal p-value defined in (2) provides a well-calibrated rule for flagging a new data point as a likely outlier. Rejecting $\mathcal{H}_0$ when $\hat{p}_{n+1} \leq \alpha$ ensures type-I error control at level $\alpha$ while avoiding excessive conservatism. Specifically, the type-I error rate closely matches $\alpha$ when the sample size $n$ is large and the nonconformity scores have a continuous distribution with no ties—a mild condition that can be achieved in practice by adding small random noise to the scores.

What remains unclear, and serves as the starting point of this paper, is how conformal p-values behave when the calibration dataset is contaminated, containing not only inliers but also a fraction of misplaced outliers.

### 2.2. Inference with Contaminated Calibration Data

In this paper, we consider a more general setting where the calibration dataset, indexed by $\mathcal{D}_{\mathrm{cal}} = [n]$, may contain both inliers ($\mathcal{D}_{\mathrm{inlier}}$), sampled i.i.d. from a distribution $\mathbb{P}_0$, and outliers ($\mathcal{D}_{\mathrm{outlier}}$), sampled i.i.d. from a different distribution $\mathbb{P}_1 \neq \mathbb{P}_0$. Thus, $\mathcal{D}_{\mathrm{cal}} = \mathcal{D}_{\mathrm{inlier}} \cup \mathcal{D}_{\mathrm{outlier}}$. The numbers of inliers and outliers, respectively $n_0 = |\mathcal{D}_{\mathrm{inlier}}|$ and $n_1 = |\mathcal{D}_{\mathrm{outlier}}|$, are treated as fixed, with $n = n_0 + n_1$. The goal remains to test the null hypothesis $\mathcal{H}_0$ that a new data point $X_{n+1}$ is an inlier, independently sampled from $\mathbb{P}_0$. Formally, this setup can be written as:

$$X_i \overset{\text{i.i.d.}}{\sim} \mathbb{P}_0, \ \forall i \in \mathcal{D}_{\mathrm{inlier}}, \quad X_i \overset{\text{i.i.d.}}{\sim} \mathbb{P}_1, \ \forall i \in \mathcal{D}_{\mathrm{outlier}},$$
$$\mathcal{D}_{\mathrm{inlier}} \cup \mathcal{D}_{\mathrm{outlier}} = \mathcal{D}_{\mathrm{cal}} = [n], \tag{3}$$
$$\mathcal{H}_0 : X_{n+1} \overset{\text{ind.}}{\sim} \mathbb{P}_0 \,.$$

In the following, we first analyze the behavior of standard conformal p-values, computed as in (2), when applied to contaminated data scenarios described by (3). Subsequently, we will propose a novel method for computing more refined conformal p-values by approximately cleaning the calibration set to remove undesired outliers.

### 2.3. Explaining the Conservativeness

Empirical results suggest that contamination by outliers in the calibration data often makes standard conformal p-values overly conservative, resulting in a type-I error rate significantly lower than the desired nominal level $\alpha$.

We begin by examining Figure 1, which provides some insight into this behavior based on the analysis of the "shuttle" dataset (shu), as detailed in Section 4. In this example, the nonconformity scores of outlier data points in the contaminated calibration set are typically larger than those of the inliers. This pattern reflects the goal of a well-designed outlier detection model: to differentiate outliers from inliers by assigning higher scores to the former. Consequently, conformal p-values computed using (2) will be inflated relative to

the ideal scenario in which all calibration points are inliers, reducing our power to detect true outliers at test time.

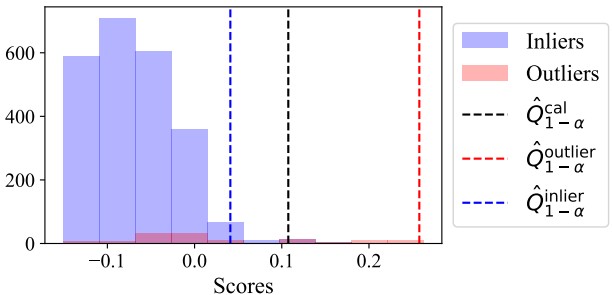

Figure 1: Histogram of nonconformity scores for inliers and outliers in a contaminated calibration subset of the "shuttle" data, with a contamination rate of 5%. The vertical lines indicate the $(1 - \alpha)$ empirical quantile of all calibration scores (black), as well as separately for inliers (blue) and outliers (red), with $\alpha = 0.02$.

This phenomenon is further corroborated by extensive numerical experiments presented in Section 4 and Appendix B.2, which consistently demonstrate this conservative behavior across nine different datasets.

While the conservativeness of conformal prediction methods in the presence of contaminated calibration data has already been observed and studied theoretically in different contexts (Einbinder et al., 2024; Sesia et al., 2024; Clarkson et al., 2024), prior works did not focus on outlier detection. Therefore, it is helpful to introduce a new theoretical result that precisely quantifies the inflation of standard conformal p-values that we often observe in practice. This will serve as a foundation for the novel method of computing adaptive conformal p-values presented in the next section.

Let $\hat{F}_1$ denote the empirical cumulative distribution function (CDF) of the scores in $\mathcal{D}_{\text{outlier}}$ and $\hat{Q}^{\text{cal}}_{1-\alpha}$ represent the $\lceil (1 - \alpha)(1 + n) \rceil$-th smallest score in the calibration set.

**Lemma 2.2.** *Under the setup defined in (3), if $\mathcal{H}_0$ is true, then for any $\alpha \in (0, 1)$,*

$$\mathbb{P}\left(\hat{p}_{n+1} \leq \alpha\right) \leq \alpha - \frac{n_1}{n_0 + 1}\left(1 - \alpha - \mathbb{E}\left[\hat{F}_1\left(\hat{Q}^{\text{cal}}_{1-\alpha}\right)\right]\right).$$

This result is related to Theorem 1 in Sesia et al. (2024), which studies the behavior of conformal prediction sets for multi-class classification (Lei et al., 2013; Romano et al., 2020) calibrated with contaminated data. The key distinction is in our treatment of the calibration set: we assume that $n_0$ and $n_1$ are fixed, whereas Sesia et al. (2024) consider a mixture model where the observed proportions of data points from different classes in the calibration set are random. While treating $n_0$ and $n_1$ as fixed is convenient

for this paper, we also include an additional result (Corollary A.3) in Appendix A, which reaches qualitatively similar conclusions by adopting an approach more closely aligned with Theorem 1 from Sesia et al. (2024).

A direct corollary of Lemma 2.2 is that standard conformal p-values are conservative when outlier scores are typically larger than inlier scores. Formally, this condition is:

**Assumption 2.3.** $\mathbb{E}[\hat{F}_1(\hat{Q}^{\text{cal}}_{1-\alpha})] < 1 - \alpha.$

If Assumption 2.3 fails—for instance, when outlier scores are smaller than inlier scores—data contamination may invalidate standard conformal p-values, inflating the type-I error rate. However, it is more common in practice that Assumption 2.3 holds, in which case contamination tends to reduce calibration power, and more so if $n_1$ is large. In particular, Assumption 2.3 holds if: (i) the outlier detection model is relatively accurate, and (ii) the outlier distribution $\mathbb{P}_1$ is not adversarial. Our experiments will show this power loss can be substantial, motivating the need for new methods that can approximately "clean up" the calibration data.

## 3. Methods

### 3.1. Key Idea: Boosting Power by Cleaning the Data

Ideally, we would like to remove all $n_1$ outliers from $\mathcal{D}_{\text{cal}}$, restoring the ideal behavior of conformal p-values calibrated on a clean dataset, as described in Proposition 2.1, and likely boosting power. However, manually labeling the entire contaminated calibration set $\mathcal{D}_{\text{cal}}$ is often impractical, especially when $n = |\mathcal{D}_{\text{cal}}|$ is large. At the same time, utilizing only a small calibration set is not always desirable.

A large calibration set is often needed because the smallest conformal p-value obtainable through (2) scales as $1/n$. Thus, a large $n$ is critical for achieving high confidence in identifying outliers, especially in "needle-in-a-haystack" scenarios (Bates et al., 2023), where a few outliers must be detected in a large test set dominated by inliers. In such cases, the ability to obtain very small p-values is essential to achieve non-trivial power while controlling the false discovery rate (Benjamini & Hochberg, 1995).

### 3.2. A Simple but Unsatisfactory Approach: Naive-Trim

The above challenge underscores the need for a method to mitigate the impact of outliers in the calibration dataset without requiring exhaustive annotation. An intuitive approach is to forgo annotations and simply remove all "suspicious" data points with large nonconformity scores. For instance, one could remove the top $m$ scores from $\mathcal{D}_{\text{cal}}$, where $m$ is a fixed guess of the true number of outliers $n_1$ in the calibration set. We refer to this approach as `Naive-Trim`.

While `Naive-Trim` can reduce conservativeness by re-

moving some outliers, it is not a satisfactory solution as it risks "over-compensating". By potentially removing also true inliers with large nonconformity scores, it can significantly skew the inlier score distribution to the left. This side effect is problematic, as it tends to invalidate conformal p-values and inflate the type-I error rate, over-correcting the conservativeness of standard conformal p-values.

This issue is particularly pronounced when $m > n_1$ or in noisy settings where the outlier detection model cannot perfectly distinguish between inliers and outliers. For example, as shown in Section 4, applying Naive-Trim to the dataset illustrated in Figure 1 results in uncontrolled inflation of the type-I error rate.

To address this challenge, we will now present a more sophisticated method, which we refer to as Label-Trim. This approach utilizes a limited labeling budget to remove outliers from $\mathcal{D}_{\text{cal}}$ in a more reliable manner, mitigating the risk of over-correcting the conformal p-value.

### 3.3. The Label-Trim Method

Consider having a limited budget to label $m < n$ calibration samples, where $m$ is much smaller than $n$. We aim to utilize this budget to remove as many outliers as possible from the calibration set without altering the inlier score distribution.

A practical approach is to annotate the $m$ largest scores in $\mathcal{D}_{\text{cal}}$, as these are most likely outliers based on the model. Denote these annotated samples as $\mathcal{D}_{\text{labeled}} \subseteq \mathcal{D}_{\text{cal}}$, and let $\mathcal{D}_{\text{labeled}}^{\text{outlier}}$ denote the subset of annotated data points that are true outliers. Removing these outliers from $\mathcal{D}_{\text{cal}}$ yields a smaller, cleaner calibration set, which we call $\mathcal{D}_{\text{cal}}^{\text{LT}} \subseteq \mathcal{D}_{\text{cal}}$.

The Label-Trim method then calculates a refined conformal p-value, now denoted as $\hat{p}_{n+1}^{\text{LT}}$, following the same procedure as in (2) with $\mathcal{D}_{\text{cal}}$ replaced by the (partially) cleaned calibration set $\mathcal{D}_{\text{cal}}^{\text{LT}}$:

$$\hat{p}_{n+1}^{\text{LT}} = \frac{1 + \sum_{i \in \mathcal{D}_{\text{cal}}^{\text{LT}}} \mathbb{I}[s(X_i) \geq s(X_{n+1})]}{1 + |\mathcal{D}_{\text{cal}}^{\text{LT}}|}. \quad (4)$$

Algorithms 1 and 2 summarize this procedure, which intuitively offers advantages over both the standard method for computing $\hat{p}_{n+1}$ in (2), by potentially increasing power, and the Naive-Trim approach, by mitigating the risk of over-correcting $\hat{p}_{n+1}$. A schematic illustration of the construction of $\mathcal{D}_{\text{cal}}^{\text{LT}}$ by Algorithm 1 is shown in Figure 2.

The following theorem provides justification for Label-Trim, demonstrating that $\hat{p}_{n+1}^{\text{LT}}$ is an approximately valid p-value under relatively mild conditions. While our method is intuitive, this result is nontrivial for two reasons. First, Label-Trim cannot guarantee the removal of all outliers from the calibration set, as it may be that $m < n_1$ or some outliers are not among the $m$ largest scores. Second, it involves annotating the $m$ largest scores,

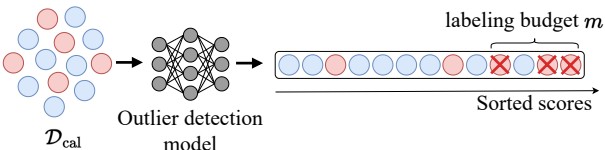

Figure 2: **A schematic illustration of the proposed active data-cleaning of the contaminated reference set (Algorithm 1).** The approach begins by computing nonconformity scores of the contaminated reference data $\mathcal{D}_{\text{cal}}$ using a pretrained outlier detection model, where blue circles denote inliers and red circles denote outliers. The scores are then sorted in increasing order, and the top $m$ samples—those most likely to be outliers—are selected for annotation. After removing the annotated outliers, the resulting (partially) cleaned set $\mathcal{D}_{\text{cal}}^{\text{LT}}$ is used for calibration.

---

**Algorithm 1** Label-trim calibration (construction phase)

1: **Input:** labeling budget $m$; contaminate calibration-set $\mathcal{D}_{\text{cal}} = \{X_i\}_{i=1}^{n}$; score function $s(\cdot)$, obtained by a pre-trained outlier detection model;
2: Compute the calibration scores $S_i = s(X_i), \forall i \in \mathcal{D}_{\text{cal}}$.
3: Sort the calibration scores, such that $S_{\pi(1)} \leq \cdots \leq S_{\pi(n)}$ where $\pi : [n] \to [n]$ is the corresponding permutation of the indices.
4: Annotate the $m$ largest scores $\mathcal{D}_{\text{labeled}} := \{(S_{\pi(i)}, Y_{\pi(i)}) : i > n - m\}$, with $Y_{\pi(i)} = 0$ if $X_{\pi(i)}$ is an inlier and $Y_{\pi(i)} = 1$ otherwise.
5: Construct the trimmed calibration set $\mathcal{D}_{\text{cal}}^{\text{LT}} = \{\pi(i) : i \leq n - m\} \cup \{j : j \in \mathcal{D}_{\text{labeled}} \text{ and } Y_j = 0\}$.
6: **Output:** trimmed calibration set $\mathcal{D}_{\text{cal}}^{\text{LT}}$.

---

**Algorithm 2** Label-trim calibration (testing phase)

1: **Input:** test point $X_{n+1}$; score function $s$; trimmed calibration set $\mathcal{D}_{\text{cal}}^{\text{LT}}$; type-I error level $\alpha$;
2: Compute the conformal p-value $\hat{p}_{n+1}^{\text{LT}}$ according to (4).
3: **Output:** reject the null hypothesis $\mathcal{H}_0$ if $\hat{p}_{n+1}^{\text{LT}} \leq \alpha$, classifying $X_{n+1}$ as an outlier.

---

revealing the true labels of some calibration points but not others, which could disrupt the exchangeability typically assumed among inlier data points in conformal inference. Therefore, this justification requires novel proof techniques and does not follow directly from existing results.

Following a notation similar to that of Lemma 2.2, let $\hat{F}_1^{\text{LT}}$ denote the empirical CDF of the scores in $\mathcal{D}_{\text{outlier}}^{\text{LT}}$. Define also $\hat{Q}_{1-\alpha}^{\text{LT}}$ as the $\hat{i}_{\text{LT}}$-th smallest element in $\{S_i\}_{i \in \mathcal{D}_{\text{cal}}^{\text{LT}}} \cup \{\infty\}$, with $\hat{i}_{\text{LT}} := \lceil (1-\alpha)(n^{\text{LT}} + 1) \rceil$ and $n^{\text{LT}} := |\mathcal{D}_{\text{cal}}^{\text{LT}}|$.

**Theorem 3.1.** *Consider the setup in* (3)*, with* $\mathcal{H}_0$ *being true.*

*For any fixed $\alpha \in (0,1)$, assume that $m \leq \alpha(n+1)$. Then,*

$$
\mathbb{P}\left(\hat{p}_{n+1}^{\mathrm{LT}} \leq \alpha\right) \leq \alpha + \frac{1}{n_0 + 1}
$$
$$
- \mathbb{E}\left[\frac{\hat{n}_1^{\mathrm{LT}}}{n_0 + 1}\left((1-\alpha) - \hat{F}_1^{\mathrm{LT}}\left(\hat{Q}_{1-\alpha}^{\mathrm{LT}}\right)\right)\right].
$$

The upper bound on the type-I error rate provided by Theorem 3.1 resembles that of Lemma 2.2 and can be interpreted as follows: `Label-Trim` produces approximately valid conformal p-values if: (i) the labeling budget is small relative to the calibration set size, i.e., $m \leq \alpha(n+1)$; and (ii) the calibration set contains a large number of inliers, $n_0$.

However, the upper bound in Theorem 3.1 also suggests that `Label-Trim` may remain overly conservative, similar to standard conformal p-values, if (i) not all outliers are removed from the calibration set ($\hat{n}_1^{\mathrm{LT}} > 0$ with high probability), and (ii) the remaining outlier scores are generally larger than the remaining inlier scores, consistent with $\mathbb{E}[\hat{F}_1^{\mathrm{LT}}(\hat{Q}_{1-\alpha}^{\mathrm{LT}})] < 1 - \alpha$, akin to Assumption 2.3.

This potential conservative behavior arises naturally from the use of a limited labeling budget, especially when the model guiding the construction of the set $\mathcal{D}_{\mathrm{labeled}}$ fails to effectively detect true outliers. Nevertheless, as we will see in the next section, `Label-Trim` often enhances power.

# 4. Experiments

We turn to evaluate the performance of conformal outlier detection methods under contaminated data. The experiments presented in this section are conducted on nine benchmark datasets: three tabular datasets, listed in Section 4.1, and six visual datasets, listed in Section 4.2.

**Methods**   We compare the following methods:

- `Standard`: The basic conformal method that uses the contaminated reference set $\mathcal{D}_{\mathrm{cal}} = \mathcal{D}_{\mathrm{inlier}} \cup \mathcal{D}_{\mathrm{outlier}}$.
- `Oracle`: An infeasible benchmark method where the reference set contains only inliers, i.e., $\mathcal{D}_{\mathrm{cal}} = \mathcal{D}_{\mathrm{inlier}}$.
- `Naive-Trim`: The baseline method from Section 3.2, which removes the top $r\%$ non-conformity scores from $\mathcal{D}_{\mathrm{cal}}$, where $r = n_1/(n_0 + n_1)$.
- `Label-Trim`: Our proposed reliable data-cleaning method from Section 3.3, applied with a labeling budget of $m = 50$ annotations to label the $m$ data points with the largest non-conformity scores from $\mathcal{D}_{\mathrm{cal}}$.
- `Small-Clean`: A baseline method that uses the labeling budget to construct a small, clean reference set by (i) randomly selecting $m$ data points from $\mathcal{D}_{\mathrm{cal}}$ and (ii) extracting the true inliers from this subset.

**Setup and performance metrics**   In all experiments, we randomly split a given dataset into disjoint training $\mathcal{D}_{\mathrm{train}}$,

calibration $\mathcal{D}_{\mathrm{cal}}$, and test sets of inliers $\mathcal{D}_{\mathrm{test}}^{\mathrm{inlier}}$ and outliers $\mathcal{D}_{\mathrm{test}}^{\mathrm{outlier}}$. To simulate a realistic setting, we construct the training and contaminated calibration sets with the same contamination rate of $r\%$. The inlier $\mathcal{D}_{\mathrm{test}}^{\mathrm{inlier}}$ and outlier $\mathcal{D}_{\mathrm{test}}^{\mathrm{outlier}}$ test sets are used to compute the type-I error and power of the outlier detection model, respectively. To ensure fair comparisons, all conformal methods use the same outlier detection model, trained on $\mathcal{D}_{\mathrm{train}}$. Performance metrics are evaluated across 100 random splits of the data. The size of each dataset, along with the details of how $\mathcal{D}_{\mathrm{train}}$, $\mathcal{D}_{\mathrm{cal}}$, and $\mathcal{D}_{\mathrm{test}}$ are constructed are provided in Appendix B.1.

## 4.1. Tabular Data

We now compare the performance of the different methods on three benchmark tabular datasets for outlier detection, previously used in the conformal literature (Bates et al., 2023). Since conclusions are similar across datasets, we focus here on results for the *shuttle* dataset (shu). Results for the *credit card* (cre) and *KDDCup99* (KDD) datasets are presented in Appendix B.2.1. For all conformal methods, we use Isolation Forest (Liu et al., 2008) as the base outlier detection model, implemented using `scikit-learn` with default hyperparameters (Buitinck et al., 2013).

Figure 3 presents the performance metrics of each method as a function of the contamination rate $r$. Following the left panel in that figure, we can see that the `Standard` conformal method results in conservative type-I error control, with a decrease in the error rate as the outlier proportion increases—a behavior that is aligned with Lemma 2.2. Notably, the type-I error of the `Oracle` method is tightly centered around $\alpha$, as guaranteed by Proposition 2.1. The `Naive-Trim` method does not control the type-I error rate, emphasizing the need for reliable data-cleaning procedures. In striking contrast, our `Label-Trim` method achieves a valid type-I error rate. At lower outlier proportions, the empirical type-I error is close to $\alpha$, but the method becomes more conservative as the outlier proportion increases. This observation aligns with the upper bound on the error rate derived in Theorem 3.1. Notably, as the contamination rate in the training data increases, the outlier detection model's ability to distinguish between inliers and outliers weakens. This, in turn, adversely affects the effectiveness of forming a subset of data points for annotation, as demonstrated in the right panel of Figure 3. The `Small-Clean` method also controls the type-I error but is more conservative than `Label-Trim` due to its much smaller reference set, which becomes even smaller as the contamination rate increases. Observe how the power of the `Small-Clean` method is lower than that of the `Standard` approach, despite the latter using a contaminated reference set. By contrast, our proposed `Label-Trim` method significantly improves the power of the `Standard` method and even achieves near-oracle performance when the outlier proportion is low.

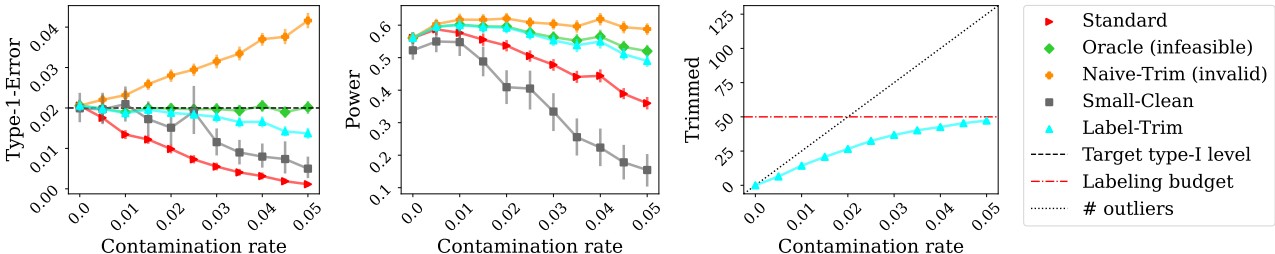

Figure 3: Comparison of conformal outlier detection methods on a tabular dataset ("shuttle") as a function of the contamination rate $r$. The target type-I error rate is $\alpha = 0.02$. Left: Empirical type-I error. Middle: Average detection rate (power), where higher values indicate better performance. Right: Number of outliers trimmed by the `Label-Trim` method. Results are averaged across 100 random splits of the data.

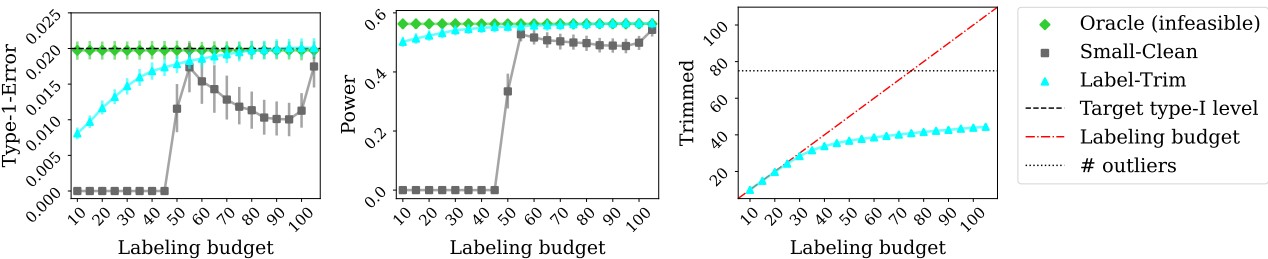

Figure 4: Comparison of conformal outlier detection methods on a tabular dataset ("shuttle") as a function of the labeling budget $m$. The contamination rate is fixed to $r = 0.03$. Other details are as in Figure 3.

Next, we study the effect of the labeling budget on the performance of our `Label-Trim` method. As shown in Figure 4, increasing the labeling budget brings the `Label-Trim` method closer to the `Oracle` in terms of both type-I error and power. Notably, even with a modest budget of 40–50 annotations, the power of `Label-Trim` is nearly indistinguishable from that of the `Oracle`. This is attributed to the method's effective trimming of outliers, as shown in the right panel. Notably, for labeling budgets $m > 50$, the condition in Theorem 3.1 no longer holds, and yet the `Label-Trim` method still achieves valid type-I error control at level $\alpha$ in practice. This highlights the robustness of the proposed method to the choice of $m$ beyond the restrictions specified in Theorem 3.1, where we attribute this robustness to the non-adversarial nature of the outlier distribution and the underlying detection model.

Figure 4 also illustrates that the `Small-Clean` method lags behind `Label-Trim` both in terms of power and conservativeness. For small labeling budgets of $m < 45$, the coarse granularity of conformal p-values (2) renders the method powerless; the smallest achievable p-value in this case is $1/(m+1) > 0.02 = \alpha$. Even for slightly larger labeling budgets, the conservative nature of the conformal p-value—specifically, the 'plus 1' term in (2)—continues to have a significant impact. This effect is rigorously quantified by the lower bound on type-I error provided in Proposition 2.1. For instance, with $m = 80$ and $\alpha = 0.02$, the lower bound is approximately $\alpha - 1/(m+1) \approx 0.0076$, which aligns closely with the empirical error rate shown in the left panel of Figure 4. Overall, these results highlight the benefits of selectively cleaning a relatively large contaminated set compared to relying on a small clean reference set, offering both improved stability and higher power.

Next, we examine how the target error level $\alpha$ affects the performance of different methods. Figure 5 shows that our `Label-Trim` method performs particularly well at low type-I error rates, especially when $\alpha$ is smaller than the contamination rate ($r = 3\%$). This behavior can be explained as follows. For a relatively accurate model, the outliers primarily distort the tail of the empirical distribution of nonconformity scores—see Figure 1. Consequently, the influence of these outliers on the rejection rule $\hat{p}_{n+1} \leq \alpha$ from (2), or $\hat{p}_{n+1}^{LT} \leq \alpha$ from (4), diminishes as $\alpha$ increases.

**Additional experiments** In Appendix B.2.1, we extend the experiments presented above. These include evaluations under higher contamination levels and additional outlier detection models, such as One-Class SVM (Schölkopf et al., 2001) and Local Outlier Factor (Breunig et al., 2000). We also examine the effect of different outlier injection strategies and the robustness of the proposed method under test-time distribution shifts in the outlier population.

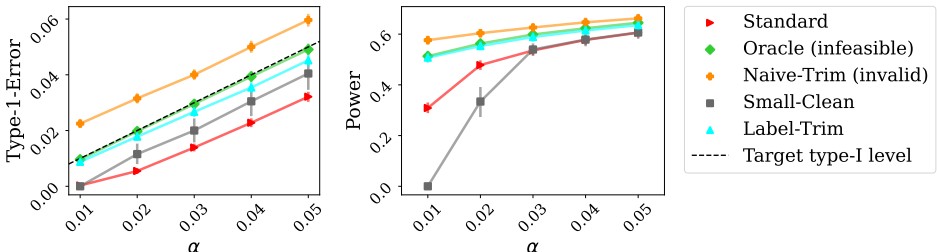

Figure 5: Comparison of conformal outlier detection methods on a tabular dataset ("shuttle") as a function of the target type-I error rate $\alpha$. The contamination rate $r$ is fixed to 3%. Other details are as in Figure 3.

## 4.2. Visual Data

In what follows, we compare all methods using benchmark visual datasets for outlier detection. Similar to Zhang et al. (2024), we construct six datasets, where the inlier samples are always images from CIFAR10 (Krizhevsky & Hinton, 2009; Krizhevsky et al., 2009) and the outlier samples vary across datasets. Specifically, the outliers are drawn from (1) MNIST (Deng, 2012), (2) SVHN (Netzer et al., 2011), (3) Texture (Cimpoi et al., 2014), (4) Places365 (Cimpoi et al., 2014), (5) TinyImageNet (Torralba et al., 2008), and (6) CIFAR100 (Krizhevsky et al., 2009). For all datasets, we use the outlier detection model proposed by Sun et al. (2021), ReAct, which operates on feature representations extracted by a pre-trained ResNet-18 model. More details are in Appendix B.1.

Table 1 summarizes the results for all six datasets. Overall, we can see a trend similar to that of the tabular data: the `Standard` and `Small-Clean` methods are valid but conservative, the `Naive-Trim` fails to control the type-I error, and our `Label-Trim` achieves a significant boost in power while practically controlling the type-I error. Notably, our `Label-Trim` method attains near-oracle performance for low contamination rates. Detailed results for each dataset, along with experiments with additional outlier detection models—specifically, the ReAct method (Sun et al., 2021) with a pre-trained VGG-19 model and the SCALE method (Xu et al., 2024) with a ResNet-18 model—are provided in Appendix B.2.2.

## 5. Discussion

In this work, we studied the robustness of conformal prediction under contaminated reference data. Motivated by empirical evidence, we characterized the conditions under which conformal outlier detection methods become too conservative. To improve power, we proposed the `Label-Trim` method, which leverages an outlier detection model and a limited labeling budget to remove outliers from the contaminated reference set. We also provided a theoretical justification for this approach, employing novel proof tech-

niques. Numerical experiments with real data confirmed that standard conformal outlier detection methods are conservative under contaminated data and demonstrated that our `Label-Trim` method can significantly enhance power.

However, the experiments also reveal a limitation of our `Label-Trim` method: while it improves power compared to standard conformal inference, it often remains too conservative, particularly when the labeling budget is very limited, leaving room for further improvement. A promising direction for future research is to enhance `Label-Trim` with *active learning* strategies (Makili et al., 2012; Fannjiang et al., 2022; Prinster et al.), enabling the removal of more outliers without increasing the labeling budget.

While we provide robustness results for calibration under contaminated reference data, our analysis assumes that the inlier calibration and test points are drawn i.i.d. from the same distribution. An important direction for future work is to extend this setting beyond the i.i.d. assumption, possibly by building on ideas from Tibshirani et al. (2019); Podkopaev & Ramdas (2021); Sesia et al. (2024); Barber et al. (2023).

Another limitation of the `Label-Trim` approach is its reliance on actively collecting new annotations. In scenarios where a flexible labeling budget is unavailable but access to a small, clean reference set is feasible, this dependency becomes restrictive. As our experiments demonstrate, the limited sample size imposes a fundamental constraint on the power of conformal outlier detection methods. This raises an intriguing question for future research: given a small clean reference set and a larger, contaminated reference set, how can we effectively and safely clean the contaminated data to enhance detection power at test time?

One potential solution could involve using the small clean reference set to calibrate a base outlier detection model. This calibrated model could then be employed to clean the larger contaminated set by removing detected outliers, while carefully accounting for inliers mistakenly classified as outliers. Exploring such a semi-supervised data-cleaning approach represents a promising direction for future work,

Table 1: Comparison of conformal outlier detection methods on six visual datasets for varying contamination rate $r$ and target type-I error level $\alpha$. The empirical type-I error values are averaged across all datasets. The empirical power is presented relative to the Standard method (higher is better), and averaged across all datasets. Results are averaged across 100 random splits of the data, with standard errors presented in parentheses.

| | Contamination rate | | | | | |
| | 1% | | 3% | | 5% | |
| Method | Power | Type-I Error | Power | Type-I Error | Power | Type-I Error |
|---|---|---|---|---|---|---|
| Standard | **1.0 (± 0.0317)** | 0.008 (± 0.0003) | **1.0 (± 0.0354)** | 0.005 (± 0.0003) | **1.0 (± 0.0408)** | 0.004 (± 0.0002) |
| Oracle (infeasible) | **1.166 (± 0.0336)** | 0.01 (± 0.0003) | **1.549 (± 0.0425)** | 0.01 (± 0.0003) | **1.961 (± 0.0531)** | 0.009 (± 0.0004) |
| Naive-Trim (invalid) | 1.659 (± 0.0342) | 0.017 (± 0.0004) | 2.79 (± 0.045) | 0.027 (± 0.0006) | 4.16 (± 0.0596) | 0.036 (± 0.0007) |
| Small-Clean | 0.0 (± 0.0) | 0.0 (± 0.0) | 0.0 (± 0.0) | 0.0 (± 0.0) | 0.0 (± 0.0) | 0.0 (± 0.0) |
| Label-Trim | **1.166 (± 0.0336)** | 0.01 (± 0.0003) | **1.517 (± 0.042)** | 0.01 (± 0.0003) | **1.786 (± 0.0498)** | 0.008 (± 0.0003) |

(a) Target type-I error rate $\alpha = 0.01$

| | Contamination rate | | | | | |
| | 1% | | 3% | | 5% | |
| Method | Power | Type-I Error | Power | Type-I Error | Power | Type-I Error |
|---|---|---|---|---|---|---|
| Standard | **1.0 (± 0.0174)** | 0.027 (± 0.0006) | **1.0 (± 0.0189)** | 0.019 (± 0.0005) | 1.0 (± 0.0212) | 0.015 (± 0.0005) |
| Oracle (infeasible) | **1.062 (± 0.0176)** | 0.03 (± 0.0006) | **1.235 (± 0.0192)** | 0.029 (± 0.0006) | **1.448 (± 0.023)** | 0.03 (± 0.0007) |
| Naive-Trim (invalid) | 1.146 (± 0.0175) | 0.035 (± 0.0006) | 1.487 (± 0.0186) | 0.043 (± 0.0007) | 1.882 (± 0.0224) | 0.052 (± 0.0008) |
| Small-Clean | 0.714 (± 0.0448) | 0.02 (± 0.0021) | 0.869 (± 0.0501) | 0.02 (± 0.002) | **1.033 (± 0.0613)** | 0.021 (± 0.0023) |
| Label-Trim | **1.041 (± 0.0177)** | 0.029 (± 0.0006) | **1.139 (± 0.019)** | 0.025 (± 0.0006) | **1.215 (± 0.0226)** | 0.021 (± 0.0006) |

(b) Target type-I error rate $\alpha = 0.03$

though we anticipate that establishing the theoretical validity of such a method may not be straightforward.

Another direction related to the above discussion is how to account for uncertainty in the labeling process. In our current formulation, we assume that all calibration points are labeled as inliers, but some may in fact be outliers. Importantly, we have no indication of which points are mislabeled, nor any signal of uncertainty in the labels. Exploring how to incorporate such uncertainty into our framework could enhance its practical utility. The line of work presented in Stutz et al. (2023); Javanmardi et al. (2024); Caprio et al. (2025) may offer a valuable starting point for such an extension: it introduces techniques for handling ambiguous labels, though in the context of multi-class classification rather than outlier detection.

Notably, our paper also provides practical guidance on how to annotate data in situations where there is uncertainty about whether a point is an inlier or an outlier. As indicated by Lemma 2.2 and supported by our empirical results, when there is uncertainty about a point's label, treating it as an inlier is a conservative strategy that preserves type-I error control. This observation suggests a compelling connection between label ambiguity (Stutz et al., 2023; Javanmardi et al., 2024; Caprio et al., 2025) and contamination in the reference set, which merits further investigation.

## Acknowledgments

M. S. was partly supported by NSF grant DMS 2210637 and by a Capital One CREDIF Research Award. Y. R. and M. B. were funded by the European Union (ERC, SafetyBounds, 101163414). Views and opinions expressed are however those of the authors only and do not necessarily reflect those of the European Union or the European Research Council Executive Agency (ERCEA). Neither the European Union nor the granting authority can be held responsible for them. This research was also partially supported by the Israel Science Foundation (ISF grant 729/21). Y. R. acknowledges additional support from the Career Advancement Fellowship at the Technion.

## Impact Statement

This paper presents work whose goal is to advance the field of Machine Learning. There are many potential societal consequences of our work, none of which we feel must be specifically highlighted here.

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

# A. Mathematical Proofs

## A.1. Auxiliary Technical Results

In this section, we begin by introducing two useful propositions, Proposition A.1 and Proposition A.2, which will be used later in the proofs presented here.

**Proposition A.1.** *Let $\mathcal{D}$ be a dataset containing $n$ scores, and define the threshold $\hat{Q}_{1-\alpha}$ as*

$$\hat{Q}_{1-\alpha} := \hat{i}\text{-th smallest element in } \mathcal{D} \cup \{\infty\},$$

*where*

$$\hat{i} := \lceil (1-\alpha)(n+1) \rceil.$$

*For any test point $X_{n+1}$, the following holds:*

$$s(X_{n+1}) > \hat{Q}_{1-\alpha} \quad \text{if and only if} \quad \hat{p}_{n+1} \leq \alpha,$$

*where $\hat{p}_{n+1}$ is the conformal p-value* (2).

*Proof of Proposition A.1.* The proof follows the definition of conformal p-value from (2), and its relation to the empirical quantile function:

$$\hat{p}_{n+1} = \frac{1 + \sum_{i=1}^{n} \mathbb{I}[s(X_i) \geq s(X_{n+1})]}{n+1} \leq \alpha \qquad \overset{(i)}{\Longleftrightarrow}$$

$$\hat{p}_{n+1} = \frac{1 + \sum_{i=1}^{n} \mathbb{I}[s(X_i) \geq s(X_{n+1})]}{n+1} \leq \frac{\lfloor \alpha(n+1) \rfloor}{n+1} \qquad \Longleftrightarrow$$

$$1 + \sum_{i=1}^{n} \mathbb{I}[s(X_i) \geq s(X_{n+1})] \leq \lfloor \alpha(n+1) \rfloor \qquad \Longleftrightarrow$$

$$1 + n - \sum_{i=1}^{n} \mathbb{I}[s(X_i) < s(X_{n+1})] \leq \lfloor \alpha(n+1) \rfloor \qquad \Longleftrightarrow$$

$$\sum_{i=1}^{n} \mathbb{I}[s(X_i) < s(X_{n+1})] \geq n + 1 - \lfloor \alpha(n+1) \rfloor \overset{(ii)}{\Longleftrightarrow}$$

$$\sum_{i=1}^{n} \mathbb{I}[s(X_i) < s(X_{n+1})] \geq \lceil (1-\alpha)(n+1) \rceil \qquad (5)$$

The labeled steps above can be explained as follows.

- (i) The values of $\hat{p}_{n+1}$ are discrete, taking values from $\{\frac{1}{n+1}, \frac{2}{n+1}, \ldots, 1\}$. Therefore, $\hat{p}_{n+1} = \frac{k}{n+1}$ for some $k \in [n+1]$. We explicitly prove that $\hat{p}_{n+1} \leq \alpha$ iff $\hat{p}_{n+1} \leq \frac{\lfloor \alpha(n+1) \rfloor}{n+1}$ as follows:

  $\Leftarrow$ Assume $\hat{p}_{n+1} \leq \frac{\lfloor \alpha(n+1) \rfloor}{n+1}$. Therefore, $\hat{p}_{n+1} \leq \frac{\lfloor \alpha(n+1) \rfloor}{n+1} \leq \frac{\alpha(n+1)}{n+1} = \alpha$.

  $\Rightarrow$ Assume $\hat{p}_{n+1} \leq \alpha$, then $\frac{k}{n+1} \leq \alpha$. This implies that $k \leq \alpha(n+1)$. Since $k$ is an integer, it follows that $k \leq \lfloor \alpha(n+1) \rfloor$. Therefore, $\hat{p}_{n+1} = \frac{k}{n+1} \leq \frac{\lfloor \alpha(n+1) \rfloor}{n+1}$.

- (ii) This step follows directly from the equality $n + 1 = \lceil (1-\alpha)(n+1) \rceil + \lfloor \alpha(n+1) \rfloor$. We explicitly prove this equality as follows:

  – The term $\lceil (1-\alpha)(n+1) \rceil$ represents the smallest integer greater than or equal to $(1-\alpha)(n+1)$. Hence, we can write:

$$\lceil (1-\alpha)(n+1) \rceil = (1-\alpha)(n+1) + \delta_1,$$

  where $0 \leq \delta_1 < 1$.

   – Similarly, the term $\lfloor \alpha(n+1) \rfloor$ represents the largest integer less than or equal to $\alpha(n+1)$. Thus:

$$\lfloor \alpha(n+1) \rfloor = \alpha(n+1) - \delta_2,$$

where $0 \leq \delta_2 < 1$.

   – Adding these two terms gives:

$$\lfloor \alpha(n+1) \rfloor + \lceil (1-\alpha)(n+1) \rceil = \alpha(n+1) - \delta_2 + (1-\alpha)(n+1) + \delta_1 = n + 1 + (\delta_1 - \delta_2).$$

Since $\lfloor \alpha(n+1) \rfloor + \lceil (1-\alpha)(n+1) \rceil$ must be an integer and $\delta_1, \delta_2 \in [0,1)$, it follows that $\delta_1 - \delta_2 = 0$.

Therefore, $\lfloor \alpha(n+1) \rfloor + \lceil (1-\alpha)(n+1) \rceil = n+1$.

To complete the proof, we now show that (5) holds if and only if $s(X_{n+1}) > \hat{Q}_{1-\alpha}$.

$\Leftarrow$ Assume $s(X_{n+1}) > \hat{Q}_{1-\alpha}$. By definition, $\sum_{i=1}^{n} \mathbb{I}[s(X_i) \leq \hat{Q}_{1-\alpha}] = \lceil (1-\alpha)(n+1) \rceil$. Then, (5) holds since

$$\sum_{i=1}^{n} \mathbb{I}[s(X_i) < s(X_{n+1})] \geq \sum_{i=1}^{n} \mathbb{I}[s(X_i) \leq \hat{Q}_{1-\alpha}] = \lceil (1-\alpha)(n+1) \rceil.$$

$\Rightarrow$ We prove this direction by contradiction, assuming that (5) holds. Now, suppose that $s(X_{n+1}) \leq \hat{Q}_{1-\alpha}$ also holds, implying that

$$\sum_{i=1}^{n} \mathbb{I}[s(X_i) < s(X_{n+1})] \leq \sum_{i=1}^{n} \mathbb{I}[s(X_i) < \hat{Q}_{1-\alpha}] \overset{(i)}{<} \lceil (1-\alpha)(n+1) \rceil,$$

which contradicts the assumption (5). Therefore, we conclude that $s(X_{n+1}) > \hat{Q}_{1-\alpha}$. The last step above can be explained as follows.

   – (i) Recall that by definition, $\hat{Q}_{1-\alpha}$ is a specific value in $\{s(X_i)\}_{i=1}^{n}$ and $\sum_{i=1}^{n} \mathbb{I}[s(X_i) \leq \hat{Q}_{1-\alpha}] = \lceil (1-\alpha)(n+1) \rceil$. This implies that

$$\sum_{i=1}^{n} \mathbb{I}[s(X_i) < \hat{Q}_{1-\alpha}] = \lceil (1-\alpha)(n+1) \rceil - \sum_{i=1}^{n} \mathbb{I}[s(X_i) = \hat{Q}_{1-\alpha}] < \lceil (1-\alpha)(n+1) \rceil.$$

In sum, $\hat{p}_{n+1} \leq \alpha$ holds if and only if (5) holds, and the latter holds if and only if $s(X_{n+1}) > \hat{Q}_{1-\alpha}$. This completes the proof. □

**Proposition A.2.** *Let $\mathcal{D}$ be a dataset containing $n$ scores, and let $S_{n+1}$ be a test score. Define the following thresholds:*

$$\hat{Q}_{1-\alpha} := \hat{i}\text{-th smallest element in } \mathcal{D} \cup \{\infty\},$$

*where*

$$\hat{i} := \lceil (1-\alpha)(n+1) \rceil.$$

*Similarly, let*

$$\hat{Q}_{1-\alpha}^{n+1} := \hat{i}\text{-th smallest element in } \mathcal{D} \cup \{S_{n+1}\}.$$

*It follows that*

$$\hat{Q}_{1-\alpha} \geq \hat{Q}_{1-\alpha}^{n+1} \text{ almost surely.}$$

*Proof of Proposition A.2.* Since the largest possible score is $\infty$, the set $\mathcal{D} \cup \{\infty\}$ almost surely contains scores that are greater or equal to those in $\mathcal{D} \cup \{S_{n+1}\}$. Consequently, $\hat{Q}_{1-\alpha} := \hat{i}$-th smallest element in $\mathcal{D} \cup \{\infty\}$ is almost surely greater than or equal to $\hat{Q}_{1-\alpha}^{n+1} := \hat{i}$-th smallest element in $\mathcal{D} \cup \{S_{n+1}\}$. □

## A.2. Explaining the Conservativeness of Standard Conformal p-Values

### A.2.1. PROOF OF LEMMA 2.2

*Proof of Lemma 2.2.* To simplify the notation define the random score $S_i := s(X_i)$ for all $i \in \mathcal{D}_{\text{cal}} \cup \{n+1\}$. Throughout the proof, we refer to the calibration set as the set of nonconformity scores corresponding to the calibration points. Without loss of generality, assume that the inliers in $\mathcal{D}_{\text{cal}}$ are located at the first $n_0$ indices. Let $\mathcal{D}_{\text{inlier}} = [n_0]$ denote the set of indices corresponding to the inlier scores. Consequently, define $\mathcal{D}_{\text{outlier}} = \{n_0 + 1, n_0 + 2, \ldots, n\}$ as the set of indices corresponding to the outlier scores in $\mathcal{D}_{\text{cal}}$. We assume the scores have no ties (which can always be achieved by adding a negligible random noise to the scores output by any model).

Given a fixed realization of the score vector $(s_1, \ldots, s_n, s_{n+1}) \in \mathbb{R}^{n+1}$, define the following two events:

- $E_{\text{in}}$: the unordered set of inlier scores, including the test score, is $\{S_1, \ldots, S_{n_0}, S_{n+1}\} = \{s_1, \ldots, s_{n_0}, s_{n+1}\}$;

- $E_{\text{out}}$: the unordered set of outlier scores is $\{S_{n_0+1}, \ldots, S_n\} = \{s_{n_0+1}, \ldots, s_n\}$.

Under the setup defined in (3), when $\mathcal{H}_0$ is true, the test score $S_{n+1}$ and the inlier scores in the calibration set are i.i.d. from $\mathbb{P}_0$. Therefore, by exchangeability, the following holds for each inlier index $i \in \mathcal{D}_{\text{inlier}} \cup \{n+1\}$:

$$\mathbb{P}\left(S_{n+1} = s_i \mid E_{\text{in}}, E_{\text{out}}\right) = \frac{1}{n_0 + 1}. \tag{6}$$

Since the calibration scores are almost-surely distinct, the probability of a null test point obtaining any outlier score is zero. Therefore, for each outlier index $j \in \mathcal{D}_{\text{outlier}}$:

$$\mathbb{P}\left(S_{n+1} = s_j \mid E_{\text{in}}, E_{\text{out}}\right) = 0. \tag{7}$$

To obtain an upper bound on the type-I error rate, $\mathbb{P}\left(\hat{p}_{n+1} \leq \alpha\right)$, we use the equivalence established in Proposition A.1. According to this result, the following holds:

$$\mathbb{P}\left(\hat{p}_{n+1} \leq \alpha\right) = \mathbb{P}\left(S_{n+1} > \hat{Q}_{1-\alpha}^{\text{cal}}\right),$$

where $\hat{Q}_{1-\alpha}^{\text{cal}}$ is the $\hat{i}_{\text{cal}}$-th smallest element in $\{S_i\}_{i=1}^n \cup \{\infty\}$ and $\hat{i}_{\text{cal}} := \lceil (1-\alpha)(n+1) \rceil$.

Moreover, define $\hat{Q}_{1-\alpha}^{n+1}$ as the $\hat{i}_{\text{cal}}$-th smallest score in $\{S_i\}_{i=1}^{n+1}$. By Proposition A.2, $\hat{Q}_{1-\alpha}^{\text{cal}} \geq \hat{Q}_{1-\alpha}^{n+1}$ almost surely.

Now, we obtain an upper bound for $\mathbb{P}\left(S_{n+1} > \hat{Q}_{1-\alpha}^{\text{cal}} \mid E_{\text{in}}, E_{\text{out}}\right)$, where the probability is taken over random permutations

of the scores conditional on $E_{\text{in}}, E_{\text{out}}$.

$$
\begin{aligned}
\mathbb{P}\left(S_{n+1} > \hat{Q}^{\text{cal}}_{1-\alpha} \mid E_{\text{in}}, E_{\text{out}}\right) &\leq \mathbb{P}\left(S_{n+1} > \hat{Q}^{\text{n+1}}_{1-\alpha} \mid E_{\text{in}}, E_{\text{out}}\right) \\
&= \mathbb{E}\left[\mathbb{I}\left[S_{n+1} > \hat{Q}^{\text{n+1}}_{1-\alpha}\right] \mid E_{\text{in}}, E_{\text{out}}\right] \\
&= \sum_{i \in \mathcal{D}_{\text{inlier}} \cup \{n+1\}} \mathbb{E}\left[\mathbb{I}[S_{n+1} = s_i]\, \mathbb{I}\left[s_i > \hat{Q}^{\text{n+1}}_{1-\alpha}\right] \mid E_{\text{in}}, E_{\text{out}}\right] \\
&\stackrel{(i)}{=} \sum_{i \in \mathcal{D}_{\text{inlier}} \cup \{n+1\}} \mathbb{I}\left[s_i > \hat{Q}^{\text{n+1}}_{1-\alpha}\right] \mathbb{P}\left(S_{n+1} = s_i \mid E_{\text{in}}, E_{\text{out}}\right) \\
&= \frac{1}{n_0 + 1} \sum_{i \in \mathcal{D}_{\text{inlier}} \cup \{n+1\}} \mathbb{I}\left[s_i > \hat{Q}^{\text{n+1}}_{1-\alpha}\right] \\
&\stackrel{(ii)}{\leq} \frac{1}{n_0 + 1}\left(\alpha(n+1) - \sum_{i \in \mathcal{D}_{\text{outlier}}} \mathbb{I}\left[s_i > \hat{Q}^{\text{n+1}}_{1-\alpha}\right]\right) \\
&= \alpha + \frac{1}{n_0 + 1}\left(\alpha n_1 - \sum_{i \in \mathcal{D}_{\text{outlier}}} \mathbb{I}\left[s_i > \hat{Q}^{\text{n+1}}_{1-\alpha}\right]\right) \\
&= \alpha - \frac{n_1}{n_0 + 1}\left(1 - \alpha - \frac{1}{n_1} \sum_{i \in \mathcal{D}_{\text{outlier}}} \mathbb{I}\left[s_i \leq \hat{Q}^{\text{n+1}}_{1-\alpha}\right]\right) \\
&\stackrel{(iii)}{\leq} \alpha - \frac{n_1}{n_0 + 1}\left(1 - \alpha - \frac{1}{n_1} \sum_{i \in \mathcal{D}_{\text{outlier}}} \mathbb{I}\left[s_i \leq \hat{Q}^{\text{cal}}_{1-\alpha}\right]\right) \\
&= \alpha - \frac{n_1}{n_0 + 1}\left(1 - \alpha - \hat{F}_1\left(\hat{Q}^{\text{cal}}_{1-\alpha}\right)\right),
\end{aligned}
$$

where $\hat{F}_1$ is the empirical CDF of the outlier scores. The labeled steps above can be explained as follows.

- (i) $\mathbb{I}\left[s_i > \hat{Q}^{\text{n+1}}_{1-\alpha}\right]$ is measurable with respect to the $\sigma$-algebra generated by $E_{\text{in}}, E_{\text{out}}$. This follows because $\hat{Q}^{\text{n+1}}_{1-\alpha}$ is the $\hat{i}_{\text{cal}}$-th smallest element of $\{s_1, \ldots, s_{n+1}\}$, which is fully determined by these variables. Thus, we can pull it out of the expectation.

- (ii) $\hat{Q}^{\text{n+1}}_{1-\alpha}$ is the $\hat{i}_{\text{cal}}$-th smallest score in $\{s_1, \ldots, s_n, s_{n+1}\}$ and $[n+1] = \mathcal{D}_{\text{outlier}} \cup \mathcal{D}_{\text{inlier}} \cup \{n+1\}$. By definition,

$$
\sum_{i \in \mathcal{D}_{\text{inlier}} \cup \{n+1\}} \mathbb{I}\left[s_i > \hat{Q}^{\text{n+1}}_{1-\alpha}\right] + \sum_{i \in \mathcal{D}_{\text{outlier}}} \mathbb{I}\left[s_i > \hat{Q}^{\text{n+1}}_{1-\alpha}\right] = \sum_{i=1}^{n+1} \mathbb{I}\left[s_i > \hat{Q}^{\text{n+1}}_{1-\alpha}\right] = \lfloor \alpha(n+1) \rfloor \leq \alpha(n+1)
$$

and therefore,

$$
\sum_{i \in \mathcal{D}_{\text{inlier}} \cup \{n+1\}} \mathbb{I}\left[s_i > \hat{Q}^{\text{n+1}}_{1-\alpha}\right] \leq \alpha(n+1) - \sum_{i \in \mathcal{D}_{\text{outlier}}} \mathbb{I}\left[s_i > \hat{Q}^{\text{n+1}}_{1-\alpha}\right]
$$

- (iii) Since $\hat{Q}^{\text{cal}}_{1-\alpha} \geq \hat{Q}^{\text{n+1}}_{1-\alpha}$ almost surely, increasing the threshold (i.e., using $\hat{Q}^{\text{cal}}_{1-\alpha}$) results in an equal or larger value of the sum.

Now, we can derive an upper bound for $\mathbb{P}\left(\hat{p}_{n+1} \leq \alpha\right)$ as follows:

$$
\begin{aligned}
\mathbb{P}\left(\hat{p}_{n+1} \leq \alpha\right) &= \mathbb{P}\left(S_{n+1} > \hat{Q}_{1-\alpha}^{\mathrm{cal}}\right) \\
&= \mathbb{E}\left[\mathbb{P}\left(S_{n+1} > \hat{Q}_{1-\alpha}^{\mathrm{cal}} \mid E_{\mathrm{in}}, E_{\mathrm{out}}\right)\right] \\
&\leq \mathbb{E}\left[\alpha - \frac{n_1}{n_0 + 1}\left(1 - \alpha - \hat{F}_1\left(\hat{Q}_{1-\alpha}^{\mathrm{cal}}\right)\right)\right] \\
&= \alpha - \frac{n_1}{n_0 + 1}\left(1 - \alpha - \mathbb{E}\left[\hat{F}_1\left(\hat{Q}_{1-\alpha}^{\mathrm{cal}}\right)\right]\right),
\end{aligned}
$$

with this expectation being taken over different realizations of the inlier and outlier nonconformity scores. $\qquad\square$

### A.2.2. AN ALTERNATIVE VIEW BASED ON MIXTURE DISTRIBUTIONS

Next, we provide an additional theoretical result concerning the conservativeness of conformal outlier detection methods, to supplement the result presented in Section 2.3 from a point of view closer to that of Sesia et al. (2024).

Specifically, we consider a contaminated calibration set, $\mathcal{D}_{\mathrm{cal}}$, which may include both inliers (samples i.i.d. from $\mathbb{P}_0$) and outliers (samples i.i.d. from $\mathbb{P}_1 \neq \mathbb{P}_0$). The goal remains to test the null hypothesis $\mathcal{H}_0$ that a new data point $X_{n+1}$ is an inlier, independently sampled from $\mathbb{P}_0$.

This setup differs from (3) in that the calibration set is drawn from a mixed distribution, where the proportion of outliers in the population is denoted by $\delta \in [0, 1)$. Hence, the numbers of inliers and outliers in the calibration set are random, rather than fixed. Formally, this setup is expressed as:

$$
\begin{aligned}
X_i &\overset{\mathrm{i.i.d.}}{\sim} \mathbb{P}_{\mathrm{mixed}} = (1 - \delta) \cdot \mathbb{P}_0 + \delta \cdot P_1, \quad \forall i \in \mathcal{D}_{\mathrm{cal}}, \\
\mathcal{H}_0 &: X_{n+1} \overset{\mathrm{ind.}}{\sim} \mathbb{P}_0.
\end{aligned}
\tag{8}
$$

Let $F_0$ and $F_1$ denote the CDFs of $\mathbb{P}_0$ and $\mathbb{P}_1$, respectively, and $\hat{Q}_{1-\alpha}^{\mathrm{cal}}$ represent the $\lceil(1 - \alpha)(n + 1)\rceil$-th smallest score in the calibration set.

**Corollary A.3** (Conservativeness). *Under the setup defined in* (8)*, if $\mathcal{H}_0$ is true, then, for any $\alpha \in (0, 1)$,*

$$
\mathbb{P}\left(\hat{p}_{n+1} \leq \alpha\right) \leq \alpha - \delta\, \mathbb{E}\left[F_0(\hat{Q}_{1-\alpha}^{\mathrm{cal}}) - F_1(\hat{Q}_{1-\alpha}^{\mathrm{cal}})\right].
$$

Corollary A.3 reformulates Theorem 1 in Sesia et al. (2024) under the setup in (8). This result quantifies the behavior of conformal outlier detection methods in the presence of contaminated data and establishes guarantees on the type-I error rate. This result complements our analysis of the conservativeness of these methods.

*Proof of Corollary A.3.* The proof adapts the argument of Theorem 1 in Sesia et al. (2024) to the outlier detection setting considered here. Specifically, we follow the structure of the original proof, making adjustments to account for the presence of inliers and outliers in the calibration set.

By Proposition A.1, we have

$$
\hat{p}_{n+1} \leq \alpha \iff S_{n+1} > \hat{Q}_{1-\alpha}^{\mathrm{cal}}.
$$

Under the null, we upper bound $\mathbb{P}\left(\hat{p}_{n+1} \leq \alpha\right)$ as follows:

$$
\begin{aligned}
\mathbb{P}_0\left(\hat{p}_{n+1} \leq \alpha\right) &= \mathbb{P}_0\left(\hat{p}_{n+1} \leq \alpha\right) + \mathbb{P}_{\mathrm{mixed}}\left(\hat{p}_{n+1} \leq \alpha\right) - \mathbb{P}_{\mathrm{mixed}}\left(\hat{p}_{n+1} \leq \alpha\right) \\
&= \mathbb{P}_{\mathrm{mixed}}\left(\hat{p}_{n+1} \leq \alpha\right) - \left[\mathbb{P}_{\mathrm{mixed}}\left(\hat{p}_{n+1} \leq \alpha\right) - \mathbb{P}_0\left(\hat{p}_{n+1} \leq \alpha\right)\right] \\
&\leq \alpha - \left[\mathbb{P}_{\mathrm{mixed}}\left(\hat{p}_{n+1} \leq \alpha\right) - \mathbb{P}_0\left(\hat{p}_{n+1} \leq \alpha\right)\right] \\
&= \alpha - \left[\left((1-\delta)\,\mathbb{P}_0\left(\hat{p}_{n+1} \leq \alpha\right) + \delta\,\mathbb{P}_1\left(\hat{p}_{n+1} \leq \alpha\right)\right) - \mathbb{P}_0\left(\hat{p}_{n+1} \leq \alpha\right)\right] \\
&= \alpha - \delta\left[\mathbb{P}_1\left(\hat{p}_{n+1} \leq \alpha\right) - \mathbb{P}_0\left(\hat{p}_{n+1} \leq \alpha\right)\right] \\
&= \alpha - \delta\left[\mathbb{P}_1\left(S_{n+1} > \hat{Q}_{1-\alpha}^{\mathrm{cal}}\right) - \mathbb{P}_0\left(S_{n+1} > \hat{Q}_{1-\alpha}^{\mathrm{cal}}\right)\right] \\
&= \alpha - \delta\left[\mathbb{P}_0\left(S_{n+1} \leq \hat{Q}_{1-\alpha}^{\mathrm{cal}}\right) - \mathbb{P}_1\left(S_{n+1} \leq \hat{Q}_{1-\alpha}^{\mathrm{cal}}\right)\right] \\
&= \alpha - \delta\,\mathbb{E}\left[\mathbb{P}_0\left(S_{n+1} \leq \hat{Q}_{1-\alpha}^{\mathrm{cal}} \mid \mathcal{D}_{\mathrm{cal}}\right) - \mathbb{P}_1\left(S_{n+1} \leq \hat{Q}_{1-\alpha}^{\mathrm{cal}} \mid \mathcal{D}_{\mathrm{cal}}\right)\right] \\
&= \alpha - \delta\,\mathbb{E}\left[F_0\left(\hat{Q}_{1-\alpha}^{\mathrm{cal}}\right) - F_1\left(\hat{Q}_{1-\alpha}^{\mathrm{cal}}\right)\right].
\end{aligned}
$$

$\square$

## A.3. Validity of the `Label-Trim` Method

### A.3.1. PROOF OF THEOREM 3.1 — MAIN STEPS

*Proof of Theorem 3.1.* As in the proof of Lemma 2.2, define the random score $S_i := s(X_i)$ for all $i \in \mathcal{D}_{\mathrm{cal}} \cup \{n+1\}$. By Proposition A.1, for any fixed $\alpha \in (0,1)$, the probability of a type-I error, $\mathbb{P}\left(\hat{p}_{n+1}^{\mathrm{LT}} \leq \alpha\right)$, can be expressed as

$$
\mathbb{P}\left(\hat{p}_{n+1}^{\mathrm{LT}} \leq \alpha\right) = \mathbb{P}\left(S_{n+1} > \hat{Q}_{1-\alpha}^{\mathrm{LT}}\right).
$$

Consider the augmented set $\{S_i\}_{i \in \mathcal{D}_{\mathrm{cal}}^{\mathrm{LT}}} \cup \{S_{n+1}\}$, which includes the test score $S_{n+1}$. Define $\hat{Q}_{1-\alpha}^{\mathrm{LT},n+1}$ as follows:

$$
\hat{Q}_{1-\alpha}^{\mathrm{LT},n+1} := \hat{i}_{\mathrm{LT}}\text{-th smallest element in } \{S_i\}_{i \in \mathcal{D}_{\mathrm{cal}}^{\mathrm{LT}}} \cup \{S_{n+1}\}.
$$

By Proposition A.2, it holds that $\hat{Q}_{1-\alpha}^{\mathrm{LT}} \geq \hat{Q}_{1-\alpha}^{\mathrm{LT},n+1}$ almost surely.

Now, consider an imaginary "mirror" version of this method that applies the label-trim algorithm with two key differences:

- it uses a larger labeling budget, $\tilde{m} = m + 1$;
- it treats $\{S_i\}_{i=1}^{n+1}$ as the calibration set instead of $\{S_i\}_{i=1}^{n}$—that is, it includes the test point in the annotation process, preserving the exchangeability with the calibration inliers.

Let $\tilde{\mathcal{D}}_{\mathrm{cal}}^{\mathrm{LT}} \cup \{n+1\}$ denote the indices of the trimmed augmented calibration set produced by the mirror procedure, let $\tilde{\mathcal{D}}_{\mathrm{labeled}}$ denote the indices of the corresponding labeled data points, and define $\tilde{\mathcal{D}}_{\mathrm{labeled}}^{\mathrm{inlier}}, \tilde{\mathcal{D}}_{\mathrm{labeled}}^{\mathrm{outlier}}$ as the corresponding subsets of inliers and outliers, respectively. Under this mirror procedure, the empirical quantile $\hat{Q}_{1-\alpha}^{\mathrm{LT},n+1}$ corresponds to

$$
\tilde{Q}_{1-\alpha}^{\mathrm{LT},n+1} := \tilde{i}_{\mathrm{LT}}\text{-th smallest element in } \{S_i\}_{i \in \tilde{\mathcal{D}}_{\mathrm{cal}}^{\mathrm{LT}}} \cup \{S_{n+1}\},
$$

where $\tilde{i}_{\mathrm{LT}} := \lceil (1-\alpha)(\tilde{n}^{\mathrm{LT}} + 1) \rceil$ and $\tilde{n}_{\mathrm{LT}} := |\tilde{\mathcal{D}}_{\mathrm{cal}}^{\mathrm{LT}}|$.

By construction of this mirror procedure, $\tilde{i}_{\mathrm{LT}} \leq \hat{i}_{\mathrm{LT}}$ almost surely, because $\tilde{n}^{\mathrm{LT}} \leq n^{\mathrm{LT}}$ almost surely and thus

$$
\tilde{i}_{\mathrm{LT}} = \lceil (1-\alpha)(\tilde{n}^{\mathrm{LT}} + 1) \rceil \leq \lceil (1-\alpha)(n^{\mathrm{LT}} + 1) \rceil = \hat{i}_{\mathrm{LT}}.
$$

Using the fact that $\tilde{i}_{\mathrm{LT}} \leq \hat{i}_{\mathrm{LT}}$ almost surely, we prove in Appendix A.3.2 that, almost surely,

$$
\tilde{Q}_{1-\alpha}^{\mathrm{LT},n+1} \leq \hat{Q}_{1-\alpha}^{\mathrm{LT},n+1}. \tag{9}
$$

Since we already knew that $\hat{Q}_{1-\alpha}^{\text{LT}} \geq \hat{Q}_{1-\alpha}^{\text{LT},n+1}$, this implies:

$$\hat{Q}_{1-\alpha}^{\text{LT}} \geq \hat{Q}_{1-\alpha}^{\text{LT},n+1} \geq \tilde{Q}_{1-\alpha}^{\text{LT},n+1}. \tag{10}$$

Therefore, the type-I error rate of the Label-Trim approach can be bounded from above by the type-I error rate of the mirror procedure, which can be studied with an approach similar to that of the proof of Lemma 2.2.

Let $\tilde{\mathcal{D}}_{\text{outlier}}^{\text{LT}}$ denote the outlier indices remaining in $\tilde{\mathcal{D}}_{\text{cal}}^{\text{LT}}$, with $\tilde{n}_1^{\text{LT}} = |\tilde{\mathcal{D}}_{\text{outlier}}^{\text{LT}}|$. As in the proof of Lemma 2.2, define $E_{\text{in}}$ and $E_{\text{out}}$ as two unordered realizations of the inlier and outlier scores in $\{S_i\}_{i=1}^{n+1}$, respectively. Appendix A.3.3 proves that

$$\mathbb{P}\left(S_{n+1} > \hat{Q}_{1-\alpha}^{\text{LT}} \mid E_{\text{in}}, E_{\text{out}}, \tilde{\mathcal{D}}_{\text{labeled}}, \tilde{\mathcal{D}}_{\text{labeled}}^{\text{inlier}}\right) \leq \alpha + \frac{1}{n_0 + 1} - \frac{\hat{n}_1^{\text{LT}}}{n_0 + 1}\left((1 - \alpha) - \hat{F}_1^{\text{LT}}(\hat{Q}_{1-\alpha}^{\text{LT}})\right), \tag{11}$$

from which it follows immediately, by marginalizing over $E_{\text{in}}, E_{\text{out}}, \tilde{\mathcal{D}}_{\text{labeled}}$, and $\tilde{\mathcal{D}}_{\text{labeled}}^{\text{inlier}}$, that

$$\mathbb{P}\left(S_{n+1} > \hat{Q}_{1-\alpha}^{\text{LT}}\right) \leq \mathbb{E}\left[\alpha + \frac{1}{n_0 + 1} - \frac{\hat{n}_1^{\text{LT}}}{n_0 + 1}\left((1 - \alpha) - \hat{F}_1^{\text{LT}}\left(\hat{Q}_{1-\alpha}^{\text{LT}}\right)\right)\right].$$

$$\square$$

### A.3.2. PROOF OF THEOREM 3.1 — PROOF OF EQUATION (9)

*Proof.* We prove (9) by analyzing two distinct cases, depending on whether $S_{n+1}$ is among the $m + 1$ largest scores or not. Recall that $m \leq \alpha(n + 1)$, by assumption, and $n^{\text{LT}} \leq n$. It is easy to see that this implies that, almost surely,

$$\hat{i}_{\text{LT}} = \lceil (1 - \alpha)(n^{\text{LT}} + 1) \rceil \leq n + 1 - (m + 1). \tag{12}$$

The mirror procedure labels and potentially removes the largest $m + 1$ scores out of the $n + 1$ total scores. Thus, $\hat{Q}_{1-\alpha}^{\text{LT},n+1}$ is smaller than all the outliers removed during trimming, i.e., $\hat{Q}_{1-\alpha}^{\text{LT},n+1} < S_j$ for all $S_j \in \tilde{\mathcal{D}}_{\text{labeled}}^{\text{outlier}}$.

- Suppose $S_{n+1}$ is among the $m + 1$ largest scores in $\{S_i\}_{i=1}^{n+1}$. This case is illustrated below:

  In this scenario, the mirror trimming approach additionally labels the test score $S_{n+1}$, which, under the null hypothesis, is an inlier and is thus not removed. As a result, both trimming procedures yield the same set; i.e., $\tilde{\mathcal{D}}_{\text{cal}}^{\text{LT}} = \mathcal{D}_{\text{cal}}^{\text{LT}}$. Consequently, $\hat{Q}_{1-\alpha}^{\text{LT},n+1} = \tilde{Q}_{1-\alpha}^{\text{LT},n+1}$.

  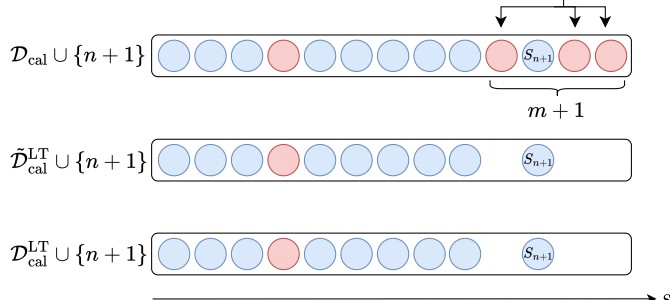

- Suppose $S_{n+1}$ is not among the $m + 1$ largest scores in $\{S_i\}_{i=1}^{n+1}$. Within this case, there are two sub-cases to consider.

  - The $(m + 1)$-th largest score in an inlier:

    In this case, the mirror trimming approach additionally labels an inlier, which is not removed. As a result, both trimming procedures yield the same set; i.e., $\tilde{\mathcal{D}}_{\text{cal}}^{\text{LT}} = \mathcal{D}_{\text{cal}}^{\text{LT}}$. Consequently, $\hat{Q}_{1-\alpha}^{\text{LT},n+1} = \tilde{Q}_{1-\alpha}^{\text{LT},n+1}$.

    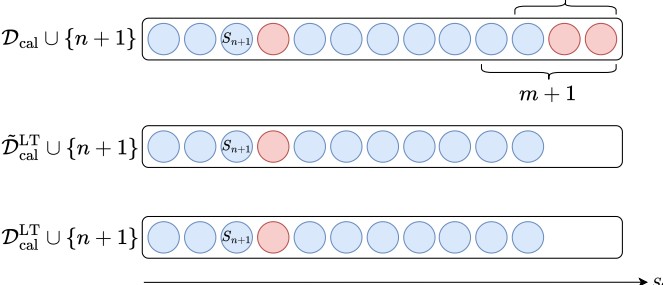

– The $(m+1)$-th largest score in an outlier:

This is the interesting case where $\tilde{n}^{\text{LT}} = n^{\text{LT}} - 1$ and the set $\tilde{\mathcal{D}}_{\text{cal}}^{\text{LT}}$ contains one fewer outlier score than $\mathcal{D}_{\text{cal}}^{\text{LT}}$. It follows from (12) that the threshold $\hat{Q}_{1-\alpha}^{\text{LT,n+1}}$ is smaller than the $m+1$ largest scores. Then, in this region, $\mathcal{D}_{\text{cal}}^{\text{LT}}$ and $\tilde{\mathcal{D}}_{\text{cal}}^{\text{LT}}$ contain the same scores. Therefore, the $\hat{i}_{\text{LT}}$-th smallest score corresponds to the same score in both $\mathcal{D}_{\text{cal}}^{\text{LT}} \cup \{n+1\}$ and $\tilde{\mathcal{D}}_{\text{cal}}^{\text{LT}} \cup \{n+1\}$. Since $\tilde{i}_{\text{LT}} \leq \hat{i}_{\text{LT}}$, it follows that $\tilde{Q}_{1-\alpha}^{\text{LT,n+1}} \leq \hat{Q}_{1-\alpha}^{\text{LT,n+1}}$.

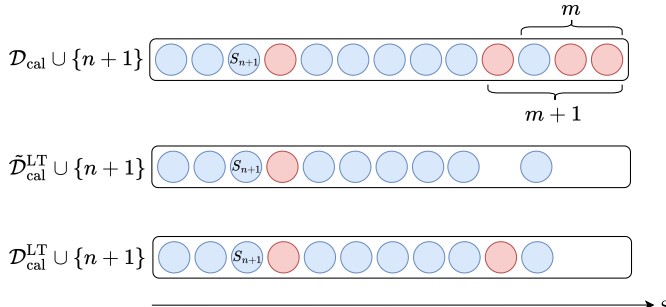

$\square$

### A.3.3. PROOF OF THEOREM 3.1 — PROOF OF EQUATION (11)

*Proof.*

$$\mathbb{P}\left(S_{n+1} > \hat{Q}_{1-\alpha}^{\text{LT}} \mid E_{\text{in}}, E_{\text{out}}, \tilde{\mathcal{D}}_{\text{labeled}}, \tilde{\mathcal{D}}_{\text{labeled}}^{\text{inlier}}\right)$$

$$\leq \mathbb{P}\left(S_{n+1} > \tilde{Q}_{1-\alpha}^{\text{LT,n+1}} \mid E_{\text{in}}, E_{\text{out}}, \tilde{\mathcal{D}}_{\text{labeled}}, \tilde{\mathcal{D}}_{\text{labeled}}^{\text{inlier}}\right)$$

$$= \sum_{i \in \mathcal{D}_{\text{inlier}} \cup \{n+1\}} \mathbb{E}\left[\mathbb{I}\left[s_i > \tilde{Q}_{1-\alpha}^{\text{LT,n+1}}\right] \cdot \mathbb{I}\left[S_{n+1} = s_i\right] \mid E_{\text{in}}, E_{\text{out}}, \tilde{\mathcal{D}}_{\text{labeled}}, \tilde{\mathcal{D}}_{\text{labeled}}^{\text{inlier}}\right]$$

$$\overset{(i)}{=} \sum_{i \in \mathcal{D}_{\text{inlier}} \cup \{n+1\}} \mathbb{I}\left[s_i > \tilde{Q}_{1-\alpha}^{\text{LT,n+1}}\right] \cdot \mathbb{P}\left(S_{n+1} = s_i \mid E_{\text{in}}, E_{\text{out}}, \tilde{\mathcal{D}}_{\text{labeled}}, \tilde{\mathcal{D}}_{\text{labeled}}^{\text{inlier}}\right)$$

$$\overset{(ii)}{=} \sum_{i \in \mathcal{D}_{\text{inlier}} \cup \{n+1\}} \mathbb{I}\left[s_i > \tilde{Q}_{1-\alpha}^{\text{LT,n+1}}\right] \cdot \mathbb{P}\left(S_{n+1} = s_i \mid E_{\text{in}}, E_{\text{out}}\right)$$

$$= \frac{1}{n_0 + 1} \sum_{i \in \mathcal{D}_{\text{inlier}} \cup \{n+1\}} \mathbb{I}\left[s_i > \tilde{Q}_{1-\alpha}^{\text{LT,n+1}}\right]$$

$$\overset{(iii)}{\leq} \frac{1}{n_0 + 1} \left(\alpha(\tilde{n}^{\text{LT}} + 1) - \sum_{i \in \tilde{\mathcal{D}}_{\text{outlier}}^{\text{LT}}} \mathbb{I}\left[s_i > \tilde{Q}_{1-\alpha}^{\text{LT,n+1}}\right]\right)$$

$$\overset{(iv)}{\leq} \frac{1}{n_0 + 1} \left(\alpha(n^{\text{LT}} + 1) - \sum_{i \in \mathcal{D}_{\text{outlier}}^{\text{LT}}} \mathbb{I}\left[s_i > \tilde{Q}_{1-\alpha}^{\text{LT,n+1}}\right] + 1\right)$$

$$\overset{(v)}{\leq} \frac{1}{n_0 + 1} \left(\alpha(n^{\text{LT}} + 1) - \sum_{i \in \mathcal{D}_{\text{outlier}}^{\text{LT}}} \mathbb{I}\left[s_i > \hat{Q}_{1-\alpha}^{\text{LT}}\right] + 1\right)$$

$$= \frac{1}{n_0 + 1} \left(\alpha(n^{\text{LT}} + 1) - \hat{n}_1^{\text{LT}} + \sum_{i \in \mathcal{D}_{\text{outlier}}^{\text{LT}}} \mathbb{I}\left[s_i \leq \hat{Q}_{1-\alpha}^{\text{LT}}\right] + 1\right)$$

$$= \alpha + \frac{1}{n_0 + 1} - \frac{\hat{n}_1^{\text{LT}}}{n_0 + 1}\left((1 - \alpha) - \hat{F}_1^{\text{LT}}(\hat{Q}_{1-\alpha}^{\text{LT}})\right).$$

The labeled steps above can be explained as follows.

- (i) $\mathbb{I}\left[s_i > \tilde{Q}_{1-\alpha}^{\text{LT,n+1}}\right]$ is measurable with respect to the $\sigma$-algebra generated by $E_{\text{in}}, E_{\text{out}}, \tilde{\mathcal{D}}_{\text{labeled}}$, and $\tilde{\mathcal{D}}_{\text{labeled}}^{\text{inlier}}$ since $\tilde{Q}_{1-\alpha}^{\text{LT,n+1}}$ is the $\tilde{i}_{\text{LT}}$-th smallest element of $\{s_1, \ldots, s_{n+1}\} \setminus \left(\tilde{\mathcal{D}}_{\text{labeled}} \setminus \tilde{\mathcal{D}}_{\text{labeled}}^{\text{inlier}}\right)$.

- (ii) The mirror procedure is applied on $\{S_i\}_{i=1}^{n+1}$, preserving the exchangeability of the test score $S_{n+1}$ with the calibration inliers. Hence, the resulting labeled sets $\tilde{\mathcal{D}}_{\text{labeled}}$ and $\tilde{\mathcal{D}}_{\text{labeled}}^{\text{inlier}}$ contain no additional information about $S_{n+1}$ beyond $E_{\text{in}}, E_{\text{out}}$.

- (iii) By definition, $\tilde{Q}_{1-\alpha}^{\text{LT},n+1}$ is the $\tilde{i}_{\text{LT}}$-th smallest element of $\{S_i\}_{i\in\tilde{\mathcal{D}}_{\text{cal}}^{\text{LT}}} \cup \{S_{n+1}\}$, where $\tilde{i}_{\text{LT}} = \lceil (1-\alpha)(\tilde{n}^{\text{LT}}+1) \rceil$. Consequently, $\lfloor \alpha(\tilde{n}^{\text{LT}}+1) \rfloor$ scores in $\{S_i\}_{i\in\tilde{\mathcal{D}}_{\text{cal}}^{\text{LT}}} \cup \{S_{n+1}\}$ are larger than $\tilde{Q}_{1-\alpha}^{\text{LT},n+1}$.

- (iv) The set $\tilde{\mathcal{D}}_{\text{outlier}}^{\text{LT}}$ is either equal to $\mathcal{D}_{\text{outlier}}^{\text{LT}}$ or contains one fewer outlier, and $\tilde{n}^{\text{LT}} \in \{n^{\text{LT}}, n^{\text{LT}}-1\}$ almost surely.

- (v) Recall from (10) that $\hat{Q}_{1-\alpha}^{\text{LT}} \geq \tilde{Q}_{1-\alpha}^{\text{LT},n+1}$ almost surely.

$\square$

# B. Supplementary Experiments and Implementation Details

## B.1. Datasets

Table 2 summarizes details of the three tabular benchmark datasets. For all tabular datasets, we perform random subsampling to construct contaminated train, calibration, and test sets. Specifically, for the shuttle and KDDCup99 datasets, the train set contains 5,000 samples, and the calibration set contains 2,500 samples, both with a contamination rate of $r = 3\%$, unless stated otherwise. The inlier and outlier test sets consist of 950 and 50 samples, respectively. For the credit-card dataset, the train set contains 2,000 samples, while the calibration and test sets follow the same setup as the Shuttle and KDDCup99 datasets.

Table 2: Summary of tabular datasets

| Dataset | Shuttle (shu) | Credit-card (cre) | KDDCup99 (KDD) |
|---|---|---|---|
| Total Samples | 58,000 | 284,807 | 494,020 |
| Number of Outliers | 12,414 | 492 | 396,743 |
| Number of Features | 9 | 29 | 41 |

For visual datasets, we use the OpenOOD benchmark (Zhang et al., 2024; Yang et al., 2022). Specifically, for each dataset and contamination rate, we perform a one-time training of an outlier detection model, which operates on feature representations extracted from a pre-trained backbone. We include results with the following model configurations:

- **ReAct with ResNet-18**: ReAct (Sun et al., 2021) operates on feature representations extracted from a pre-trained ResNet-18 model (Zhang et al., 2024; He et al., 2016). The model applies a percentile-based threshold (set to 90%) to truncate activations, where the threshold is computed on the contaminated train set. These truncated activations then pass through the fully connected layer of the pre-trained ResNet-18. The outlier score is computed using an energy-based log-sum-exp function applied to these truncated activations.

- **ReAct with VGG-19**: Same as above, but with a pre-trained VGG-19 backbone (Chen) instead of ResNet-18.

- **SCALE with ResNet-18**: SCALE (Xu et al., 2024) operates on feature representations extracted from a pre-trained ResNet-18 model (Zhang et al., 2024; He et al., 2016). The model rescales the activations using a sample-specific factor, defined as the sum of all activations divided by the sum of activations below a certain percentile (set to 65%). Similar to ReAct, the outlier score is computed using an energy-based log-sum-exp function applied to the rescaled activations.

After training, we save the outlier scores for the remaining outlier samples and the CIFAR-10 test set. We randomly subsample this pool of scores to construct the calibration and test sets, ensuring that all sets are disjoint.

The sizes of the train and calibration sets are 2,000 and 3,000, respectively, with the same contamination rate. The inlier and outlier test sets consist of 950 and 50 samples, respectively.

## B.2. Additional Experiments

### B.2.1. TABULAR DATASETS

In this section, we provide additional real-data experiments conducted on the *credit card* (cre) and *KDDCup99* datasets, complementing the analysis provided for the shuttle dataset in the main manuscript. Figures 6 to 12 corresponds to and extends the figures presented in the main text.

We further report supplementary results across all three tabular datasets considered in this work, including: additional outlier detection models (Figures 13 and 14); higher contamination levels (Figure 15); strategic outlier injection (Figures 16 to 19); and test-time distributional shifts in the outlier population (Figure 20).

**Results as a function of the contamination rate** In Figure 3 of the main manuscript, we analyze the performance of conformal outlier methods as a function of the contamination rate $r$. Here, we repeat the same experiment on the credit-card

and KDDCup99 datasets (Figures 6 and 7). The performance trends are similar to the one presented in the main manuscript: both the `Standard` and `Small-Clean` methods achieve valid type-I error rate but exhibit conservative behavior. In contrast, the `Naive-Trim` method fails to control the type-I error rate. The `Label-Trim` method attains improved power while practically controlling the type-I error at level $\alpha$.

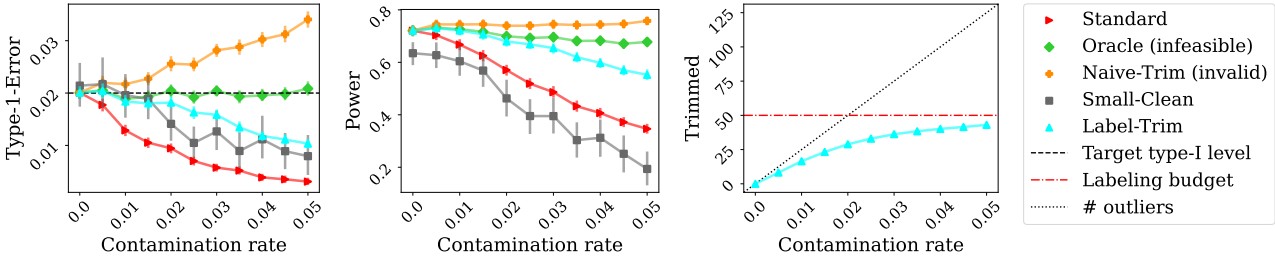

Figure 6: Comparison of conformal outlier detection methods on real dataset "credit-card" as a function of the contamination rate $r$. Other details are as in Figure 3.

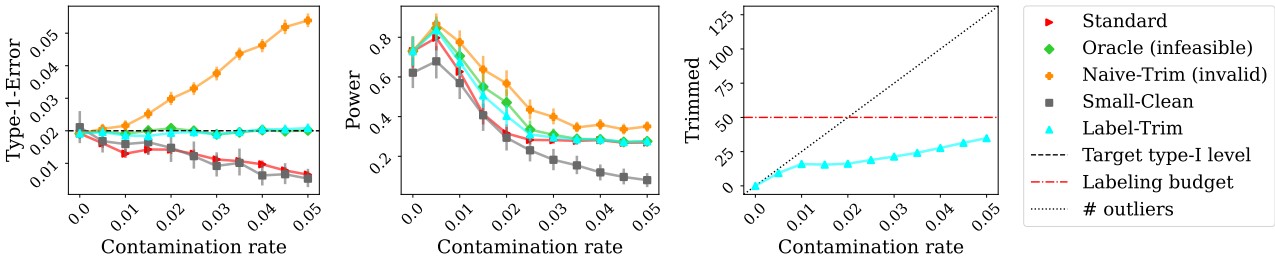

Figure 7: Comparison of conformal outlier detection methods on real dataset "KDDCup99" as a function of the contamination rate $r$. Other details are as in Figure 3.

**Results as a function of the labeling budget**   In Figure 4 of the main manuscript, we evaluate the performance of the `Label-Trim`, `Small-Clean` and, `Oracle` methods as a function of the labeling budget $m$. Figures 8 and 9 extend this analysis to the credit-card and KDDCup99 datasets, respectively. Consistent with the trends observed in Figure 4, increasing the labeling budget improves the performance of the `Label-Trim` method in terms of both type-I error and power. Notably, although the condition in Theorem 3.1 is no longer satisfied for $m > 50$, the `Label-Trim` method still maintains valid type-I error control at the desired level $\alpha$.

For the KDDCup99 dataset, however, we observe that the `Small-Clean` method shows higher power than the `Oracle` across several labeling budgets ($m \geq 55$). This is due to the high variability in the dataset and the small sample size used by this method, resulting in significant variance in type-I error. To illustrate this, Figure 10 presents a box plot showing the variability and instability of the `Small-Clean` method in this regime. While this leads to a higher average power, this variability is undesirable in practice.

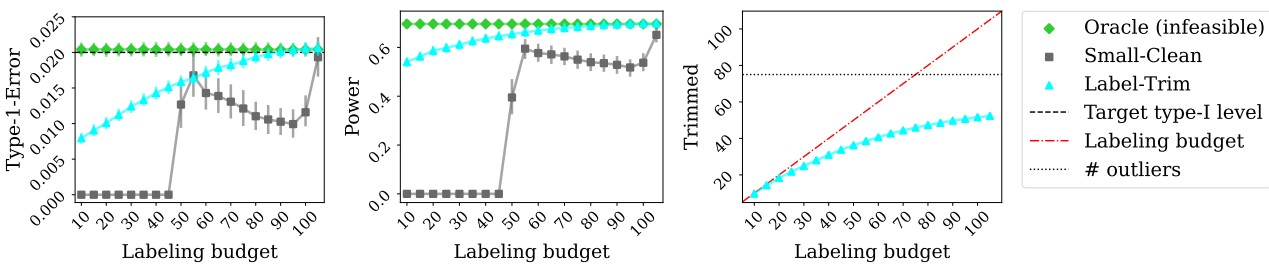

Figure 8: Performance on real dataset "credit-card" as a function of the labeling budget $m$. The contamination rate is $r = 0.03$. Other details are as in Figure 3.

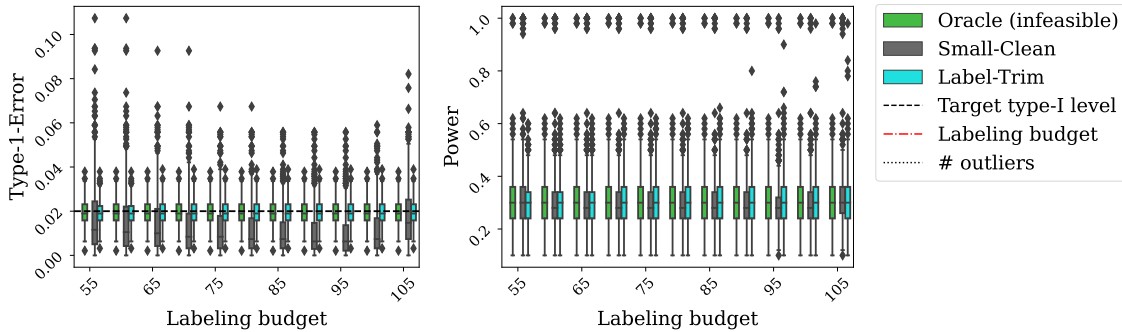

Figure 9: Performance on real dataset "KDDCup99" as a function of the labeling budget $m$. The contamination rate is $r = 0.03$. Results are averages across 400 random splits of the data. Other details are as in Figure 3.

Figure 10: Performance on real dataset "KDDCup99" as a function of the labeling budget $m$. The contamination rate is $r = 0.03$. Results are averages across 400 random splits of the data. Other details are as in Figure 3.

**Results as a function of the target type-I error level** In Figure 5 we examine the performance of conformal outlier detection methods as a function of the target type-I error level $\alpha$. Here, we replicate the experiments on the credit-card and KDDCup99 datasets (Figures 11 and 12). In line with the trends observed in Figure 5, the `Label-Trim` method performs well at low type-I error rates. Notably, for $\alpha = 0.01$, the `Label-Trim` method outperforms the baselines, while practically controlling the type-I error at level $\alpha$, even though the condition of Theorem 3.1 is not satisfied in this case. This highlights the robustness of our approach.

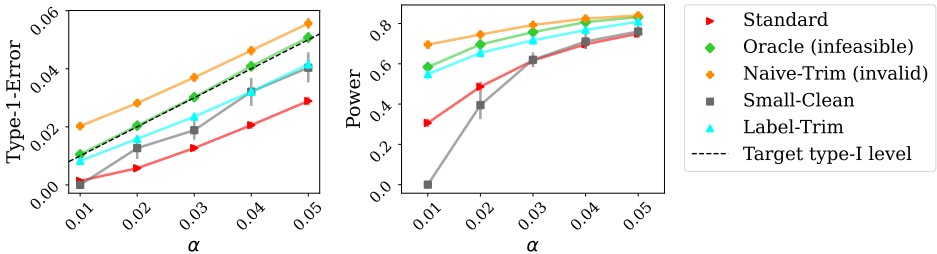

Figure 11: Comparison of conformal outlier detection methods on real dataset "credit-card" as a function of the target type-I error rate $\alpha$. The contamination rate $r$ is fixed to 3%; other details are as in Figure 3.

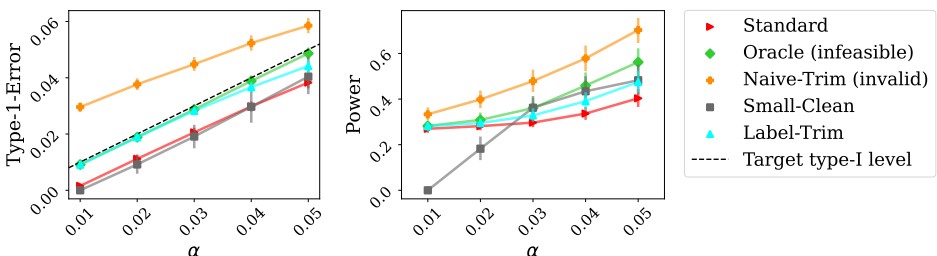

Figure 12: Comparison of conformal outlier detection methods on real dataset "KDDCup99" as a function of the target type-I error rate $\alpha$. The contamination rate $r$ is fixed to 3%; other details are as in Figure 3.

**Results with additional outlier detection models** Here, we consider two additional outlier detection models: Local Outlier Factor (LOF) with 100 estimators and One-Class Support Vector Machine (OC-SVM) with an RBF kernel, both implemented via `scikit-learn`. Figures 13 and 14 present the performance of all methods when applied with LOF and OC-SVM, respectively. As can be seen, the proposed `Label-Trim` controls the type-I error across both models. When the outlier detection model is less discriminative (e.g., LOF or OC-SVM on the KDDCup99 dataset), the candidate set becomes less effective and power decreases. Nevertheless, our method improves over the baselines, including the `Small-Clean` method.

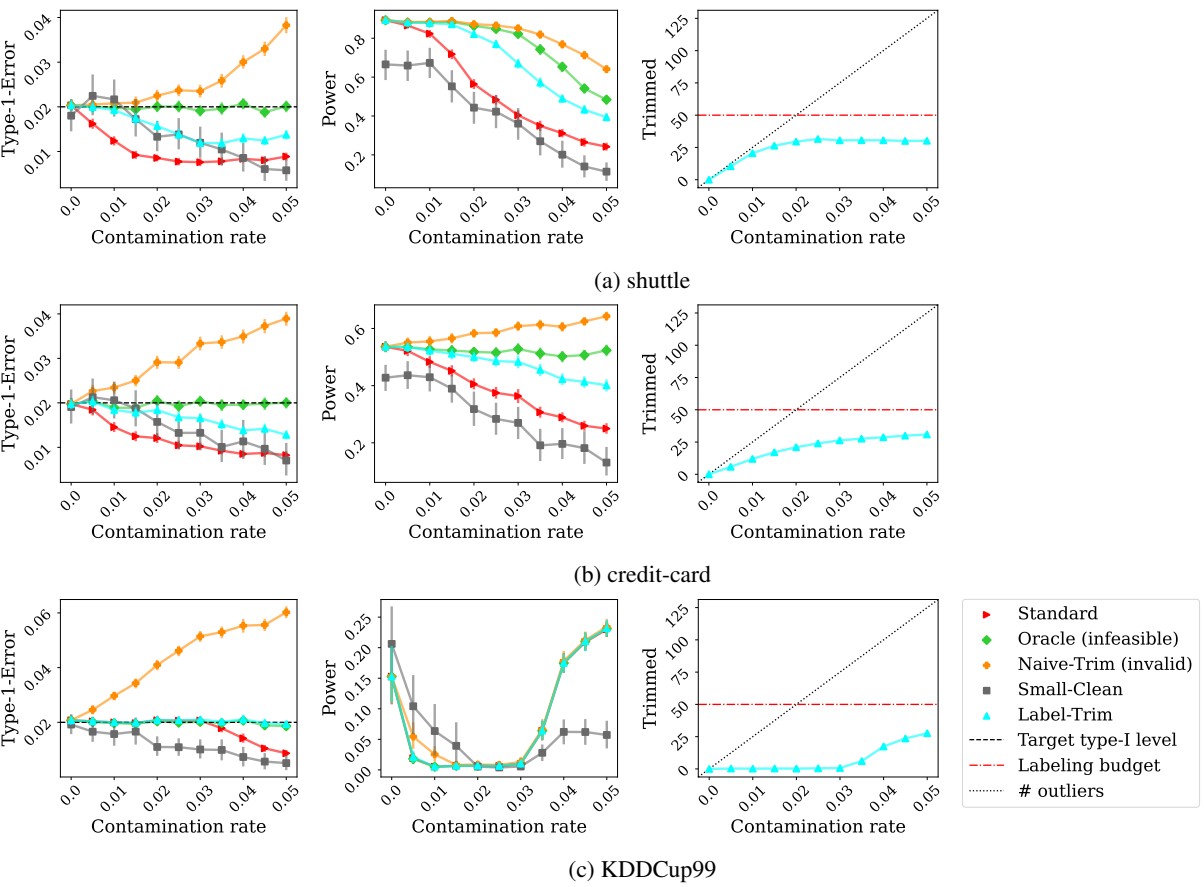

Figure 13: Comparison of conformal outlier detection methods on real datasets as a function of the contamination rate $r$. All methods utilize a Local Outlier Factor model. Other details are as in Figure 3 in the main manuscript.

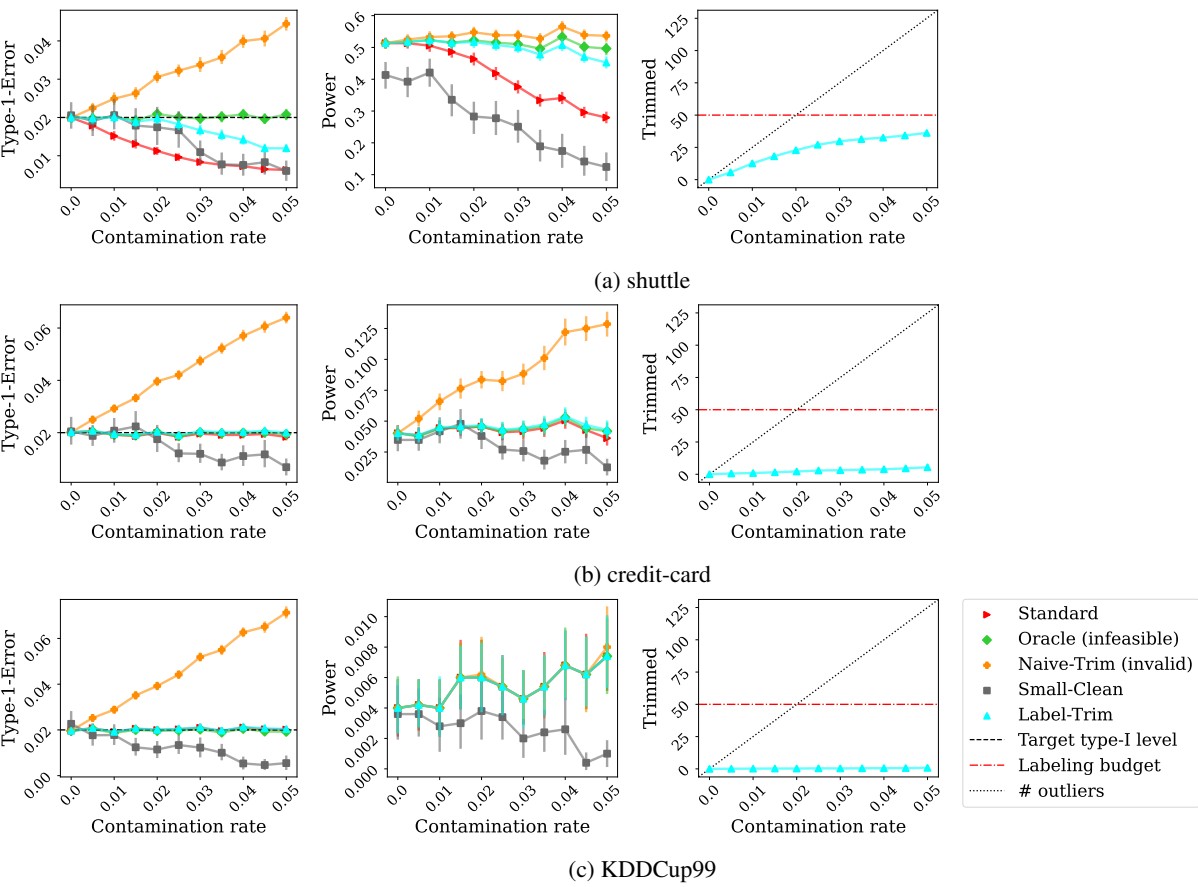

(a) shuttle

(b) credit-card

(c) KDDCup99

Figure 14: Comparison of conformal outlier detection methods on real datasets as a function of the contamination rate $r$. All methods utilize a One-Class SVM model. Other details are as in Figure 3 in the main manuscript.

**Results with higher contamination rate**  Figure 15 shows the performance of conformal outlier detection methods as a function of the contamination rate $r$, evaluated at higher levels than those considered in the main manuscript. The performance trend aligns with that observed at lower contamination rates, across all datasets. Both the `Standard` and `Small-Clean` methods control the type-I error at the nominal level but remain overly conservative. In contrast, `Naive-Trim` fails to control the type-I error. The proposed `Label-Trim` method achieves improved power while controlling the type-I error at level $\alpha$.

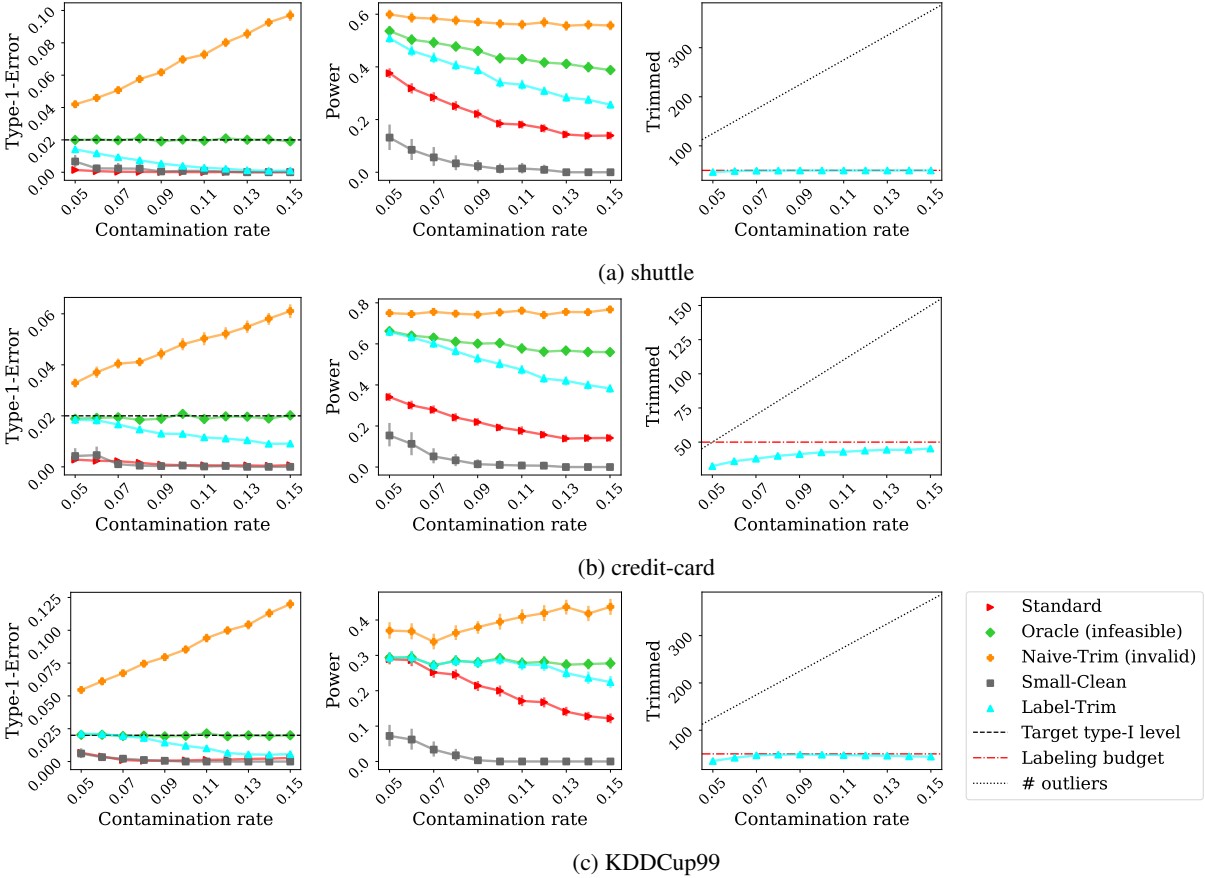

(a) shuttle

(b) credit-card

(c) KDDCup99

Figure 15: Comparison of conformal outlier detection methods on real tabular datasets as a function of the contamination rate $r$. Other details are as in Figure 3 in the main manuscript.

**Results with strategic outlier injection**  We extend the experimental setting from the main manuscript by evaluating different outlier injection strategies. Instead of injecting outliers at random, we selected outliers that more closely resemble inliers—specifically, outliers whose nonconformity scores fall below a given score percentile. Figure 16 illustrates the nonconformity scores for both outliers and inliers under different outlier injection strategies for the "shuttle" dataset, highlighting that lower-percentile outliers increasingly resemble inliers. This sets the stage to evaluate the performance of our proposed method under these challenging settings. Figure 17 presents the performance of all conformal methods on the same dataset and outlier injection strategies, as a function of the target type-I error level. Following that figure, we see that `Label-Trim` controls the type-I error while improving power. Similar trends hold on the credit-card and KDDCup99 datasets (Figures 18 and 19).

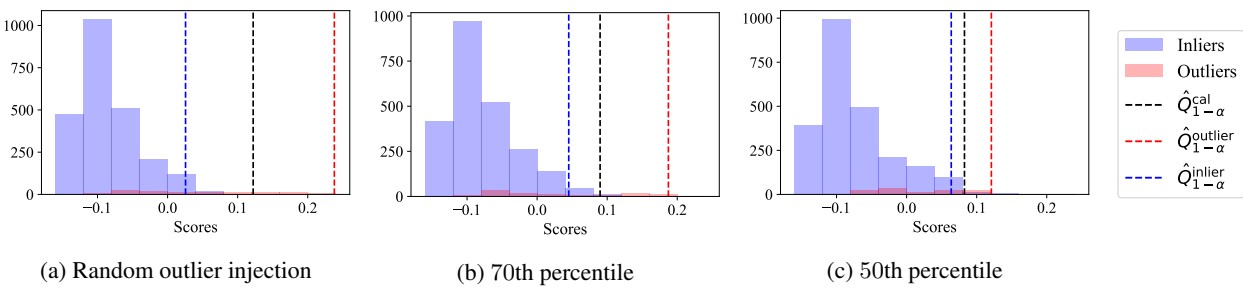

(a) Random outlier injection  (b) 70th percentile  (c) 50th percentile

Figure 16: Histogram of nonconformity scores for inliers and outliers in a contaminated calibration subset of the "shuttle" data, with a contamination rate of 5% for different outlier injection strategies.

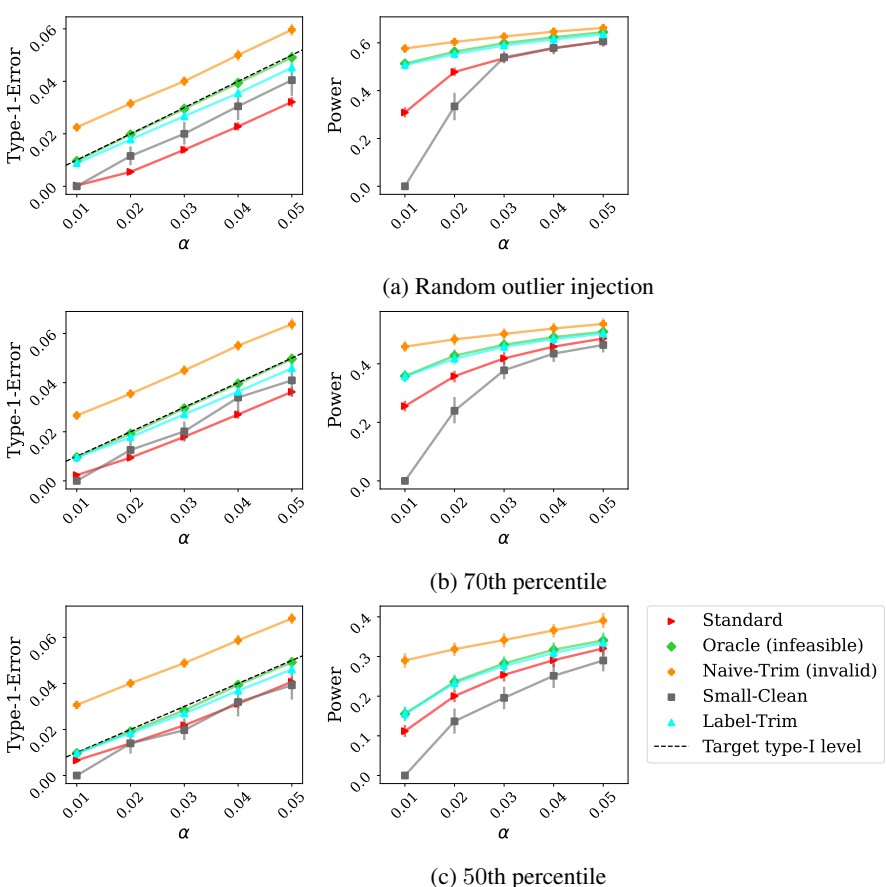

(a) Random outlier injection

(b) 70th percentile

(c) 50th percentile

Figure 17: Performance on a real dataset "shuttle" as a function of the target type-I error rate $\alpha$ for different outlier injection strategies. Other details are as in Figure 5 in the main manuscript.

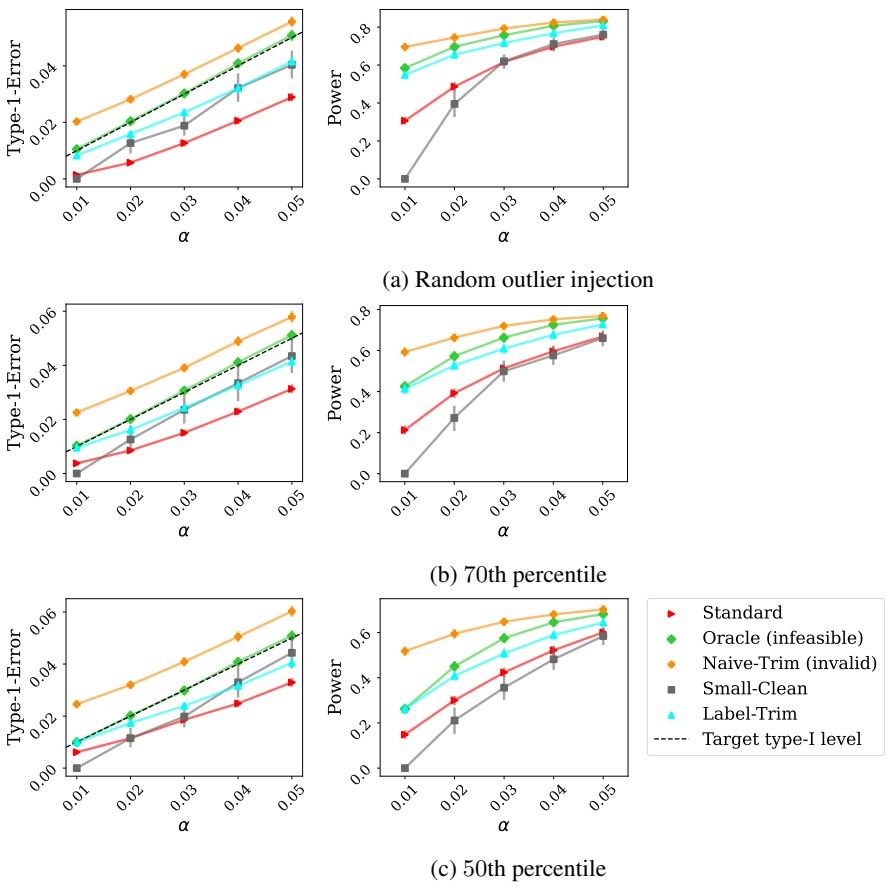

(a) Random outlier injection

(b) 70th percentile

(c) 50th percentile

Figure 18: Performance on a real dataset "credit-card" as a function of the target type-I error rate $\alpha$ for different outlier injection strategies. Other details are as in Figure 5 in the main manuscript.

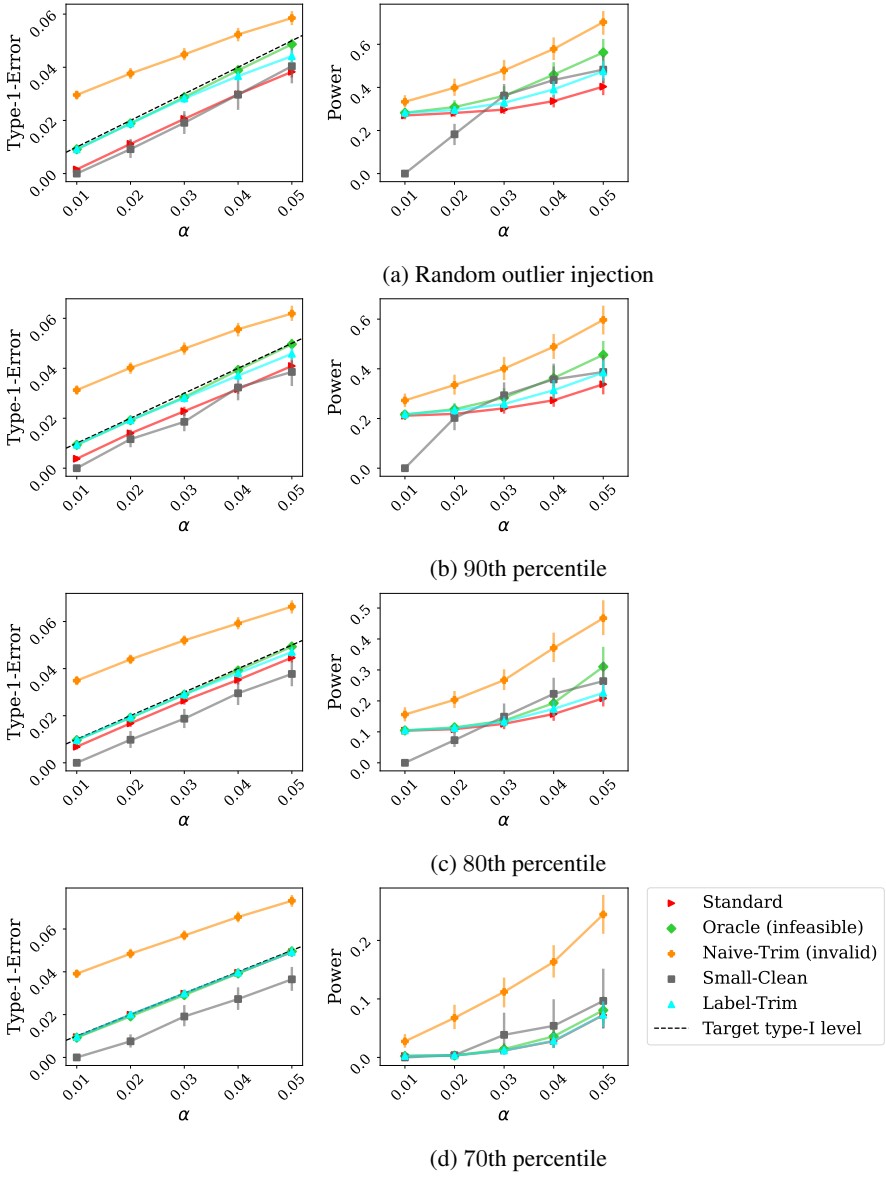

(a) Random outlier injection

(b) 90th percentile

(c) 80th percentile

(d) 70th percentile

Figure 19: Performance on a real dataset "KDDCup99" as a function of the target type-I error rate $\alpha$ for different outlier injection strategies. Other details are as in Figure 5 in the main manuscript.

**Results with test-time drifting outliers** We further investigate the robustness of the proposed method under test-time distribution shift in the outlier population. Using the injection strategies described earlier, we simulate a shift on the "shuttle" dataset where the outlier distribution gradually changes over time. Specifically, while the training and calibration sets contain high-percentile outliers, the test set progressively includes more challenging, low-percentile outliers. As shown in Figure 20, `Label-Trim` maintains type-I error control throughout the distribution shift, outperforming all baselines and approaching oracle performance.

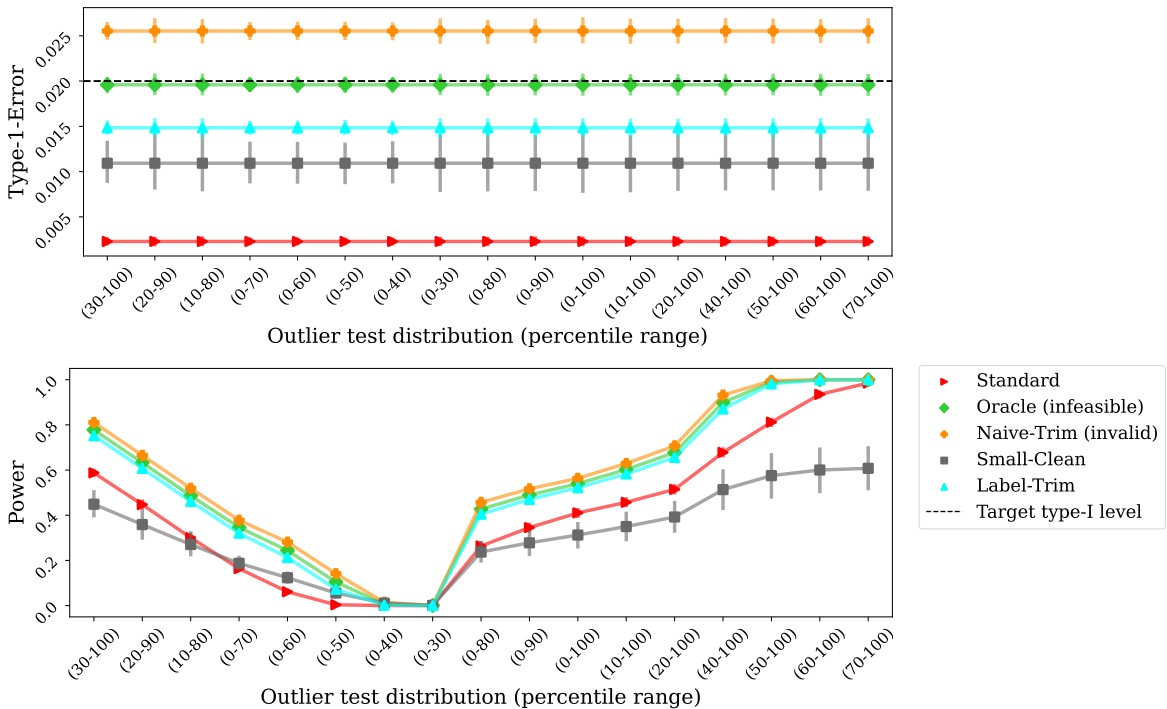

Figure 20: Performance on a real dataset "shuttle" as a function of drift in the outlier test distribution. Outliers in the training and calibration sets are drawn from the 30th to 100th percentile of the outlier score distribution, while the test set varies across different percentile ranges. Other details are as in Figure 3 in the main manuscript.

### B.2.2. VISUAL DATASETS

Table 3 summarizes the results across all six datasets for a target type-I error rate of $\alpha = 0.02$, showing trends consistent with those observed in Table 1 of the main manuscript.

Table 3: Comparison of conformal outlier detection methods on six visual datasets for varying contamination rate $r$ and target type-I error level $\alpha = 0.02$. The empirical type-I error values are averaged across all datasets. The empirical power is presented relative to the `Standard` method (higher is better), and averaged across all datasets. Results are averaged across 100 random splits of the data, with standard errors presented in parentheses.

| | Contamination rate | | | | | |
| | 1% | | 3% | | 5% | |
| Method | Power | Type-I Error | Power | Type-I Error | Power | Type-I Error |
|---|---|---|---|---|---|---|
| Standard | **1.0 ($\pm$ 0.0204)** | 0.017 ($\pm$ 0.0004) | **1.0 ($\pm$ 0.0238)** | 0.012 ($\pm$ 0.0004) | **1.0 ($\pm$ 0.0272)** | 0.009 ($\pm$ 0.0003) |
| Oracle (infeasible) | **1.096 ($\pm$ 0.0214)** | 0.021 ($\pm$ 0.0005) | **1.33 ($\pm$ 0.025)** | 0.02 ($\pm$ 0.0005) | **1.674 ($\pm$ 0.0311)** | 0.02 ($\pm$ 0.0006) |
| Naive-Trim (invalid) | 1.249 ($\pm$ 0.0219) | 0.026 ($\pm$ 0.0005) | 1.777 ($\pm$ 0.0254) | 0.035 ($\pm$ 0.0006) | 2.452 ($\pm$ 0.0324) | 0.044 ($\pm$ 0.0007) |
| Small-Clean | 0.819 ($\pm$ 0.0592) | 0.018 ($\pm$ 0.002) | 0.613 ($\pm$ 0.0753) | 0.011 ($\pm$ 0.0018) | 0.406 ($\pm$ 0.0769) | 0.006 ($\pm$ 0.0013) |
| Label-Trim | **1.079 ($\pm$ 0.0212)** | 0.02 ($\pm$ 0.0005) | **1.23 ($\pm$ 0.0247)** | 0.017 ($\pm$ 0.0004) | **1.381 ($\pm$ 0.0298)** | 0.014 ($\pm$ 0.0005) |

(a) Target type-I error rate $\alpha = 0.02$

We also report performance averaged across the six datasets for two additional models: ReAct (Sun et al., 2021) with a VGG-19 backbone and SCALE (Xu et al., 2024) with a ResNet-18 backbone, shown in Tables 4 and 5, respectively. These results align with our main findings and further demonstrate the effectiveness of the proposed `Label-Trim` method.

Following this, we provide detailed results for each dataset, reporting type-I error rates and the power of each method, with the power reported relative to the `Standard` method (normalized to 1). The actual power of the `Standard` method is included in the last row of each table. Specifically, Tables 6 to 11 present the detailed per-dataset results corresponding to Tables 1 and 3. Additionally, detailed per-dataset results for the additional models are provided in Tables 12 to 17 and Tables 18 to 23, corresponding to Table 4 and Table 5, respectively.

Table 4: Comparison of conformal outlier detection methods on six visual datasets for varying contamination rate $r$ and target type-I error level $\alpha$. All methods utilize the ReAct (Sun et al., 2021) method with a pretrained VGG-19. Other details are as in Table 1 in the main manuscript.

| | Contamination rate | | | | | |
| | 1% | | 3% | | 5% | |
| Method | Power | Type-I Error | Power | Type-I Error | Power | Type-I Error |
|---|---|---|---|---|---|---|
| Standard | **1.0 (± 0.0317)** | 0.009 (± 0.0004) | **1.0 (± 0.0377)** | 0.006 (± 0.0003) | **1.0 (± 0.0422)** | 0.004 (± 0.0002) |
| Oracle (infeasible) | **1.095 (± 0.033)** | 0.01 (± 0.0004) | **1.368 (± 0.0418)** | 0.01 (± 0.0004) | **1.633 (± 0.0488)** | 0.009 (± 0.0003) |
| Naive-Trim (invalid) | 1.464 (± 0.0373) | 0.018 (± 0.0005) | 2.5 (± 0.0531) | 0.031 (± 0.0006) | 3.447 (± 0.0592) | 0.04 (± 0.0008) |
| Small-Clean | 0.0 (± 0.0) | 0.0 (± 0.0) | 0.0 (± 0.0) | 0.0 (± 0.0) | 0.0 (± 0.0) | 0.0 (± 0.0) |
| Label-Trim | **1.095 (± 0.033)** | 0.01 (± 0.0004) | **1.364 (± 0.0418)** | 0.01 (± 0.0004) | **1.606 (± 0.0487)** | 0.009 (± 0.0003) |

(a) Target type-I error rate $\alpha = 0.01$

| | Contamination rate | | | | | |
| | 1% | | 3% | | 5% | |
| Method | Power | Type-I Error | Power | Type-I Error | Power | Type-I Error |
|---|---|---|---|---|---|---|
| Standard | **1.0 (± 0.0251)** | 0.018 (± 0.0005) | **1.0 (± 0.027)** | 0.014 (± 0.0004) | **1.0 (± 0.0291)** | 0.01 (± 0.0004) |
| Oracle (infeasible) | **1.068 (± 0.0247)** | 0.021 (± 0.0005) | **1.221 (± 0.0297)** | 0.02 (± 0.0005) | **1.37 (± 0.0323)** | 0.019 (± 0.0006) |
| Naive-Trim (invalid) | 1.252 (± 0.0256) | 0.028 (± 0.0006) | 1.759 (± 0.0326) | 0.039 (± 0.0007) | 2.212 (± 0.0345) | 0.048 (± 0.0008) |
| Small-Clean | 0.813 (± 0.0593) | 0.017 (± 0.0019) | 0.593 (± 0.0737) | 0.011 (± 0.0018) | 0.352 (± 0.0663) | 0.005 (± 0.0013) |
| Label-Trim | **1.056 (± 0.0244)** | 0.02 (± 0.0005) | **1.177 (± 0.0292)** | 0.019 (± 0.0005) | **1.253 (± 0.0312)** | 0.016 (± 0.0005) |

(b) Target type-I error rate $\alpha = 0.02$

| | Contamination rate | | | | | |
| | 1% | | 3% | | 5% | |
| Method | Power | Type-I Error | Power | Type-I Error | Power | Type-I Error |
|---|---|---|---|---|---|---|
| Standard | **1.0 (± 0.0202)** | 0.028 (± 0.0006) | **1.0 (± 0.0238)** | 0.022 (± 0.0005) | **1.0 (± 0.0247)** | 0.017 (± 0.0005) |
| Oracle (infeasible) | **1.074 (± 0.02)** | 0.031 (± 0.0006) | **1.194 (± 0.0256)** | 0.03 (± 0.0007) | **1.318 (± 0.0271)** | 0.029 (± 0.0007) |
| Naive-Trim (invalid) | 1.186 (± 0.0202) | 0.037 (± 0.0007) | 1.51 (± 0.0248) | 0.048 (± 0.0008) | 1.848 (± 0.0272) | 0.056 (± 0.0009) |
| Small-Clean | 0.709 (± 0.045) | 0.019 (± 0.0018) | 0.85 (± 0.0541) | 0.021 (± 0.002) | 0.932 (± 0.0568) | 0.019 (± 0.002) |
| Label-Trim | **1.048 (± 0.0202)** | 0.03 (± 0.0006) | **1.118 (± 0.0248)** | 0.027 (± 0.0006) | **1.167 (± 0.026)** | 0.023 (± 0.0006) |

(c) Target type-I error rate $\alpha = 0.03$

Table 5: Comparison of conformal outlier detection methods on six visual datasets for varying contamination rate $r$ and target type-I error level $\alpha$. All methods utilize the SCALE (Xu et al., 2024) method with a pretrained ResNet-18. Other details are as in Table 1 in the main manuscript.

| | Contamination rate | | | | | |
| | 1% | | 3% | | 5% | |
| Method | Power | Type-I Error | Power | Type-I Error | Power | Type-I Error |
|---|---|---|---|---|---|---|
| Standard | **1.0** (± **0.0358**) | 0.009 (± 0.0003) | **1.0** (± **0.0375**) | 0.006 (± 0.0003) | **1.0** (± **0.0407**) | 0.004 (± 0.0002) |
| Oracle (infeasible) | **1.144** (± **0.037**) | 0.01 (± 0.0003) | **1.472** (± **0.0474**) | 0.01 (± 0.0003) | **1.708** (± **0.0522**) | 0.01 (± 0.0004) |
| Naive-Trim (invalid) | 1.736 (± 0.0414) | 0.017 (± 0.0004) | 2.93 (± 0.0553) | 0.028 (± 0.0006) | 4.144 (± 0.0614) | 0.038 (± 0.0007) |
| Small-Clean | 0.0 (± 0.0) | 0.0 (± 0.0) | 0.0 (± 0.0) | 0.0 (± 0.0) | 0.0 (± 0.0) | 0.0 (± 0.0) |
| Label-Trim | **1.144** (± **0.0371**) | 0.01 (± 0.0003) | **1.456** (± **0.0469**) | 0.01 (± 0.0003) | **1.627** (± **0.0501**) | 0.009 (± 0.0003) |

(a) Target type-I error rate $\alpha = 0.01$

| | Contamination rate | | | | | |
| | 1% | | 3% | | 5% | |
| Method | Power | Type-I Error | Power | Type-I Error | Power | Type-I Error |
|---|---|---|---|---|---|---|
| Standard | **1.0** (± **0.0236**) | 0.018 (± 0.0004) | **1.0** (± **0.0267**) | 0.013 (± 0.0004) | **1.0** (± **0.0288**) | 0.01 (± 0.0003) |
| Oracle (infeasible) | **1.076** (± **0.0244**) | 0.021 (± 0.0005) | **1.306** (± **0.0279**) | 0.02 (± 0.0005) | **1.583** (± **0.0316**) | 0.02 (± 0.0005) |
| Naive-Trim (invalid) | 1.247 (± 0.0253) | 0.027 (± 0.0006) | 1.86 (± 0.0294) | 0.036 (± 0.0006) | 2.532 (± 0.0335) | 0.047 (± 0.0008) |
| Small-Clean | 0.831 (± 0.0598) | 0.018 (± 0.0019) | 0.609 (± 0.0765) | 0.011 (± 0.0017) | 0.348 (± 0.0744) | 0.005 (± 0.0014) |
| Label-Trim | **1.062** (± **0.0243**) | 0.02 (± 0.0005) | **1.226** (± **0.0284**) | 0.018 (± 0.0005) | **1.368** (± **0.0315**) | 0.015 (± 0.0004) |

(b) Target type-I error rate $\alpha = 0.02$

| | Contamination rate | | | | | |
| | 1% | | 3% | | 5% | |
| Method | Power | Type-I Error | Power | Type-I Error | Power | Type-I Error |
|---|---|---|---|---|---|---|
| Standard | **1.0** (± **0.0201**) | 0.027 (± 0.0006) | **1.0** (± **0.0212**) | 0.021 (± 0.0005) | **1.0** (± **0.022**) | 0.016 (± 0.0004) |
| Oracle (infeasible) | **1.082** (± **0.0204**) | 0.031 (± 0.0006) | **1.233** (± **0.0227**) | 0.029 (± 0.0006) | **1.398** (± **0.0237**) | 0.03 (± 0.0006) |
| Naive-Trim (invalid) | 1.19 (± 0.0197) | 0.036 (± 0.0006) | 1.547 (± 0.0229) | 0.044 (± 0.0007) | 1.9 (± 0.0239) | 0.055 (± 0.0009) |
| Small-Clean | 0.712 (± 0.045) | 0.019 (± 0.0019) | 0.862 (± 0.055) | 0.02 (± 0.002) | 0.971 (± 0.0606) | 0.021 (± 0.0021) |
| Label-Trim | **1.05** (± **0.0204**) | 0.029 (± 0.0006) | **1.116** (± **0.022**) | 0.025 (± 0.0006) | **1.178** (± **0.0224**) | 0.022 (± 0.0005) |

(c) Target type-I error rate $\alpha = 0.03$

Table 6: Comparison of conformal outlier detection methods on Texture dataset (outliers) and CIFAR-10 dataset (inliers) for varying contamination rate $r$ and target type-I error level $\alpha$. All methods utilize the ReAct (Sun et al., 2021) method with a pretrained ResNet-18. The empirical power is presented relative to the Standard method (higher is better). Results are averaged across 100 random splits of the data, with standard errors presented in parentheses.

| | Contamination rate | | | | | |
| | 1% | | 3% | | 5% | |
| Method | Power | Type-I Error | Power | Type-I Error | Power | Type-I Error |
|---|---|---|---|---|---|---|
| Standard | **1.0 ($\pm$ 0.0295)** | 0.008 ($\pm$ 0.0003) | **1.0 ($\pm$ 0.0369)** | 0.006 ($\pm$ 0.0003) | **1.0 ($\pm$ 0.0405)** | 0.004 ($\pm$ 0.0002) |
| Oracle (infeasible) | **1.173 ($\pm$ 0.0295)** | 0.01 ($\pm$ 0.0003) | **1.455 ($\pm$ 0.0432)** | 0.01 ($\pm$ 0.0003) | **1.824 ($\pm$ 0.051)** | 0.009 ($\pm$ 0.0004) |
| Naive-Trim (invalid) | 1.722 ($\pm$ 0.035) | 0.017 ($\pm$ 0.0004) | 2.668 ($\pm$ 0.0428) | 0.028 ($\pm$ 0.0006) | 4.008 ($\pm$ 0.0599) | 0.037 ($\pm$ 0.0007) |
| Small-Clean | 0.0 ($\pm$ 0.0) | 0.0 ($\pm$ 0.0) | 0.0 ($\pm$ 0.0) | 0.0 ($\pm$ 0.0) | 0.0 ($\pm$ 0.0) | 0.0 ($\pm$ 0.0) |
| Label-Trim | **1.173 ($\pm$ 0.0295)** | 0.01 ($\pm$ 0.0003) | **1.448 ($\pm$ 0.0428)** | 0.01 ($\pm$ 0.0003) | **1.73 ($\pm$ 0.0477)** | 0.009 ($\pm$ 0.0003) |
| Standard Power | 0.194 ($\pm$ 0.0057) | | 0.159 ($\pm$ 0.0059) | | 0.123 ($\pm$ 0.005) | |

(a) Target type-I error rate $\alpha = 0.01$

| | Contamination rate | | | | | |
| | 1% | | 3% | | 5% | |
| Method | Power | Type-I Error | Power | Type-I Error | Power | Type-I Error |
|---|---|---|---|---|---|---|
| Standard | **1.0 ($\pm$ 0.0202)** | 0.017 ($\pm$ 0.0005) | **1.0 ($\pm$ 0.0239)** | 0.012 ($\pm$ 0.0004) | **1.0 ($\pm$ 0.0264)** | 0.009 ($\pm$ 0.0003) |
| Oracle (infeasible) | **1.109 ($\pm$ 0.0199)** | 0.021 ($\pm$ 0.0005) | **1.315 ($\pm$ 0.0259)** | 0.02 ($\pm$ 0.0005) | **1.623 ($\pm$ 0.0346)** | 0.02 ($\pm$ 0.0006) |
| Naive-Trim (invalid) | 1.24 ($\pm$ 0.0193) | 0.026 ($\pm$ 0.0006) | 1.781 ($\pm$ 0.026) | 0.035 ($\pm$ 0.0007) | 2.431 ($\pm$ 0.0334) | 0.045 ($\pm$ 0.0007) |
| Small-Clean | 0.819 ($\pm$ 0.0599) | 0.018 ($\pm$ 0.002) | 0.702 ($\pm$ 0.0824) | 0.014 ($\pm$ 0.0025) | 0.478 ($\pm$ 0.0842) | 0.007 ($\pm$ 0.0016) |
| Label-Trim | **1.089 ($\pm$ 0.0199)** | 0.02 ($\pm$ 0.0005) | **1.219 ($\pm$ 0.0253)** | 0.017 ($\pm$ 0.0004) | **1.353 ($\pm$ 0.0303)** | 0.014 ($\pm$ 0.0005) |
| Standard Power | 0.337 ($\pm$ 0.0068) | | 0.272 ($\pm$ 0.0065) | | 0.224 ($\pm$ 0.0059) | |

(b) Target type-I error rate $\alpha = 0.02$

| | Contamination rate | | | | | |
| | 1% | | 3% | | 5% | |
| Method | Power | Type-I Error | Power | Type-I Error | Power | Type-I Error |
|---|---|---|---|---|---|---|
| Standard | **1.0 ($\pm$ 0.0155)** | 0.027 ($\pm$ 0.0006) | **1.0 ($\pm$ 0.0197)** | 0.02 ($\pm$ 0.0005) | 1.0 ($\pm$ 0.0215) | 0.015 ($\pm$ 0.0005) |
| Oracle (infeasible) | **1.068 ($\pm$ 0.0152)** | 0.03 ($\pm$ 0.0006) | **1.224 ($\pm$ 0.0191)** | 0.03 ($\pm$ 0.0006) | **1.427 ($\pm$ 0.0249)** | 0.03 ($\pm$ 0.0007) |
| Naive-Trim (invalid) | 1.15 ($\pm$ 0.0152) | 0.036 ($\pm$ 0.0006) | 1.514 ($\pm$ 0.0178) | 0.043 ($\pm$ 0.0007) | 1.919 ($\pm$ 0.0229) | 0.052 ($\pm$ 0.0008) |
| Small-Clean | 0.743 ($\pm$ 0.0441) | 0.02 ($\pm$ 0.002) | 0.883 ($\pm$ 0.0549) | 0.021 ($\pm$ 0.0025) | **1.062 ($\pm$ 0.0665)** | 0.024 ($\pm$ 0.0027) |
| Label-Trim | **1.046 ($\pm$ 0.0153)** | 0.029 ($\pm$ 0.0006) | **1.13 ($\pm$ 0.019)** | 0.025 ($\pm$ 0.0005) | **1.215 ($\pm$ 0.0245)** | 0.021 ($\pm$ 0.0005) |
| Standard Power | 0.421 ($\pm$ 0.0065) | | 0.357 ($\pm$ 0.007) | | 0.309 ($\pm$ 0.0067) | |

(c) Target type-I error rate $\alpha = 0.03$

Table 7: Comparison of conformal outlier detection methods on SVHN dataset (outliers) and CIFAR-10 dataset (inliers) for varying contamination rate $r$ and target type-I error level $\alpha$. All methods utilize the ReAct (Sun et al., 2021) method with a pretrained ResNet-18. The empirical power is presented relative to the `Standard` method (higher is better). Results are averaged across 100 random splits of the data, with standard errors presented in parentheses.

| | Contamination rate | | | | | |
| | 1% | | 3% | | 5% | |
| Method | Power | Type-I Error | Power | Type-I Error | Power | Type-I Error |
|---|---|---|---|---|---|---|
| Standard | **1.0 (± 0.0255)** | 0.008 (± 0.0003) | **1.0 (± 0.0316)** | 0.004 (± 0.0002) | **1.0 (± 0.0388)** | 0.002 (± 0.0002) |
| Oracle (infeasible) | **1.192 (± 0.0262)** | 0.01 (± 0.0003) | **1.654 (± 0.0368)** | 0.01 (± 0.0003) | **2.208 (± 0.052)** | 0.009 (± 0.0004) |
| Naive-Trim (invalid) | 1.573 (± 0.0271) | 0.016 (± 0.0004) | 2.689 (± 0.0353) | 0.025 (± 0.0006) | 4.073 (± 0.0533) | 0.033 (± 0.0007) |
| Small-Clean | 0.0 (± 0.0) | 0.0 (± 0.0) | 0.0 (± 0.0) | 0.0 (± 0.0) | 0.0 (± 0.0) | 0.0 (± 0.0) |
| Label-Trim | **1.192 (± 0.0262)** | 0.01 (± 0.0003) | **1.601 (± 0.0367)** | 0.009 (± 0.0003) | **1.895 (± 0.0448)** | 0.007 (± 0.0003) |
| Standard Power | 0.271 (± 0.0069) | | 0.191 (± 0.006) | | 0.137 (± 0.0053) | |

(a) Target type-I error rate $\alpha = 0.01$

| | Contamination rate | | | | | |
| | 1% | | 3% | | 5% | |
| Method | Power | Type-I Error | Power | Type-I Error | Power | Type-I Error |
|---|---|---|---|---|---|---|
| Standard | **1.0 (± 0.0168)** | 0.016 (± 0.0004) | **1.0 (± 0.0218)** | 0.01 (± 0.0003) | **1.0 (± 0.0245)** | 0.007 (± 0.0003) |
| Oracle (infeasible) | **1.096 (± 0.0173)** | 0.021 (± 0.0005) | **1.404 (± 0.0211)** | 0.02 (± 0.0005) | **1.828 (± 0.0304)** | 0.02 (± 0.0006) |
| Naive-Trim (invalid) | 1.2 (± 0.0175) | 0.026 (± 0.0005) | 1.721 (± 0.0216) | 0.032 (± 0.0006) | 2.42 (± 0.0281) | 0.041 (± 0.0007) |
| Small-Clean | 0.833 (± 0.053) | 0.02 (± 0.0026) | 0.598 (± 0.0754) | 0.01 (± 0.0016) | 0.509 (± 0.0929) | 0.008 (± 0.0019) |
| Label-Trim | **1.08 (± 0.0176)** | 0.02 (± 0.0005) | **1.277 (± 0.0228)** | 0.016 (± 0.0004) | **1.469 (± 0.0288)** | 0.013 (± 0.0004) |
| Standard Power | 0.429 (± 0.0072) | | 0.332 (± 0.0072) | | 0.25 (± 0.0061) | |

(b) Target type-I error rate $\alpha = 0.02$

| | Contamination rate | | | | | |
| | 1% | | 3% | | 5% | |
| Method | Power | Type-I Error | Power | Type-I Error | Power | Type-I Error |
|---|---|---|---|---|---|---|
| Standard | **1.0 (± 0.0145)** | 0.026 (± 0.0005) | **1.0 (± 0.0169)** | 0.017 (± 0.0004) | 1.0 (± 0.0204) | 0.012 (± 0.0004) |
| Oracle (infeasible) | **1.063 (± 0.014)** | 0.03 (± 0.0006) | **1.246 (± 0.0158)** | 0.029 (± 0.0006) | **1.513 (± 0.0212)** | 0.03 (± 0.0007) |
| Naive-Trim (invalid) | 1.115 (± 0.0137) | 0.035 (± 0.0006) | 1.395 (± 0.0148) | 0.041 (± 0.0007) | 1.825 (± 0.0199) | 0.049 (± 0.0008) |
| Small-Clean | 0.73 (± 0.0419) | 0.021 (± 0.0027) | 0.948 (± 0.0447) | 0.022 (± 0.0019) | **1.092 (± 0.0625)** | 0.022 (± 0.0023) |
| Label-Trim | **1.042 (± 0.0143)** | 0.029 (± 0.0006) | **1.144 (± 0.0151)** | 0.024 (± 0.0005) | **1.266 (± 0.0209)** | 0.019 (± 0.0006) |
| Standard Power | 0.517 (± 0.0075) | | 0.44 (± 0.0074) | | 0.355 (± 0.0072) | |

(c) Target type-I error rate $\alpha = 0.03$

Table 8: Comparison of conformal outlier detection methods on Places365 dataset (outliers) and CIFAR-10 dataset (inliers) for varying contamination rate $r$ and target type-I error level $\alpha$. All methods utilize the ReAct (Sun et al., 2021) method with a pretrained ResNet-18. The empirical power is presented relative to the `Standard` method (higher is better). Results are averaged across 100 random splits of the data, with standard errors presented in parentheses.

| | Contamination rate | | | | | |
| | 1% | | 3% | | 5% | |
| Method | Power | Type-I Error | Power | Type-I Error | Power | Type-I Error |
| --- | --- | --- | --- | --- | --- | --- |
| Standard | **1.0 ($\pm$ 0.033)** | 0.009 ($\pm$ 0.0003) | **1.0 ($\pm$ 0.0364)** | 0.006 ($\pm$ 0.0003) | **1.0 ($\pm$ 0.0431)** | 0.004 ($\pm$ 0.0002) |
| Oracle (infeasible) | **1.139 ($\pm$ 0.0339)** | 0.01 ($\pm$ 0.0003) | **1.47 ($\pm$ 0.0423)** | 0.01 ($\pm$ 0.0003) | **1.839 ($\pm$ 0.0558)** | 0.009 ($\pm$ 0.0004) |
| Naive-Trim (invalid) | 1.624 ($\pm$ 0.0339) | 0.017 ($\pm$ 0.0004) | 2.73 ($\pm$ 0.0487) | 0.028 ($\pm$ 0.0006) | 4.12 ($\pm$ 0.069) | 0.039 ($\pm$ 0.0007) |
| Small-Clean | 0.0 ($\pm$ 0.0) | 0.0 ($\pm$ 0.0) | 0.0 ($\pm$ 0.0) | 0.0 ($\pm$ 0.0) | 0.0 ($\pm$ 0.0) | 0.0 ($\pm$ 0.0) |
| Label-Trim | **1.139 ($\pm$ 0.0339)** | 0.01 ($\pm$ 0.0003) | **1.466 ($\pm$ 0.0421)** | 0.01 ($\pm$ 0.0003) | **1.707 ($\pm$ 0.0531)** | 0.009 ($\pm$ 0.0003) |
| Standard Power | 0.193 ($\pm$ 0.0064) | | 0.148 ($\pm$ 0.0054) | | 0.115 ($\pm$ 0.005) | |

(a) Target type-I error rate $\alpha = 0.01$

| | Contamination rate | | | | | |
| | 1% | | 3% | | 5% | |
| Method | Power | Type-I Error | Power | Type-I Error | Power | Type-I Error |
| --- | --- | --- | --- | --- | --- | --- |
| Standard | **1.0 ($\pm$ 0.0209)** | 0.017 ($\pm$ 0.0005) | **1.0 ($\pm$ 0.0241)** | 0.013 ($\pm$ 0.0004) | **1.0 ($\pm$ 0.0298)** | 0.009 ($\pm$ 0.0003) |
| Oracle (infeasible) | **1.091 ($\pm$ 0.0224)** | 0.021 ($\pm$ 0.0005) | **1.271 ($\pm$ 0.0258)** | 0.02 ($\pm$ 0.0005) | **1.599 ($\pm$ 0.0308)** | 0.02 ($\pm$ 0.0006) |
| Naive-Trim (invalid) | 1.246 ($\pm$ 0.0225) | 0.027 ($\pm$ 0.0006) | 1.753 ($\pm$ 0.0277) | 0.036 ($\pm$ 0.0007) | 2.462 ($\pm$ 0.0376) | 0.046 ($\pm$ 0.0007) |
| Small-Clean | 0.799 ($\pm$ 0.0632) | 0.018 ($\pm$ 0.0021) | 0.549 ($\pm$ 0.0716) | 0.01 ($\pm$ 0.0016) | 0.255 ($\pm$ 0.0595) | 0.003 ($\pm$ 0.0009) |
| Label-Trim | **1.071 ($\pm$ 0.0219)** | 0.02 ($\pm$ 0.0005) | **1.187 ($\pm$ 0.0247)** | 0.017 ($\pm$ 0.0004) | **1.361 ($\pm$ 0.0306)** | 0.015 ($\pm$ 0.0005) |
| Standard Power | 0.316 ($\pm$ 0.0066) | | 0.261 ($\pm$ 0.0063) | | 0.213 ($\pm$ 0.0064) | |

(b) Target type-I error rate $\alpha = 0.02$

| | Contamination rate | | | | | |
| | 1% | | 3% | | 5% | |
| Method | Power | Type-I Error | Power | Type-I Error | Power | Type-I Error |
| --- | --- | --- | --- | --- | --- | --- |
| Standard | **1.0 ($\pm$ 0.018)** | 0.027 ($\pm$ 0.0006) | **1.0 ($\pm$ 0.0204)** | 0.02 ($\pm$ 0.0005) | 1.0 ($\pm$ 0.021) | 0.015 ($\pm$ 0.0005) |
| Oracle (infeasible) | **1.055 ($\pm$ 0.0189)** | 0.03 ($\pm$ 0.0006) | **1.231 ($\pm$ 0.0213)** | 0.029 ($\pm$ 0.0006) | **1.403 ($\pm$ 0.025)** | 0.03 ($\pm$ 0.0007) |
| Naive-Trim (invalid) | 1.145 ($\pm$ 0.0199) | 0.036 ($\pm$ 0.0006) | 1.496 ($\pm$ 0.0217) | 0.044 ($\pm$ 0.0007) | 1.883 ($\pm$ 0.0261) | 0.054 ($\pm$ 0.0008) |
| Small-Clean | 0.746 ($\pm$ 0.0469) | 0.021 ($\pm$ 0.002) | 0.833 ($\pm$ 0.0504) | 0.018 ($\pm$ 0.0018) | **1.029 ($\pm$ 0.0593)** | 0.022 ($\pm$ 0.0022) |
| Label-Trim | **1.038 ($\pm$ 0.0188)** | 0.029 ($\pm$ 0.0006) | **1.149 ($\pm$ 0.0211)** | 0.025 ($\pm$ 0.0006) | **1.186 ($\pm$ 0.0228)** | 0.022 ($\pm$ 0.0006) |
| Standard Power | 0.395 ($\pm$ 0.0071) | | 0.335 ($\pm$ 0.0068) | | 0.302 ($\pm$ 0.0063) | |

(c) Target type-I error rate $\alpha = 0.03$

Table 9: Comparison of conformal outlier detection methods on MNIST dataset (outliers) and CIFAR-10 dataset (inliers) for varying contamination rate $r$ and target type-I error level $\alpha$. All methods utilize the ReAct (Sun et al., 2021) method with a pretrained ResNet-18. The empirical power is presented relative to the `Standard` method (higher is better). Results are averaged across 100 random splits of the data, with standard errors presented in parentheses.

| | Contamination rate | | | | | |
| --- | --- | --- | --- | --- | --- | --- |
| | 1% | | 3% | | 5% | |
| Method | Power | Type-I Error | Power | Type-I Error | Power | Type-I Error |
| Standard | **1.0** ($\pm$ **0.0248**) | 0.008 ($\pm$ 0.0003) | **1.0** ($\pm$ **0.0283**) | 0.004 ($\pm$ 0.0002) | **1.0** ($\pm$ **0.0363**) | 0.003 ($\pm$ 0.0002) |
| Oracle (infeasible) | **1.241** ($\pm$ **0.0291**) | 0.01 ($\pm$ 0.0003) | **1.758** ($\pm$ **0.0352**) | 0.01 ($\pm$ 0.0003) | **2.554** ($\pm$ **0.0601**) | 0.009 ($\pm$ 0.0004) |
| Naive-Trim (invalid) | 1.629 ($\pm$ 0.0292) | 0.016 ($\pm$ 0.0004) | 2.722 ($\pm$ 0.0363) | 0.023 ($\pm$ 0.0005) | 4.718 ($\pm$ 0.0552) | 0.03 ($\pm$ 0.0006) |
| Small-Clean | 0.0 ($\pm$ 0.0) | 0.0 ($\pm$ 0.0) | 0.0 ($\pm$ 0.0) | 0.0 ($\pm$ 0.0) | 0.0 ($\pm$ 0.0) | 0.0 ($\pm$ 0.0) |
| Label-Trim | **1.241** ($\pm$ **0.0291**) | 0.01 ($\pm$ 0.0003) | **1.636** ($\pm$ **0.0331**) | 0.009 ($\pm$ 0.0003) | **2.113** ($\pm$ **0.0555**) | 0.007 ($\pm$ 0.0003) |
| Standard Power | 0.282 ($\pm$ 0.007) | | 0.208 ($\pm$ 0.0059) | | 0.131 ($\pm$ 0.0048) | |

(a) Target type-I error rate $\alpha = 0.01$

| | Contamination rate | | | | | |
| --- | --- | --- | --- | --- | --- | --- |
| | 1% | | 3% | | 5% | |
| Method | Power | Type-I Error | Power | Type-I Error | Power | Type-I Error |
| Standard | **1.0** ($\pm$ **0.0173**) | 0.016 ($\pm$ 0.0004) | **1.0** ($\pm$ **0.0205**) | 0.01 ($\pm$ 0.0003) | **1.0** ($\pm$ **0.027**) | 0.007 ($\pm$ 0.0003) |
| Oracle (infeasible) | **1.117** ($\pm$ **0.0179**) | 0.021 ($\pm$ 0.0005) | **1.452** ($\pm$ **0.0209**) | 0.02 ($\pm$ 0.0005) | **1.971** ($\pm$ **0.0297**) | 0.02 ($\pm$ 0.0006) |
| Naive-Trim (invalid) | 1.231 ($\pm$ 0.0179) | 0.025 ($\pm$ 0.0005) | 1.759 ($\pm$ 0.0207) | 0.031 ($\pm$ 0.0006) | 2.531 ($\pm$ 0.0273) | 0.038 ($\pm$ 0.0007) |
| Small-Clean | 0.844 ($\pm$ 0.0514) | 0.019 ($\pm$ 0.0019) | 0.617 ($\pm$ 0.0738) | 0.01 ($\pm$ 0.0017) | 0.438 ($\pm$ 0.0817) | 0.005 ($\pm$ 0.0011) |
| Label-Trim | **1.096** ($\pm$ **0.0181**) | 0.02 ($\pm$ 0.0005) | **1.308** ($\pm$ **0.0206**) | 0.015 ($\pm$ 0.0004) | **1.493** ($\pm$ **0.0295**) | 0.012 ($\pm$ 0.0004) |
| Standard Power | 0.465 ($\pm$ 0.008) | | 0.36 ($\pm$ 0.0074) | | 0.264 ($\pm$ 0.0071) | |

(b) Target type-I error rate $\alpha = 0.02$

| | Contamination rate | | | | | |
| --- | --- | --- | --- | --- | --- | --- |
| | 1% | | 3% | | 5% | |
| Method | Power | Type-I Error | Power | Type-I Error | Power | Type-I Error |
| Standard | **1.0** ($\pm$ **0.0147**) | 0.025 ($\pm$ 0.0005) | **1.0** ($\pm$ **0.0151**) | 0.016 ($\pm$ 0.0004) | 1.0 ($\pm$ 0.0201) | 0.011 ($\pm$ 0.0004) |
| Oracle (infeasible) | **1.072** ($\pm$ **0.0149**) | 0.03 ($\pm$ 0.0006) | **1.295** ($\pm$ **0.0155**) | 0.029 ($\pm$ 0.0006) | **1.617** ($\pm$ **0.0186**) | 0.03 ($\pm$ 0.0007) |
| Naive-Trim (invalid) | 1.116 ($\pm$ 0.0144) | 0.034 ($\pm$ 0.0006) | 1.424 ($\pm$ 0.0142) | 0.039 ($\pm$ 0.0007) | 1.851 ($\pm$ 0.0178) | 0.046 ($\pm$ 0.0007) |
| Small-Clean | 0.699 ($\pm$ 0.0394) | 0.019 ($\pm$ 0.0019) | 0.884 ($\pm$ 0.0464) | 0.019 ($\pm$ 0.0018) | **1.164** ($\pm$ **0.0576**) | 0.02 ($\pm$ 0.002) |
| Label-Trim | **1.049** ($\pm$ **0.0148**) | 0.029 ($\pm$ 0.0006) | **1.158** ($\pm$ **0.016**) | 0.023 ($\pm$ 0.0005) | **1.273** ($\pm$ **0.0218**) | 0.017 ($\pm$ 0.0005) |
| Standard Power | 0.576 ($\pm$ 0.0085) | | 0.483 ($\pm$ 0.0073) | | 0.382 ($\pm$ 0.0077) | |

(c) Target type-I error rate $\alpha = 0.03$

Table 10: Comparison of conformal outlier detection methods on CIFAR-100 dataset (outliers) and CIFAR-10 dataset (inliers) for varying contamination rate $r$ and target type-I error level $\alpha$. All methods utilize the ReAct (Sun et al., 2021) method with a pretrained ResNet-18. The empirical power is presented relative to the `Standard` method (higher is better). Results are averaged across 100 random splits of the data, with standard errors presented in parentheses.

| | Contamination rate | | | | | |
| | 1% | | 3% | | 5% | |
| Method | Power | Type-I Error | Power | Type-I Error | Power | Type-I Error |
|---|---|---|---|---|---|---|
| Standard | **1.0 (± 0.0393)** | 0.009 (± 0.0003) | **1.0 (± 0.0391)** | 0.007 (± 0.0003) | **1.0 (± 0.042)** | 0.005 (± 0.0003) |
| Oracle (infeasible) | **1.116 (± 0.0417)** | 0.01 (± 0.0003) | **1.463 (± 0.05)** | 0.01 (± 0.0003) | **1.588 (± 0.0477)** | 0.009 (± 0.0004) |
| Naive-Trim (invalid) | 1.733 (± 0.0397) | 0.018 (± 0.0005) | 3.06 (± 0.0585) | 0.03 (± 0.0006) | 4.054 (± 0.0631) | 0.041 (± 0.0007) |
| Small-Clean | 0.0 (± 0.0) | 0.0 (± 0.0) | 0.0 (± 0.0) | 0.0 (± 0.0) | 0.0 (± 0.0) | 0.0 (± 0.0) |
| Label-Trim | **1.116 (± 0.0417)** | 0.01 (± 0.0003) | **1.463 (± 0.05)** | 0.01 (± 0.0003) | **1.587 (± 0.0468)** | 0.009 (± 0.0004) |
| Standard Power | 0.145 (± 0.0057) | | 0.118 (± 0.0046) | | 0.104 (± 0.0044) | |

(a) Target type-I error rate $\alpha = 0.01$

| | Contamination rate | | | | | |
| | 1% | | 3% | | 5% | |
| Method | Power | Type-I Error | Power | Type-I Error | Power | Type-I Error |
|---|---|---|---|---|---|---|
| Standard | **1.0 (± 0.023)** | 0.018 (± 0.0005) | **1.0 (± 0.0265)** | 0.014 (± 0.0004) | **1.0 (± 0.0264)** | 0.011 (± 0.0004) |
| Oracle (infeasible) | **1.082 (± 0.0256)** | 0.021 (± 0.0005) | **1.265 (± 0.0299)** | 0.02 (± 0.0005) | **1.476 (± 0.0317)** | 0.02 (± 0.0006) |
| Naive-Trim (invalid) | 1.294 (± 0.0266) | 0.027 (± 0.0006) | 1.831 (± 0.0288) | 0.038 (± 0.0006) | 2.464 (± 0.0372) | 0.049 (± 0.0008) |
| Small-Clean | 0.844 (± 0.0709) | 0.019 (± 0.0022) | 0.686 (± 0.0797) | 0.012 (± 0.0017) | 0.392 (± 0.078) | 0.006 (± 0.0014) |
| Label-Trim | **1.064 (± 0.0247)** | 0.02 (± 0.0005) | **1.193 (± 0.0293)** | 0.018 (± 0.0004) | **1.288 (± 0.0296)** | 0.016 (± 0.0005) |
| Standard Power | 0.255 (± 0.0059) | | 0.225 (± 0.006) | | 0.19 (± 0.005) | |

(b) Target type-I error rate $\alpha = 0.02$

| | Contamination rate | | | | | |
| | 1% | | 3% | | 5% | |
| Method | Power | Type-I Error | Power | Type-I Error | Power | Type-I Error |
|---|---|---|---|---|---|---|
| Standard | **1.0 (± 0.0204)** | 0.027 (± 0.0006) | **1.0 (± 0.0214)** | 0.022 (± 0.0005) | **1.0 (± 0.0228)** | 0.018 (± 0.0005) |
| Oracle (infeasible) | **1.054 (± 0.0212)** | 0.031 (± 0.0006) | **1.184 (± 0.0228)** | 0.03 (± 0.0006) | **1.33 (± 0.0246)** | 0.03 (± 0.0007) |
| Naive-Trim (invalid) | 1.184 (± 0.0213) | 0.036 (± 0.0006) | 1.523 (± 0.0222) | 0.046 (± 0.0007) | 1.94 (± 0.026) | 0.056 (± 0.0009) |
| Small-Clean | 0.709 (± 0.0527) | 0.021 (± 0.0023) | 0.842 (± 0.0545) | 0.021 (± 0.0019) | 0.947 (± 0.0628) | 0.02 (± 0.002) |
| Label-Trim | **1.029 (± 0.0212)** | 0.029 (± 0.0006) | **1.114 (± 0.0222)** | 0.026 (± 0.0006) | **1.16 (± 0.0229)** | 0.023 (± 0.0006) |
| Standard Power | 0.332 (± 0.0068) | | 0.302 (± 0.0064) | | 0.263 (± 0.006) | |

(c) Target type-I error rate $\alpha = 0.03$

Table 11: Comparison of conformal outlier detection methods on TinyImageNet dataset (outliers) and CIFAR-10 dataset (inliers) for varying contamination rate $r$ and target type-I error level $\alpha$. All methods utilize the ReAct (Sun et al., 2021) method with a pretrained ResNet-18. The empirical power is presented relative to the `Standard` method (higher is better). Results are averaged across 100 random splits of the data, with standard errors presented in parentheses.

| | Contamination rate | | | | | |
| --- | --- | --- | --- | --- | --- | --- |
| | 1% | | 3% | | 5% | |
| Method | Power | Type-I Error | Power | Type-I Error | Power | Type-I Error |
| Standard | **1.0** ($\pm$ **0.0381**) | 0.009 ($\pm$ 0.0003) | **1.0** ($\pm$ **0.0398**) | 0.006 ($\pm$ 0.0003) | **1.0** ($\pm$ **0.0439**) | 0.004 ($\pm$ 0.0002) |
| Oracle (infeasible) | **1.137** ($\pm$ **0.0409**) | 0.01 ($\pm$ 0.0003) | **1.492** ($\pm$ **0.0474**) | 0.01 ($\pm$ 0.0003) | **1.754** ($\pm$ **0.0518**) | 0.009 ($\pm$ 0.0004) |
| Naive-Trim (invalid) | 1.675 ($\pm$ 0.0401) | 0.018 ($\pm$ 0.0004) | 2.872 ($\pm$ 0.0485) | 0.028 ($\pm$ 0.0006) | 3.986 ($\pm$ 0.0568) | 0.039 ($\pm$ 0.0007) |
| Small-Clean | 0.0 ($\pm$ 0.0) | 0.0 ($\pm$ 0.0) | 0.0 ($\pm$ 0.0) | 0.0 ($\pm$ 0.0) | 0.0 ($\pm$ 0.0) | 0.0 ($\pm$ 0.0) |
| Label-Trim | **1.137** ($\pm$ **0.0409**) | 0.01 ($\pm$ 0.0003) | **1.488** ($\pm$ **0.0472**) | 0.01 ($\pm$ 0.0003) | **1.681** ($\pm$ **0.0511**) | 0.009 ($\pm$ 0.0003) |
| Standard Power | 0.168 ($\pm$ 0.0064) | | 0.133 ($\pm$ 0.0053) | | 0.115 ($\pm$ 0.005) | |

(a) Target type-I error rate $\alpha = 0.01$

| | Contamination rate | | | | | |
| --- | --- | --- | --- | --- | --- | --- |
| | 1% | | 3% | | 5% | |
| Method | Power | Type-I Error | Power | Type-I Error | Power | Type-I Error |
| Standard | **1.0** ($\pm$ **0.0239**) | 0.018 ($\pm$ 0.0004) | **1.0** ($\pm$ **0.0262**) | 0.013 ($\pm$ 0.0004) | **1.0** ($\pm$ **0.0293**) | 0.01 ($\pm$ 0.0004) |
| Oracle (infeasible) | **1.084** ($\pm$ **0.0253**) | 0.021 ($\pm$ 0.0005) | **1.275** ($\pm$ **0.0263**) | 0.02 ($\pm$ 0.0005) | **1.544** ($\pm$ **0.0293**) | 0.02 ($\pm$ 0.0006) |
| Naive-Trim (invalid) | 1.284 ($\pm$ 0.0274) | 0.027 ($\pm$ 0.0006) | 1.819 ($\pm$ 0.0277) | 0.036 ($\pm$ 0.0006) | 2.406 ($\pm$ 0.0308) | 0.046 ($\pm$ 0.0008) |
| Small-Clean | 0.777 ($\pm$ 0.0566) | 0.015 ($\pm$ 0.0015) | 0.525 ($\pm$ 0.0687) | 0.009 ($\pm$ 0.0015) | 0.366 ($\pm$ 0.0653) | 0.005 ($\pm$ 0.0011) |
| Label-Trim | **1.074** ($\pm$ **0.0249**) | 0.02 ($\pm$ 0.0005) | **1.193** ($\pm$ **0.0254**) | 0.017 ($\pm$ 0.0004) | **1.322** ($\pm$ **0.03**) | 0.015 ($\pm$ 0.0005) |
| Standard Power | 0.285 ($\pm$ 0.0068) | | 0.243 ($\pm$ 0.0064) | | 0.209 ($\pm$ 0.0061) | |

(b) Target type-I error rate $\alpha = 0.02$

| | Contamination rate | | | | | |
| --- | --- | --- | --- | --- | --- | --- |
| | 1% | | 3% | | 5% | |
| Method | Power | Type-I Error | Power | Type-I Error | Power | Type-I Error |
| Standard | **1.0** ($\pm$ **0.0213**) | 0.027 ($\pm$ 0.0006) | **1.0** ($\pm$ **0.0202**) | 0.02 ($\pm$ 0.0005) | **1.0** ($\pm$ **0.0216**) | 0.016 ($\pm$ 0.0005) |
| Oracle (infeasible) | **1.062** ($\pm$ **0.0216**) | 0.03 ($\pm$ 0.0006) | **1.23** ($\pm$ **0.0205**) | 0.03 ($\pm$ 0.0006) | **1.397** ($\pm$ **0.0236**) | 0.03 ($\pm$ 0.0007) |
| Naive-Trim (invalid) | 1.168 ($\pm$ 0.0203) | 0.036 ($\pm$ 0.0006) | 1.569 ($\pm$ 0.0207) | 0.045 ($\pm$ 0.0007) | 1.877 ($\pm$ 0.0217) | 0.054 ($\pm$ 0.0009) |
| Small-Clean | 0.656 ($\pm$ 0.0438) | 0.017 ($\pm$ 0.0017) | 0.825 ($\pm$ 0.0496) | 0.019 ($\pm$ 0.0019) | 0.904 ($\pm$ 0.0589) | 0.02 ($\pm$ 0.0024) |
| Label-Trim | **1.04** ($\pm$ **0.0219**) | 0.029 ($\pm$ 0.0006) | **1.139** ($\pm$ **0.0208**) | 0.026 ($\pm$ 0.0006) | **1.19** ($\pm$ **0.0225**) | 0.022 ($\pm$ 0.0006) |
| Standard Power | 0.368 ($\pm$ 0.0078) | | 0.318 ($\pm$ 0.0064) | | 0.29 ($\pm$ 0.0063) | |

(c) Target type-I error rate $\alpha = 0.03$

Table 12: Comparison of conformal outlier detection methods on Texture dataset (outliers) and CIFAR-10 dataset (inliers) for varying contamination rate $r$ and target type-I error level $\alpha$. All methods utilize the ReAct (Sun et al., 2021) method with a pretrained VGG-19. The empirical power is presented relative to the Standard method (higher is better). Results are averaged across 100 random splits of the data, with standard errors presented in parentheses.

| | Contamination rate | | | | | |
| | 1% | | 3% | | 5% | |
| Method | Power | Type-I Error | Power | Type-I Error | Power | Type-I Error |
|---|---|---|---|---|---|---|
| Standard | **1.0 (± 0.026)** | 0.008 (± 0.0003) | **1.0 (± 0.0312)** | 0.005 (± 0.0002) | **1.0 (± 0.0342)** | 0.003 (± 0.0002) |
| Oracle (infeasible) | **1.119 (± 0.0263)** | 0.01 (± 0.0004) | **1.421 (± 0.032)** | 0.01 (± 0.0004) | **1.833 (± 0.0437)** | 0.009 (± 0.0003) |
| Naive-Trim (invalid) | 1.368 (± 0.0278) | 0.017 (± 0.0005) | 2.168 (± 0.0379) | 0.029 (± 0.0006) | 3.383 (± 0.0481) | 0.037 (± 0.0007) |
| Small-Clean | 0.0 (± 0.0) | 0.0 (± 0.0) | 0.0 (± 0.0) | 0.0 (± 0.0) | 0.0 (± 0.0) | 0.0 (± 0.0) |
| Label-Trim | **1.119 (± 0.0263)** | 0.01 (± 0.0004) | **1.401 (± 0.0321)** | 0.01 (± 0.0003) | **1.712 (± 0.0429)** | 0.008 (± 0.0003) |
| Standard Power | 0.239 (± 0.0062) | 0.008 (± 0.0003) | 0.188 (± 0.0059) | 0.005 (± 0.0002) | 0.139 (± 0.0048) | 0.003 (± 0.0002) |

(a) Target type-I error rate $\alpha = 0.01$

| | Contamination rate | | | | | |
| | 1% | | 3% | | 5% | |
| Method | Power | Type-I Error | Power | Type-I Error | Power | Type-I Error |
|---|---|---|---|---|---|---|
| Standard | **1.0 (± 0.0202)** | 0.017 (± 0.0005) | **1.0 (± 0.0223)** | 0.012 (± 0.0004) | **1.0 (± 0.0249)** | 0.008 (± 0.0003) |
| Oracle (infeasible) | **1.083 (± 0.0186)** | 0.021 (± 0.0005) | **1.253 (± 0.0246)** | 0.02 (± 0.0005) | **1.502 (± 0.0292)** | 0.019 (± 0.0006) |
| Naive-Trim (invalid) | 1.223 (± 0.0201) | 0.027 (± 0.0006) | 1.605 (± 0.0241) | 0.037 (± 0.0007) | 2.15 (± 0.0295) | 0.045 (± 0.0008) |
| Small-Clean | 0.841 (± 0.052) | 0.018 (± 0.0018) | 0.752 (± 0.0805) | 0.016 (± 0.0025) | 0.437 (± 0.071) | 0.006 (± 0.0012) |
| Label-Trim | **1.072 (± 0.0185)** | 0.02 (± 0.0005) | **1.2 (± 0.0234)** | 0.018 (± 0.0005) | **1.323 (± 0.0276)** | 0.015 (± 0.0005) |
| Standard Power | 0.331 (± 0.0067) | 0.017 (± 0.0005) | 0.283 (± 0.0063) | 0.012 (± 0.0004) | 0.237 (± 0.0059) | 0.008 (± 0.0003) |

(b) Target type-I error rate $\alpha = 0.02$

| | Contamination rate | | | | | |
| | 1% | | 3% | | 5% | |
| Method | Power | Type-I Error | Power | Type-I Error | Power | Type-I Error |
|---|---|---|---|---|---|---|
| Standard | **1.0 (± 0.0164)** | 0.027 (± 0.0006) | **1.0 (± 0.02)** | 0.02 (± 0.0005) | 1.0 (± 0.0202) | 0.015 (± 0.0005) |
| Oracle (infeasible) | **1.075 (± 0.0153)** | 0.031 (± 0.0006) | **1.179 (± 0.0201)** | 0.03 (± 0.0007) | **1.356 (± 0.022)** | 0.029 (± 0.0007) |
| Naive-Trim (invalid) | 1.146 (± 0.0155) | 0.036 (± 0.0007) | 1.408 (± 0.0187) | 0.045 (± 0.0008) | 1.75 (± 0.0228) | 0.053 (± 0.0009) |
| Small-Clean | 0.801 (± 0.0382) | 0.022 (± 0.0018) | 0.937 (± 0.0569) | 0.025 (± 0.0028) | **1.016 (± 0.0548)** | 0.02 (± 0.0021) |
| Label-Trim | **1.053 (± 0.016)** | 0.03 (± 0.0006) | **1.117 (± 0.02)** | 0.027 (± 0.0006) | **1.208 (± 0.0221)** | 0.022 (± 0.0006) |
| Standard Power | 0.408 (± 0.0067) | 0.027 (± 0.0006) | 0.353 (± 0.0071) | 0.02 (± 0.0005) | 0.312 (± 0.0063) | 0.015 (± 0.0005) |

(c) Target type-I error rate $\alpha = 0.03$

Table 13: Comparison of conformal outlier detection methods on SVHN dataset (outliers) and CIFAR-10 dataset (inliers) for varying contamination rate $r$ and target type-I error level $\alpha$. All methods utilize the ReAct (Sun et al., 2021) method with a pretrained VGG-19. The empirical power is presented relative to the `Standard` method (higher is better). Results are averaged across 100 random splits of the data, with standard errors presented in parentheses.

| | Contamination rate | | | | | |
| --- | --- | --- | --- | --- | --- | --- |
| | 1% | | 3% | | 5% | |
| Method | Power | Type-I Error | Power | Type-I Error | Power | Type-I Error |
| Standard | **1.0** ($\pm$ **0.0297**) | 0.009 ($\pm$ 0.0003) | **1.0** ($\pm$ **0.0355**) | 0.006 ($\pm$ 0.0003) | **1.0** ($\pm$ **0.0463**) | 0.004 ($\pm$ 0.0002) |
| Oracle (infeasible) | **1.085** ($\pm$ **0.0305**) | 0.01 ($\pm$ 0.0004) | **1.287** ($\pm$ **0.0384**) | 0.01 ($\pm$ 0.0004) | **1.607** ($\pm$ **0.0452**) | 0.009 ($\pm$ 0.0003) |
| Naive-Trim (invalid) | 1.355 ($\pm$ 0.0356) | 0.018 ($\pm$ 0.0005) | 2.261 ($\pm$ 0.0511) | 0.031 ($\pm$ 0.0006) | 3.202 ($\pm$ 0.053) | 0.04 ($\pm$ 0.0007) |
| Small-Clean | 0.0 ($\pm$ 0.0) | 0.0 ($\pm$ 0.0) | 0.0 ($\pm$ 0.0) | 0.0 ($\pm$ 0.0) | 0.0 ($\pm$ 0.0) | 0.0 ($\pm$ 0.0) |
| Label-Trim | **1.085** ($\pm$ **0.0305**) | 0.01 ($\pm$ 0.0004) | **1.287** ($\pm$ **0.0384**) | 0.01 ($\pm$ 0.0004) | **1.573** ($\pm$ **0.0467**) | 0.009 ($\pm$ 0.0003) |
| Standard Power | 0.182 ($\pm$ 0.0054) | 0.009 ($\pm$ 0.0003) | 0.145 ($\pm$ 0.0052) | 0.006 ($\pm$ 0.0003) | 0.12 ($\pm$ 0.0056) | 0.004 ($\pm$ 0.0002) |

(a) Target type-I error rate $\alpha = 0.01$

| | Contamination rate | | | | | |
| --- | --- | --- | --- | --- | --- | --- |
| | 1% | | 3% | | 5% | |
| Method | Power | Type-I Error | Power | Type-I Error | Power | Type-I Error |
| Standard | **1.0** ($\pm$ **0.026**) | 0.018 ($\pm$ 0.0005) | **1.0** ($\pm$ **0.0274**) | 0.014 ($\pm$ 0.0004) | **1.0** ($\pm$ **0.0285**) | 0.01 ($\pm$ 0.0003) |
| Oracle (infeasible) | **1.038** ($\pm$ **0.026**) | 0.021 ($\pm$ 0.0005) | **1.167** ($\pm$ **0.0298**) | 0.02 ($\pm$ 0.0005) | **1.289** ($\pm$ **0.0294**) | 0.019 ($\pm$ 0.0006) |
| Naive-Trim (invalid) | 1.222 ($\pm$ 0.0269) | 0.028 ($\pm$ 0.0006) | 1.77 ($\pm$ 0.0354) | 0.039 ($\pm$ 0.0007) | 2.198 ($\pm$ 0.0315) | 0.048 ($\pm$ 0.0008) |
| Small-Clean | 0.894 ($\pm$ 0.0609) | 0.019 ($\pm$ 0.0019) | 0.504 ($\pm$ 0.07) | 0.009 ($\pm$ 0.0015) | 0.389 ($\pm$ 0.0722) | 0.006 ($\pm$ 0.0015) |
| Label-Trim | **1.035** ($\pm$ **0.0256**) | 0.02 ($\pm$ 0.0005) | **1.146** ($\pm$ **0.0293**) | 0.019 ($\pm$ 0.0005) | **1.217** ($\pm$ **0.0301**) | 0.017 ($\pm$ 0.0005) |
| Standard Power | 0.249 ($\pm$ 0.0065) | 0.018 ($\pm$ 0.0005) | 0.216 ($\pm$ 0.0059) | 0.014 ($\pm$ 0.0004) | 0.196 ($\pm$ 0.0056) | 0.01 ($\pm$ 0.0003) |

(b) Target type-I error rate $\alpha = 0.02$

| | Contamination rate | | | | | |
| --- | --- | --- | --- | --- | --- | --- |
| | 1% | | 3% | | 5% | |
| Method | Power | Type-I Error | Power | Type-I Error | Power | Type-I Error |
| Standard | **1.0** ($\pm$ **0.0219**) | 0.028 ($\pm$ 0.0006) | **1.0** ($\pm$ **0.0251**) | 0.023 ($\pm$ 0.0005) | **1.0** ($\pm$ **0.0248**) | 0.018 ($\pm$ 0.0005) |
| Oracle (infeasible) | **1.074** ($\pm$ **0.0225**) | 0.031 ($\pm$ 0.0006) | **1.206** ($\pm$ **0.0281**) | 0.03 ($\pm$ 0.0007) | **1.302** ($\pm$ **0.0261**) | 0.029 ($\pm$ 0.0007) |
| Naive-Trim (invalid) | 1.224 ($\pm$ 0.0216) | 0.037 ($\pm$ 0.0007) | 1.626 ($\pm$ 0.0277) | 0.048 ($\pm$ 0.0008) | 1.962 ($\pm$ 0.0257) | 0.056 ($\pm$ 0.0009) |
| Small-Clean | 0.788 ($\pm$ 0.0485) | 0.02 ($\pm$ 0.0019) | 0.878 ($\pm$ 0.0536) | 0.019 ($\pm$ 0.0018) | 0.986 ($\pm$ 0.0625) | 0.02 ($\pm$ 0.0021) |
| Label-Trim | **1.05** ($\pm$ **0.0224**) | 0.03 ($\pm$ 0.0006) | **1.134** ($\pm$ **0.0265**) | 0.028 ($\pm$ 0.0006) | **1.16** ($\pm$ **0.0247**) | 0.024 ($\pm$ 0.0006) |
| Standard Power | 0.307 ($\pm$ 0.0067) | 0.028 ($\pm$ 0.0006) | 0.266 ($\pm$ 0.0067) | 0.023 ($\pm$ 0.0005) | 0.241 ($\pm$ 0.006) | 0.018 ($\pm$ 0.0005) |

(c) Target type-I error rate $\alpha = 0.03$

Table 14: Comparison of conformal outlier detection methods on Places365 dataset (outliers) and CIFAR-10 dataset (inliers) for varying contamination rate $r$ and target type-I error level $\alpha$. All methods utilize the ReAct (Sun et al., 2021) method with a pretrained VGG-19. The empirical power is presented relative to the `Standard` method (higher is better). Results are averaged across 100 random splits of the data, with standard errors presented in parentheses.

| | Contamination rate | | | | | |
| | 1% | | 3% | | 5% | |
| Method | Power | Type-I Error | Power | Type-I Error | Power | Type-I Error |
|---|---|---|---|---|---|---|
| Standard | **1.0 (± 0.0289)** | 0.009 (± 0.0004) | **1.0 (± 0.0383)** | 0.006 (± 0.0003) | **1.0 (± 0.0382)** | 0.005 (± 0.0003) |
| Oracle (infeasible) | **1.084 (± 0.0295)** | 0.01 (± 0.0004) | **1.444 (± 0.0407)** | 0.01 (± 0.0004) | **1.602 (± 0.0488)** | 0.009 (± 0.0003) |
| Naive-Trim (invalid) | 1.454 (± 0.0346) | 0.018 (± 0.0005) | 2.623 (± 0.0521) | 0.03 (± 0.0006) | 3.405 (± 0.0592) | 0.04 (± 0.0008) |
| Small-Clean | 0.0 (± 0.0) | 0.0 (± 0.0) | 0.0 (± 0.0) | 0.0 (± 0.0) | 0.0 (± 0.0) | 0.0 (± 0.0) |
| Label-Trim | **1.084 (± 0.0295)** | 0.01 (± 0.0004) | **1.441 (± 0.0404)** | 0.01 (± 0.0003) | **1.578 (± 0.0476)** | 0.009 (± 0.0003) |
| Standard Power | 0.176 (± 0.0051) | 0.009 (± 0.0004) | 0.134 (± 0.0051) | 0.006 (± 0.0003) | 0.12 (± 0.0046) | 0.005 (± 0.0003) |

(a) Target type-I error rate $\alpha = 0.01$

| | Contamination rate | | | | | |
| | 1% | | 3% | | 5% | |
| Method | Power | Type-I Error | Power | Type-I Error | Power | Type-I Error |
|---|---|---|---|---|---|---|
| Standard | **1.0 (± 0.0236)** | 0.018 (± 0.0005) | **1.0 (± 0.0251)** | 0.014 (± 0.0004) | **1.0 (± 0.0296)** | 0.01 (± 0.0004) |
| Oracle (infeasible) | **1.074 (± 0.0238)** | 0.021 (± 0.0005) | **1.247 (± 0.0291)** | 0.02 (± 0.0005) | **1.35 (± 0.0332)** | 0.019 (± 0.0006) |
| Naive-Trim (invalid) | 1.236 (± 0.0236) | 0.028 (± 0.0006) | 1.781 (± 0.0308) | 0.039 (± 0.0007) | 2.166 (± 0.0346) | 0.048 (± 0.0009) |
| Small-Clean | 0.71 (± 0.0577) | 0.015 (± 0.0017) | 0.569 (± 0.0713) | 0.011 (± 0.0016) | 0.268 (± 0.0606) | 0.004 (± 0.0013) |
| Label-Trim | **1.058 (± 0.0241)** | 0.02 (± 0.0005) | **1.198 (± 0.0281)** | 0.019 (± 0.0005) | **1.231 (± 0.0315)** | 0.016 (± 0.0005) |
| Standard Power | 0.259 (± 0.0061) | 0.018 (± 0.0005) | 0.226 (± 0.0057) | 0.014 (± 0.0004) | 0.205 (± 0.0061) | 0.01 (± 0.0004) |

(b) Target type-I error rate $\alpha = 0.02$

| | Contamination rate | | | | | |
| | 1% | | 3% | | 5% | |
| Method | Power | Type-I Error | Power | Type-I Error | Power | Type-I Error |
|---|---|---|---|---|---|---|
| Standard | **1.0 (± 0.0188)** | 0.028 (± 0.0006) | **1.0 (± 0.0221)** | 0.022 (± 0.0005) | **1.0 (± 0.0261)** | 0.017 (± 0.0005) |
| Oracle (infeasible) | **1.088 (± 0.0187)** | 0.031 (± 0.0006) | **1.197 (± 0.0233)** | 0.03 (± 0.0007) | **1.32 (± 0.0288)** | 0.029 (± 0.0007) |
| Naive-Trim (invalid) | 1.186 (± 0.0185) | 0.037 (± 0.0007) | 1.499 (± 0.0249) | 0.048 (± 0.0008) | 1.83 (± 0.026) | 0.056 (± 0.0009) |
| Small-Clean | 0.652 (± 0.045) | 0.017 (± 0.0017) | 0.85 (± 0.0541) | 0.021 (± 0.0018) | 0.921 (± 0.054) | 0.02 (± 0.002) |
| Label-Trim | **1.056 (± 0.019)** | 0.03 (± 0.0006) | **1.12 (± 0.0232)** | 0.027 (± 0.0006) | **1.168 (± 0.0276)** | 0.023 (± 0.0006) |
| Standard Power | 0.321 (± 0.006) | 0.028 (± 0.0006) | 0.294 (± 0.0065) | 0.022 (± 0.0005) | 0.263 (± 0.0069) | 0.017 (± 0.0005) |

(c) Target type-I error rate $\alpha = 0.03$

Table 15: Comparison of conformal outlier detection methods on MNIST dataset (outliers) and CIFAR-10 dataset (inliers) for varying contamination rate $r$ and target type-I error level $\alpha$. All methods utilize the ReAct (Sun et al., 2021) method with a pretrained VGG-19. The empirical power is presented relative to the `Standard` method (higher is better). Results are averaged across 100 random splits of the data, with standard errors presented in parentheses.

| | Contamination rate | | | | | |
| --- | --- | --- | --- | --- | --- | --- |
| | 1% | | 3% | | 5% | |
| Method | Power | Type-I Error | Power | Type-I Error | Power | Type-I Error |
| Standard | **1.0 ($\pm$ 0.0315)** | 0.009 ($\pm$ 0.0004) | **1.0 ($\pm$ 0.0376)** | 0.007 ($\pm$ 0.0003) | **1.0 ($\pm$ 0.0428)** | 0.005 ($\pm$ 0.0003) |
| Oracle (infeasible) | **1.089 ($\pm$ 0.0325)** | 0.01 ($\pm$ 0.0004) | **1.349 ($\pm$ 0.0414)** | 0.01 ($\pm$ 0.0004) | **1.547 ($\pm$ 0.0488)** | 0.009 ($\pm$ 0.0003) |
| Naive-Trim (invalid) | 1.493 ($\pm$ 0.0352) | 0.018 ($\pm$ 0.0005) | 2.499 ($\pm$ 0.0571) | 0.031 ($\pm$ 0.0006) | 3.315 ($\pm$ 0.0664) | 0.041 ($\pm$ 0.0008) |
| Small-Clean | 0.0 ($\pm$ 0.0) | 0.0 ($\pm$ 0.0) | 0.0 ($\pm$ 0.0) | 0.0 ($\pm$ 0.0) | 0.0 ($\pm$ 0.0) | 0.0 ($\pm$ 0.0) |
| Label-Trim | **1.089 ($\pm$ 0.0325)** | 0.01 ($\pm$ 0.0004) | **1.347 ($\pm$ 0.0414)** | 0.01 ($\pm$ 0.0004) | **1.549 ($\pm$ 0.0492)** | 0.009 ($\pm$ 0.0003) |
| Standard Power | 0.162 ($\pm$ 0.0051) | 0.009 ($\pm$ 0.0004) | 0.134 ($\pm$ 0.005) | 0.007 ($\pm$ 0.0003) | 0.116 ($\pm$ 0.005) | 0.005 ($\pm$ 0.0003) |

(a) Target type-I error rate $\alpha = 0.01$

| | Contamination rate | | | | | |
| --- | --- | --- | --- | --- | --- | --- |
| | 1% | | 3% | | 5% | |
| Method | Power | Type-I Error | Power | Type-I Error | Power | Type-I Error |
| Standard | **1.0 ($\pm$ 0.0234)** | 0.018 ($\pm$ 0.0005) | **1.0 ($\pm$ 0.0283)** | 0.014 ($\pm$ 0.0004) | **1.0 ($\pm$ 0.0298)** | 0.01 ($\pm$ 0.0004) |
| Oracle (infeasible) | **1.068 ($\pm$ 0.0225)** | 0.021 ($\pm$ 0.0005) | **1.219 ($\pm$ 0.0313)** | 0.02 ($\pm$ 0.0005) | **1.36 ($\pm$ 0.0343)** | 0.019 ($\pm$ 0.0006) |
| Naive-Trim (invalid) | 1.246 ($\pm$ 0.0245) | 0.028 ($\pm$ 0.0006) | 1.76 ($\pm$ 0.0341) | 0.039 ($\pm$ 0.0007) | 2.167 ($\pm$ 0.0395) | 0.049 ($\pm$ 0.0009) |
| Small-Clean | 0.781 ($\pm$ 0.0592) | 0.017 ($\pm$ 0.0019) | 0.617 ($\pm$ 0.0743) | 0.012 ($\pm$ 0.0018) | 0.302 ($\pm$ 0.0582) | 0.005 ($\pm$ 0.0011) |
| Label-Trim | **1.053 ($\pm$ 0.0224)** | 0.02 ($\pm$ 0.0005) | **1.176 ($\pm$ 0.0318)** | 0.019 ($\pm$ 0.0005) | **1.242 ($\pm$ 0.0326)** | 0.016 ($\pm$ 0.0005) |
| Standard Power | 0.244 ($\pm$ 0.0057) | 0.018 ($\pm$ 0.0005) | 0.214 ($\pm$ 0.0061) | 0.014 ($\pm$ 0.0004) | 0.194 ($\pm$ 0.0058) | 0.01 ($\pm$ 0.0004) |

(b) Target type-I error rate $\alpha = 0.02$

| | Contamination rate | | | | | |
| --- | --- | --- | --- | --- | --- | --- |
| | 1% | | 3% | | 5% | |
| Method | Power | Type-I Error | Power | Type-I Error | Power | Type-I Error |
| Standard | **1.0 ($\pm$ 0.0195)** | 0.028 ($\pm$ 0.0006) | **1.0 ($\pm$ 0.0247)** | 0.023 ($\pm$ 0.0006) | **1.0 ($\pm$ 0.0262)** | 0.018 ($\pm$ 0.0005) |
| Oracle (infeasible) | **1.062 ($\pm$ 0.0196)** | 0.031 ($\pm$ 0.0006) | **1.189 ($\pm$ 0.0279)** | 0.03 ($\pm$ 0.0007) | **1.296 ($\pm$ 0.0305)** | 0.029 ($\pm$ 0.0007) |
| Naive-Trim (invalid) | 1.178 ($\pm$ 0.0208) | 0.037 ($\pm$ 0.0007) | 1.501 ($\pm$ 0.0257) | 0.049 ($\pm$ 0.0008) | 1.799 ($\pm$ 0.0319) | 0.057 ($\pm$ 0.0009) |
| Small-Clean | 0.648 ($\pm$ 0.0456) | 0.017 ($\pm$ 0.0018) | 0.888 ($\pm$ 0.0515) | 0.023 ($\pm$ 0.002) | 0.964 ($\pm$ 0.0563) | 0.021 ($\pm$ 0.0018) |
| Label-Trim | **1.039 ($\pm$ 0.0196)** | 0.03 ($\pm$ 0.0006) | **1.109 ($\pm$ 0.0258)** | 0.027 ($\pm$ 0.0006) | **1.145 ($\pm$ 0.0268)** | 0.024 ($\pm$ 0.0006) |
| Standard Power | 0.305 ($\pm$ 0.0059) | 0.028 ($\pm$ 0.0006) | 0.276 ($\pm$ 0.0068) | 0.023 ($\pm$ 0.0006) | 0.25 ($\pm$ 0.0066) | 0.018 ($\pm$ 0.0005) |

(c) Target type-I error rate $\alpha = 0.03$

Table 16: Comparison of conformal outlier detection methods on CIFAR-100 dataset (outliers) and CIFAR-10 dataset (inliers) for varying contamination rate $r$ and target type-I error level $\alpha$. All methods utilize the ReAct (Sun et al., 2021) method with a pretrained VGG-19. The empirical power is presented relative to the `Standard` method (higher is better). Results are averaged across 100 random splits of the data, with standard errors presented in parentheses.

| | Contamination rate | | | | | |
| | 1% | | 3% | | 5% | |
| Method | Power | Type-I Error | Power | Type-I Error | Power | Type-I Error |
|---|---|---|---|---|---|---|
| Standard | **1.0 ($\pm$ 0.0388)** | 0.009 ($\pm$ 0.0004) | **1.0 ($\pm$ 0.0466)** | 0.007 ($\pm$ 0.0003) | **1.0 ($\pm$ 0.0472)** | 0.005 ($\pm$ 0.0003) |
| Oracle (infeasible) | **1.089 ($\pm$ 0.0409)** | 0.01 ($\pm$ 0.0004) | **1.34 ($\pm$ 0.0508)** | 0.01 ($\pm$ 0.0004) | **1.554 ($\pm$ 0.0539)** | 0.009 ($\pm$ 0.0003) |
| Naive-Trim (invalid) | 1.558 ($\pm$ 0.0455) | 0.018 ($\pm$ 0.0005) | 2.816 ($\pm$ 0.0647) | 0.032 ($\pm$ 0.0007) | 3.58 ($\pm$ 0.0691) | 0.042 ($\pm$ 0.0008) |
| Small-Clean | 0.0 ($\pm$ 0.0) | 0.0 ($\pm$ 0.0) | 0.0 ($\pm$ 0.0) | 0.0 ($\pm$ 0.0) | 0.0 ($\pm$ 0.0) | 0.0 ($\pm$ 0.0) |
| Label-Trim | **1.089 ($\pm$ 0.0409)** | 0.01 ($\pm$ 0.0004) | **1.34 ($\pm$ 0.0508)** | 0.01 ($\pm$ 0.0004) | **1.572 ($\pm$ 0.053)** | 0.009 ($\pm$ 0.0003) |
| Standard Power | 0.137 ($\pm$ 0.0053) | 0.009 ($\pm$ 0.0004) | 0.111 ($\pm$ 0.0052) | 0.007 ($\pm$ 0.0003) | 0.1 ($\pm$ 0.0047) | 0.005 ($\pm$ 0.0003) |

(a) Target type-I error rate $\alpha = 0.01$

| | Contamination rate | | | | | |
| | 1% | | 3% | | 5% | |
| Method | Power | Type-I Error | Power | Type-I Error | Power | Type-I Error |
|---|---|---|---|---|---|---|
| Standard | **1.0 ($\pm$ 0.0288)** | 0.019 ($\pm$ 0.0005) | **1.0 ($\pm$ 0.0308)** | 0.015 ($\pm$ 0.0005) | **1.0 ($\pm$ 0.0307)** | 0.012 ($\pm$ 0.0004) |
| Oracle (infeasible) | **1.07 ($\pm$ 0.0289)** | 0.021 ($\pm$ 0.0005) | **1.219 ($\pm$ 0.0332)** | 0.02 ($\pm$ 0.0005) | **1.316 ($\pm$ 0.0362)** | 0.019 ($\pm$ 0.0006) |
| Naive-Trim (invalid) | 1.293 ($\pm$ 0.0301) | 0.028 ($\pm$ 0.0006) | 1.811 ($\pm$ 0.0395) | 0.04 ($\pm$ 0.0007) | 2.24 ($\pm$ 0.0396) | 0.051 ($\pm$ 0.0009) |
| Small-Clean | 0.831 ($\pm$ 0.0636) | 0.018 ($\pm$ 0.002) | 0.646 ($\pm$ 0.0826) | 0.013 ($\pm$ 0.0021) | 0.317 ($\pm$ 0.0665) | 0.005 ($\pm$ 0.0016) |
| Label-Trim | **1.057 ($\pm$ 0.0281)** | 0.02 ($\pm$ 0.0005) | **1.17 ($\pm$ 0.0327)** | 0.019 ($\pm$ 0.0005) | **1.233 ($\pm$ 0.0343)** | 0.017 ($\pm$ 0.0006) |
| Standard Power | 0.217 ($\pm$ 0.0063) | 0.019 ($\pm$ 0.0005) | 0.192 ($\pm$ 0.0059) | 0.015 ($\pm$ 0.0005) | 0.175 ($\pm$ 0.0054) | 0.012 ($\pm$ 0.0004) |

(b) Target type-I error rate $\alpha = 0.02$

| | Contamination rate | | | | | |
| | 1% | | 3% | | 5% | |
| Method | Power | Type-I Error | Power | Type-I Error | Power | Type-I Error |
|---|---|---|---|---|---|---|
| Standard | **1.0 ($\pm$ 0.023)** | 0.028 ($\pm$ 0.0006) | **1.0 ($\pm$ 0.0272)** | 0.023 ($\pm$ 0.0006) | **1.0 ($\pm$ 0.0269)** | 0.019 ($\pm$ 0.0005) |
| Oracle (infeasible) | **1.061 ($\pm$ 0.0233)** | 0.031 ($\pm$ 0.0006) | **1.183 ($\pm$ 0.0278)** | 0.03 ($\pm$ 0.0007) | **1.292 ($\pm$ 0.0306)** | 0.029 ($\pm$ 0.0007) |
| Naive-Trim (invalid) | 1.184 ($\pm$ 0.0249) | 0.038 ($\pm$ 0.0007) | 1.512 ($\pm$ 0.0294) | 0.05 ($\pm$ 0.0008) | 1.868 ($\pm$ 0.0324) | 0.059 ($\pm$ 0.0009) |
| Small-Clean | 0.706 ($\pm$ 0.0461) | 0.02 ($\pm$ 0.002) | 0.816 ($\pm$ 0.0583) | 0.021 ($\pm$ 0.0022) | 0.856 ($\pm$ 0.0579) | 0.018 ($\pm$ 0.0021) |
| Label-Trim | **1.039 ($\pm$ 0.0226)** | 0.03 ($\pm$ 0.0006) | **1.111 ($\pm$ 0.0275)** | 0.028 ($\pm$ 0.0006) | **1.154 ($\pm$ 0.0291)** | 0.024 ($\pm$ 0.0006) |
| Standard Power | 0.282 ($\pm$ 0.0065) | 0.028 ($\pm$ 0.0006) | 0.254 ($\pm$ 0.0069) | 0.023 ($\pm$ 0.0006) | 0.228 ($\pm$ 0.0062) | 0.019 ($\pm$ 0.0005) |

(c) Target type-I error rate $\alpha = 0.03$

Table 17: Comparison of conformal outlier detection methods on TinyImageNet dataset (outliers) and CIFAR-10 dataset (inliers) for varying contamination rate $r$ and target type-I error level $\alpha$. All methods utilize the ReAct (Sun et al., 2021) method with a pretrained VGG-19. The empirical power is presented relative to the `Standard` method (higher is better). Results are averaged across 100 random splits of the data, with standard errors presented in parentheses.

| | Contamination rate | | | | | |
| | 1% | | 3% | | 5% | |
| Method | Power | Type-I Error | Power | Type-I Error | Power | Type-I Error |
| --- | --- | --- | --- | --- | --- | --- |
| Standard | **1.0 ($\pm$ 0.0353)** | 0.009 ($\pm$ 0.0004) | **1.0 ($\pm$ 0.037)** | 0.007 ($\pm$ 0.0003) | **1.0 ($\pm$ 0.0446)** | 0.005 ($\pm$ 0.0002) |
| Oracle (infeasible) | **1.104 ($\pm$ 0.0383)** | 0.01 ($\pm$ 0.0004) | **1.37 ($\pm$ 0.0474)** | 0.01 ($\pm$ 0.0004) | **1.653 ($\pm$ 0.0523)** | 0.009 ($\pm$ 0.0003) |
| Naive-Trim (invalid) | 1.556 ($\pm$ 0.0452) | 0.018 ($\pm$ 0.0005) | 2.635 ($\pm$ 0.0555) | 0.031 ($\pm$ 0.0006) | 3.796 ($\pm$ 0.0594) | 0.04 ($\pm$ 0.0008) |
| Small-Clean | 0.0 ($\pm$ 0.0) | 0.0 ($\pm$ 0.0) | 0.0 ($\pm$ 0.0) | 0.0 ($\pm$ 0.0) | 0.0 ($\pm$ 0.0) | 0.0 ($\pm$ 0.0) |
| Label-Trim | **1.104 ($\pm$ 0.0383)** | 0.01 ($\pm$ 0.0004) | **1.37 ($\pm$ 0.0474)** | 0.01 ($\pm$ 0.0004) | **1.651 ($\pm$ 0.0526)** | 0.009 ($\pm$ 0.0003) |
| Standard Power | 0.151 ($\pm$ 0.0053) | 0.009 ($\pm$ 0.0004) | 0.128 ($\pm$ 0.0047) | 0.007 ($\pm$ 0.0003) | 0.105 ($\pm$ 0.0047) | 0.005 ($\pm$ 0.0002) |

(a) Target type-I error rate $\alpha = 0.01$

| | Contamination rate | | | | | |
| | 1% | | 3% | | 5% | |
| Method | Power | Type-I Error | Power | Type-I Error | Power | Type-I Error |
| --- | --- | --- | --- | --- | --- | --- |
| Standard | **1.0 ($\pm$ 0.0284)** | 0.019 ($\pm$ 0.0005) | **1.0 ($\pm$ 0.0279)** | 0.014 ($\pm$ 0.0004) | **1.0 ($\pm$ 0.0313)** | 0.011 ($\pm$ 0.0004) |
| Oracle (infeasible) | **1.074 ($\pm$ 0.0281)** | 0.021 ($\pm$ 0.0005) | **1.221 ($\pm$ 0.0303)** | 0.02 ($\pm$ 0.0005) | **1.4 ($\pm$ 0.0317)** | 0.019 ($\pm$ 0.0006) |
| Naive-Trim (invalid) | 1.295 ($\pm$ 0.0281) | 0.028 ($\pm$ 0.0006) | 1.826 ($\pm$ 0.0314) | 0.039 ($\pm$ 0.0007) | 2.351 ($\pm$ 0.0323) | 0.049 ($\pm$ 0.0008) |
| Small-Clean | 0.823 ($\pm$ 0.0624) | 0.017 ($\pm$ 0.0018) | 0.471 ($\pm$ 0.0636) | 0.008 ($\pm$ 0.0012) | 0.401 ($\pm$ 0.0691) | 0.005 ($\pm$ 0.001) |
| Label-Trim | **1.06 ($\pm$ 0.0275)** | 0.02 ($\pm$ 0.0005) | **1.173 ($\pm$ 0.0302)** | 0.019 ($\pm$ 0.0005) | **1.272 ($\pm$ 0.0311)** | 0.016 ($\pm$ 0.0005) |
| Standard Power | 0.239 ($\pm$ 0.0068) | 0.019 ($\pm$ 0.0005) | 0.213 ($\pm$ 0.0059) | 0.014 ($\pm$ 0.0004) | 0.187 ($\pm$ 0.0059) | 0.011 ($\pm$ 0.0004) |

(b) Target type-I error rate $\alpha = 0.02$

| | Contamination rate | | | | | |
| | 1% | | 3% | | 5% | |
| Method | Power | Type-I Error | Power | Type-I Error | Power | Type-I Error |
| --- | --- | --- | --- | --- | --- | --- |
| Standard | **1.0 ($\pm$ 0.0217)** | 0.028 ($\pm$ 0.0006) | **1.0 ($\pm$ 0.0239)** | 0.023 ($\pm$ 0.0005) | **1.0 ($\pm$ 0.0239)** | 0.018 ($\pm$ 0.0005) |
| Oracle (infeasible) | **1.086 ($\pm$ 0.0207)** | 0.031 ($\pm$ 0.0006) | **1.209 ($\pm$ 0.0264)** | 0.03 ($\pm$ 0.0007) | **1.344 ($\pm$ 0.0247)** | 0.029 ($\pm$ 0.0007) |
| Naive-Trim (invalid) | 1.196 ($\pm$ 0.0201) | 0.037 ($\pm$ 0.0007) | 1.512 ($\pm$ 0.0223) | 0.048 ($\pm$ 0.0008) | 1.879 ($\pm$ 0.0245) | 0.057 ($\pm$ 0.0009) |
| Small-Clean | 0.662 ($\pm$ 0.0464) | 0.018 ($\pm$ 0.0018) | 0.733 ($\pm$ 0.05) | 0.017 ($\pm$ 0.0016) | 0.848 ($\pm$ 0.0556) | 0.017 ($\pm$ 0.0018) |
| Label-Trim | **1.054 ($\pm$ 0.0218)** | 0.03 ($\pm$ 0.0006) | **1.12 ($\pm$ 0.0256)** | 0.027 ($\pm$ 0.0006) | **1.167 ($\pm$ 0.0254)** | 0.023 ($\pm$ 0.0006) |
| Standard Power | 0.313 ($\pm$ 0.0068) | 0.028 ($\pm$ 0.0006) | 0.278 ($\pm$ 0.0066) | 0.023 ($\pm$ 0.0005) | 0.252 ($\pm$ 0.006) | 0.018 ($\pm$ 0.0005) |

(c) Target type-I error rate $\alpha = 0.03$

Table 18: Comparison of conformal outlier detection methods on Texture dataset (outliers) and CIFAR-10 dataset (inliers) for varying contamination rate $r$ and target type-I error level $\alpha$. All methods utilize the SCALE (Xu et al., 2024) method with a pretrained ResNet-18. The empirical power is presented relative to the `Standard` method (higher is better). Results are averaged across 100 random splits of the data, with standard errors presented in parentheses.

|  | Contamination rate | | | | | |
|---|---|---|---|---|---|---|
|  | 1% | | 3% | | 5% | |
| Method | Power | Type-I Error | Power | Type-I Error | Power | Type-I Error |
| Standard | **1.0 (± 0.0416)** | 0.009 (± 0.0003) | **1.0 (± 0.0373)** | 0.006 (± 0.0003) | **1.0 (± 0.036)** | 0.005 (± 0.0002) |
| Oracle (infeasible) | **1.129 (± 0.0436)** | 0.01 (± 0.0003) | **1.399 (± 0.0502)** | 0.01 (± 0.0003) | **1.545 (± 0.0506)** | 0.01 (± 0.0004) |
| Naive-Trim (invalid) | 1.757 (± 0.0508) | 0.018 (± 0.0004) | 2.978 (± 0.061) | 0.029 (± 0.0006) | 4.043 (± 0.0598) | 0.04 (± 0.0007) |
| Small-Clean | 0.0 (± 0.0) | 0.0 (± 0.0) | 0.0 (± 0.0) | 0.0 (± 0.0) | 0.0 (± 0.0) | 0.0 (± 0.0) |
| Label-Trim | **1.129 (± 0.0436)** | 0.01 (± 0.0003) | **1.399 (± 0.0502)** | 0.01 (± 0.0003) | **1.526 (± 0.0482)** | 0.009 (± 0.0003) |
| Standard Power | 0.146 (± 0.0061) | 0.009 (± 0.0003) | 0.125 (± 0.0047) | 0.006 (± 0.0003) | 0.107 (± 0.0038) | 0.005 (± 0.0002) |

(a) $\alpha = 0.01$

|  | Contamination rate | | | | | |
|---|---|---|---|---|---|---|
|  | 1% | | 3% | | 5% | |
| Method | Power | Type-I Error | Power | Type-I Error | Power | Type-I Error |
| Standard | **1.0 (± 0.029)** | 0.018 (± 0.0004) | **1.0 (± 0.0269)** | 0.014 (± 0.0004) | **1.0 (± 0.0286)** | 0.011 (± 0.0003) |
| Oracle (infeasible) | **1.089 (± 0.0286)** | 0.021 (± 0.0005) | **1.292 (± 0.0312)** | 0.02 (± 0.0005) | **1.518 (± 0.0278)** | 0.02 (± 0.0005) |
| Naive-Trim (invalid) | 1.295 (± 0.0312) | 0.027 (± 0.0006) | 1.904 (± 0.0338) | 0.037 (± 0.0006) | 2.51 (± 0.0339) | 0.048 (± 0.0008) |
| Small-Clean | 0.792 (± 0.0585) | 0.017 (± 0.0018) | 0.636 (± 0.0789) | 0.012 (± 0.0019) | 0.288 (± 0.0601) | 0.005 (± 0.0013) |
| Label-Trim | **1.069 (± 0.0285)** | 0.02 (± 0.0005) | **1.228 (± 0.0309)** | 0.018 (± 0.0005) | **1.333 (± 0.0291)** | 0.015 (± 0.0004) |
| Standard Power | 0.259 (± 0.0075) | 0.018 (± 0.0004) | 0.226 (± 0.0061) | 0.014 (± 0.0004) | 0.188 (± 0.0054) | 0.011 (± 0.0003) |

(b) $\alpha = 0.02$

|  | Contamination rate | | | | | |
|---|---|---|---|---|---|---|
|  | 1% | | 3% | | 5% | |
| Method | Power | Type-I Error | Power | Type-I Error | Power | Type-I Error |
| Standard | **1.0 (± 0.024)** | 0.028 (± 0.0006) | **1.0 (± 0.0222)** | 0.022 (± 0.0005) | **1.0 (± 0.0205)** | 0.017 (± 0.0005) |
| Oracle (infeasible) | **1.089 (± 0.0227)** | 0.031 (± 0.0006) | **1.228 (± 0.0248)** | 0.029 (± 0.0006) | **1.394 (± 0.0218)** | 0.03 (± 0.0006) |
| Naive-Trim (invalid) | 1.204 (± 0.023) | 0.036 (± 0.0006) | 1.557 (± 0.0251) | 0.045 (± 0.0007) | 1.919 (± 0.0242) | 0.056 (± 0.0009) |
| Small-Clean | 0.659 (± 0.0441) | 0.019 (± 0.0018) | 0.809 (± 0.0541) | 0.019 (± 0.002) | 0.837 (± 0.0579) | 0.017 (± 0.002) |
| Label-Trim | **1.057 (± 0.0231)** | 0.03 (± 0.0006) | **1.115 (± 0.0238)** | 0.026 (± 0.0006) | **1.173 (± 0.021)** | 0.023 (± 0.0005) |
| Standard Power | 0.338 (± 0.0081) | 0.028 (± 0.0006) | 0.307 (± 0.0068) | 0.022 (± 0.0005) | 0.263 (± 0.0054) | 0.017 (± 0.0005) |

(c) $\alpha = 0.03$

Table 19: Comparison of conformal outlier detection methods on SVHN dataset (outliers) and CIFAR-10 dataset (inliers) for varying contamination rate $r$ and target type-I error level $\alpha$. All methods utilize the SCALE (Xu et al., 2024) method with a pretrained ResNet-18. The empirical power is presented relative to the `Standard` method (higher is better). Results are averaged across 100 random splits of the data, with standard errors presented in parentheses.

| | Contamination rate | | | | | |
| | 1% | | 3% | | 5% | |
| Method | Power | Type-I Error | Power | Type-I Error | Power | Type-I Error |
|---|---|---|---|---|---|---|
| Standard | **1.0 (± 0.0308)** | 0.009 (± 0.0003) | **1.0 (± 0.0386)** | 0.006 (± 0.0003) | **1.0 (± 0.0401)** | 0.004 (± 0.0002) |
| Oracle (infeasible) | **1.144 (± 0.0321)** | 0.01 (± 0.0003) | **1.526 (± 0.0466)** | 0.01 (± 0.0003) | **1.868 (± 0.058)** | 0.01 (± 0.0004) |
| Naive-Trim (invalid) | 1.743 (± 0.033) | 0.017 (± 0.0004) | 2.838 (± 0.0519) | 0.027 (± 0.0005) | 4.211 (± 0.0669) | 0.038 (± 0.0007) |
| Small-Clean | 0.0 (± 0.0) | 0.0 (± 0.0) | 0.0 (± 0.0) | 0.0 (± 0.0) | 0.0 (± 0.0) | 0.0 (± 0.0) |
| Label-Trim | **1.144 (± 0.0321)** | 0.01 (± 0.0003) | **1.513 (± 0.0462)** | 0.01 (± 0.0003) | **1.749 (± 0.0529)** | 0.009 (± 0.0003) |
| Standard Power | 0.191 (± 0.0059) | 0.009 (± 0.0003) | 0.143 (± 0.0055) | 0.006 (± 0.0003) | 0.115 (± 0.0046) | 0.004 (± 0.0002) |

(a) $\alpha = 0.01$

| | Contamination rate | | | | | |
| | 1% | | 3% | | 5% | |
| Method | Power | Type-I Error | Power | Type-I Error | Power | Type-I Error |
|---|---|---|---|---|---|---|
| Standard | **1.0 (± 0.0189)** | 0.017 (± 0.0004) | **1.0 (± 0.027)** | 0.012 (± 0.0003) | **1.0 (± 0.0299)** | 0.01 (± 0.0003) |
| Oracle (infeasible) | **1.082 (± 0.0204)** | 0.021 (± 0.0005) | **1.342 (± 0.0262)** | 0.02 (± 0.0005) | **1.642 (± 0.0331)** | 0.02 (± 0.0005) |
| Naive-Trim (invalid) | 1.214 (± 0.021) | 0.027 (± 0.0006) | 1.858 (± 0.0271) | 0.036 (± 0.0006) | 2.462 (± 0.0359) | 0.046 (± 0.0008) |
| Small-Clean | 0.788 (± 0.0566) | 0.018 (± 0.002) | 0.662 (± 0.0764) | 0.011 (± 0.0015) | 0.46 (± 0.0819) | 0.007 (± 0.0016) |
| Label-Trim | **1.068 (± 0.0207)** | 0.02 (± 0.0005) | **1.253 (± 0.0275)** | 0.017 (± 0.0005) | **1.403 (± 0.0332)** | 0.014 (± 0.0004) |
| Standard Power | 0.336 (± 0.0064) | 0.017 (± 0.0004) | 0.256 (± 0.0069) | 0.012 (± 0.0003) | 0.216 (± 0.0064) | 0.01 (± 0.0003) |

(b) $\alpha = 0.02$

| | Contamination rate | | | | | |
| | 1% | | 3% | | 5% | |
| Method | Power | Type-I Error | Power | Type-I Error | Power | Type-I Error |
|---|---|---|---|---|---|---|
| Standard | **1.0 (± 0.0173)** | 0.027 (± 0.0006) | **1.0 (± 0.0196)** | 0.02 (± 0.0005) | **1.0 (± 0.0229)** | 0.015 (± 0.0004) |
| Oracle (infeasible) | **1.075 (± 0.017)** | 0.031 (± 0.0006) | **1.247 (± 0.0215)** | 0.029 (± 0.0006) | **1.39 (± 0.0246)** | 0.03 (± 0.0006) |
| Naive-Trim (invalid) | 1.162 (± 0.0163) | 0.036 (± 0.0006) | 1.519 (± 0.0195) | 0.044 (± 0.0007) | 1.805 (± 0.0245) | 0.054 (± 0.0008) |
| Small-Clean | 0.686 (± 0.0437) | 0.019 (± 0.002) | 0.857 (± 0.0528) | 0.019 (± 0.0017) | 0.913 (± 0.0586) | 0.018 (± 0.0018) |
| Label-Trim | **1.046 (± 0.0176)** | 0.029 (± 0.0006) | **1.122 (± 0.0212)** | 0.025 (± 0.0005) | **1.179 (± 0.0225)** | 0.022 (± 0.0005) |
| Standard Power | 0.411 (± 0.0071) | 0.027 (± 0.0006) | 0.342 (± 0.0067) | 0.02 (± 0.0005) | 0.311 (± 0.0071) | 0.015 (± 0.0004) |

(c) $\alpha = 0.03$

Table 20: Comparison of conformal outlier detection methods on Places365 dataset (outliers) and CIFAR-10 dataset (inliers) for varying contamination rate $r$ and target type-I error level $\alpha$. All methods utilize the SCALE (Xu et al., 2024) method with a pretrained ResNet-18. The empirical power is presented relative to the `Standard` method (higher is better). Results are averaged across 100 random splits of the data, with standard errors presented in parentheses.

| | Contamination rate | | | | | |
| | 1% | | 3% | | 5% | |
| Method | Power | Type-I Error | Power | Type-I Error | Power | Type-I Error |
|---|---|---|---|---|---|---|
| Standard | **1.0 ($\pm$ 0.0354)** | 0.009 ($\pm$ 0.0003) | **1.0 ($\pm$ 0.0381)** | 0.006 ($\pm$ 0.0003) | **1.0 ($\pm$ 0.0462)** | 0.004 ($\pm$ 0.0002) |
| Oracle (infeasible) | **1.164 ($\pm$ 0.0361)** | 0.01 ($\pm$ 0.0003) | **1.489 ($\pm$ 0.0456)** | 0.01 ($\pm$ 0.0003) | **1.773 ($\pm$ 0.0555)** | 0.01 ($\pm$ 0.0004) |
| Naive-Trim (invalid) | 1.729 ($\pm$ 0.0387) | 0.017 ($\pm$ 0.0004) | 2.944 ($\pm$ 0.0532) | 0.028 ($\pm$ 0.0006) | 4.136 ($\pm$ 0.0545) | 0.038 ($\pm$ 0.0007) |
| Small-Clean | 0.0 ($\pm$ 0.0) | 0.0 ($\pm$ 0.0) | 0.0 ($\pm$ 0.0) | 0.0 ($\pm$ 0.0) | 0.0 ($\pm$ 0.0) | 0.0 ($\pm$ 0.0) |
| Label-Trim | **1.164 ($\pm$ 0.0361)** | 0.01 ($\pm$ 0.0003) | **1.483 ($\pm$ 0.0452)** | 0.01 ($\pm$ 0.0003) | **1.704 ($\pm$ 0.0533)** | 0.009 ($\pm$ 0.0003) |
| Standard Power | 0.174 ($\pm$ 0.0061) | 0.009 ($\pm$ 0.0003) | 0.133 ($\pm$ 0.0051) | 0.006 ($\pm$ 0.0003) | 0.112 ($\pm$ 0.0052) | 0.004 ($\pm$ 0.0002) |

(a) $\alpha = 0.01$

| | Contamination rate | | | | | |
| | 1% | | 3% | | 5% | |
| Method | Power | Type-I Error | Power | Type-I Error | Power | Type-I Error |
|---|---|---|---|---|---|---|
| Standard | **1.0 ($\pm$ 0.0221)** | 0.018 ($\pm$ 0.0004) | **1.0 ($\pm$ 0.0258)** | 0.013 ($\pm$ 0.0004) | **1.0 ($\pm$ 0.0296)** | 0.01 ($\pm$ 0.0003) |
| Oracle (infeasible) | **1.068 ($\pm$ 0.0237)** | 0.021 ($\pm$ 0.0005) | **1.291 ($\pm$ 0.0272)** | 0.02 ($\pm$ 0.0005) | **1.564 ($\pm$ 0.0325)** | 0.02 ($\pm$ 0.0005) |
| Naive-Trim (invalid) | 1.233 ($\pm$ 0.0249) | 0.027 ($\pm$ 0.0006) | 1.782 ($\pm$ 0.0278) | 0.036 ($\pm$ 0.0006) | 2.483 ($\pm$ 0.0308) | 0.046 ($\pm$ 0.0008) |
| Small-Clean | 0.823 ($\pm$ 0.0588) | 0.018 ($\pm$ 0.0019) | 0.595 ($\pm$ 0.074) | 0.011 ($\pm$ 0.0016) | 0.331 ($\pm$ 0.0746) | 0.005 ($\pm$ 0.0014) |
| Label-Trim | **1.058 ($\pm$ 0.0236)** | 0.02 ($\pm$ 0.0005) | **1.212 ($\pm$ 0.0272)** | 0.018 ($\pm$ 0.0005) | **1.357 ($\pm$ 0.0312)** | 0.015 ($\pm$ 0.0004) |
| Standard Power | 0.302 ($\pm$ 0.0067) | 0.018 ($\pm$ 0.0004) | 0.25 ($\pm$ 0.0065) | 0.013 ($\pm$ 0.0004) | 0.207 ($\pm$ 0.0061) | 0.01 ($\pm$ 0.0003) |

(b) $\alpha = 0.02$

| | Contamination rate | | | | | |
| | 1% | | 3% | | 5% | |
| Method | Power | Type-I Error | Power | Type-I Error | Power | Type-I Error |
|---|---|---|---|---|---|---|
| Standard | **1.0 ($\pm$ 0.0201)** | 0.028 ($\pm$ 0.0006) | **1.0 ($\pm$ 0.0203)** | 0.021 ($\pm$ 0.0005) | 1.0 ($\pm$ 0.0225) | 0.016 ($\pm$ 0.0004) |
| Oracle (infeasible) | **1.084 ($\pm$ 0.0218)** | 0.031 ($\pm$ 0.0006) | **1.229 ($\pm$ 0.0211)** | 0.029 ($\pm$ 0.0006) | **1.383 ($\pm$ 0.0228)** | 0.03 ($\pm$ 0.0006) |
| Naive-Trim (invalid) | 1.186 ($\pm$ 0.0217) | 0.036 ($\pm$ 0.0006) | 1.534 ($\pm$ 0.0216) | 0.044 ($\pm$ 0.0007) | 1.866 ($\pm$ 0.0213) | 0.055 ($\pm$ 0.0009) |
| Small-Clean | 0.712 ($\pm$ 0.0449) | 0.019 ($\pm$ 0.0018) | 0.832 ($\pm$ 0.0525) | 0.019 ($\pm$ 0.0018) | **1.025 ($\pm$ 0.0644)** | 0.023 ($\pm$ 0.0022) |
| Label-Trim | **1.044 ($\pm$ 0.0206)** | 0.029 ($\pm$ 0.0006) | **1.121 ($\pm$ 0.0208)** | 0.026 ($\pm$ 0.0005) | **1.167 ($\pm$ 0.0219)** | 0.022 ($\pm$ 0.0005) |
| Standard Power | 0.374 ($\pm$ 0.0075) | 0.028 ($\pm$ 0.0006) | 0.327 ($\pm$ 0.0066) | 0.021 ($\pm$ 0.0005) | 0.291 ($\pm$ 0.0065) | 0.016 ($\pm$ 0.0004) |

(c) $\alpha = 0.03$

Table 21: Comparison of conformal outlier detection methods on MNIST dataset (outliers) and CIFAR-10 dataset (inliers) for varying contamination rate $r$ and target type-I error level $\alpha$. All methods utilize the SCALE (Xu et al., 2024) method with a pretrained ResNet-18. The empirical power is presented relative to the `Standard` method (higher is better). Results are averaged across 100 random splits of the data, with standard errors presented in parentheses.

| | Contamination rate | | | | | |
| | 1% | | 3% | | 5% | |
| Method | Power | Type-I Error | Power | Type-I Error | Power | Type-I Error |
|---|---|---|---|---|---|---|
| Standard | **1.0 ($\pm$ 0.0286)** | 0.008 ($\pm$ 0.0003) | **1.0 ($\pm$ 0.0284)** | 0.004 ($\pm$ 0.0002) | **1.0 ($\pm$ 0.0366)** | 0.003 ($\pm$ 0.0002) |
| Oracle (infeasible) | **1.174 ($\pm$ 0.0308)** | 0.01 ($\pm$ 0.0003) | **1.625 ($\pm$ 0.0389)** | 0.01 ($\pm$ 0.0003) | **1.961 ($\pm$ 0.045)** | 0.01 ($\pm$ 0.0004) |
| Naive-Trim (invalid) | 1.641 ($\pm$ 0.0326) | 0.016 ($\pm$ 0.0004) | 2.781 ($\pm$ 0.0382) | 0.025 ($\pm$ 0.0006) | 3.78 ($\pm$ 0.0441) | 0.032 ($\pm$ 0.0006) |
| Small-Clean | 0.0 ($\pm$ 0.0) | 0.0 ($\pm$ 0.0) | 0.0 ($\pm$ 0.0) | 0.0 ($\pm$ 0.0) | 0.0 ($\pm$ 0.0) | 0.0 ($\pm$ 0.0) |
| Label-Trim | **1.174 ($\pm$ 0.0308)** | 0.01 ($\pm$ 0.0003) | **1.555 ($\pm$ 0.0366)** | 0.01 ($\pm$ 0.0003) | **1.701 ($\pm$ 0.041)** | 0.008 ($\pm$ 0.0003) |
| Standard Power | 0.251 ($\pm$ 0.0072) | 0.008 ($\pm$ 0.0003) | 0.181 ($\pm$ 0.0051) | 0.004 ($\pm$ 0.0002) | 0.152 ($\pm$ 0.0056) | 0.003 ($\pm$ 0.0002) |

(a) $\alpha = 0.01$

| | Contamination rate | | | | | |
| | 1% | | 3% | | 5% | |
| Method | Power | Type-I Error | Power | Type-I Error | Power | Type-I Error |
|---|---|---|---|---|---|---|
| Standard | **1.0 ($\pm$ 0.0195)** | 0.017 ($\pm$ 0.0004) | **1.0 ($\pm$ 0.0216)** | 0.011 ($\pm$ 0.0003) | **1.0 ($\pm$ 0.0244)** | 0.008 ($\pm$ 0.0003) |
| Oracle (infeasible) | **1.095 ($\pm$ 0.0198)** | 0.021 ($\pm$ 0.0005) | **1.416 ($\pm$ 0.0218)** | 0.02 ($\pm$ 0.0005) | **1.776 ($\pm$ 0.0279)** | 0.02 ($\pm$ 0.0005) |
| Naive-Trim (invalid) | 1.212 ($\pm$ 0.0195) | 0.026 ($\pm$ 0.0006) | 1.84 ($\pm$ 0.0223) | 0.032 ($\pm$ 0.0006) | 2.456 ($\pm$ 0.0243) | 0.04 ($\pm$ 0.0007) |
| Small-Clean | 0.849 ($\pm$ 0.0589) | 0.02 ($\pm$ 0.0021) | 0.605 ($\pm$ 0.0715) | 0.009 ($\pm$ 0.0015) | 0.366 ($\pm$ 0.0772) | 0.005 ($\pm$ 0.0014) |
| Label-Trim | **1.076 ($\pm$ 0.0197)** | 0.02 ($\pm$ 0.0005) | **1.3 ($\pm$ 0.0227)** | 0.016 ($\pm$ 0.0005) | **1.424 ($\pm$ 0.0272)** | 0.013 ($\pm$ 0.0004) |
| Standard Power | 0.415 ($\pm$ 0.0081) | 0.017 ($\pm$ 0.0004) | 0.317 ($\pm$ 0.0069) | 0.011 ($\pm$ 0.0003) | 0.257 ($\pm$ 0.0063) | 0.008 ($\pm$ 0.0003) |

(b) $\alpha = 0.02$

| | Contamination rate | | | | | |
| | 1% | | 3% | | 5% | |
| Method | Power | Type-I Error | Power | Type-I Error | Power | Type-I Error |
|---|---|---|---|---|---|---|
| Standard | **1.0 ($\pm$ 0.0161)** | 0.026 ($\pm$ 0.0006) | **1.0 ($\pm$ 0.0164)** | 0.018 ($\pm$ 0.0005) | 1.0 ($\pm$ 0.0191) | 0.012 ($\pm$ 0.0004) |
| Oracle (infeasible) | **1.086 ($\pm$ 0.0154)** | 0.031 ($\pm$ 0.0006) | **1.272 ($\pm$ 0.0167)** | 0.029 ($\pm$ 0.0006) | **1.518 ($\pm$ 0.02)** | 0.03 ($\pm$ 0.0006) |
| Naive-Trim (invalid) | 1.161 ($\pm$ 0.0144) | 0.035 ($\pm$ 0.0006) | 1.483 ($\pm$ 0.0168) | 0.041 ($\pm$ 0.0007) | 1.852 ($\pm$ 0.0172) | 0.048 ($\pm$ 0.0008) |
| Small-Clean | 0.778 ($\pm$ 0.0434) | 0.022 ($\pm$ 0.002) | 0.848 ($\pm$ 0.0471) | 0.018 ($\pm$ 0.0018) | **1.075 ($\pm$ 0.0562)** | 0.02 ($\pm$ 0.0021) |
| Label-Trim | **1.056 ($\pm$ 0.0161)** | 0.029 ($\pm$ 0.0006) | **1.138 ($\pm$ 0.0162)** | 0.024 ($\pm$ 0.0005) | **1.237 ($\pm$ 0.0195)** | 0.019 ($\pm$ 0.0005) |
| Standard Power | 0.505 ($\pm$ 0.0081) | 0.026 ($\pm$ 0.0006) | 0.43 ($\pm$ 0.007) | 0.018 ($\pm$ 0.0005) | 0.362 ($\pm$ 0.0069) | 0.012 ($\pm$ 0.0004) |

(c) $\alpha = 0.03$

Table 22: Comparison of conformal outlier detection methods on CIFAR-100 dataset (outliers) and CIFAR-10 dataset (inliers) for varying contamination rate $r$ and target type-I error level $\alpha$. All methods utilize the SCALE (Xu et al., 2024) method with a pretrained ResNet-18. The empirical power is presented relative to the `Standard` method (higher is better). Results are averaged across 100 random splits of the data, with standard errors presented in parentheses.

| | Contamination rate | | | | | |
| | 1% | | 3% | | 5% | |
| Method | Power | Type-I Error | Power | Type-I Error | Power | Type-I Error |
| --- | --- | --- | --- | --- | --- | --- |
| Standard | **1.0 (± 0.0399)** | 0.009 (± 0.0003) | **1.0 (± 0.0432)** | 0.007 (± 0.0003) | **1.0 (± 0.047)** | 0.006 (± 0.0003) |
| Oracle (infeasible) | **1.113 (± 0.0396)** | 0.01 (± 0.0003) | **1.355 (± 0.0496)** | 0.01 (± 0.0003) | **1.478 (± 0.0566)** | 0.01 (± 0.0004) |
| Naive-Trim (invalid) | 1.848 (± 0.0507) | 0.018 (± 0.0005) | 3.161 (± 0.0652) | 0.03 (± 0.0006) | 4.447 (± 0.0722) | 0.042 (± 0.0007) |
| Small-Clean | 0.0 (± 0.0) | 0.0 (± 0.0) | 0.0 (± 0.0) | 0.0 (± 0.0) | 0.0 (± 0.0) | 0.0 (± 0.0) |
| Label-Trim | **1.116 (± 0.04)** | 0.01 (± 0.0003) | **1.355 (± 0.0496)** | 0.01 (± 0.0003) | **1.491 (± 0.0582)** | 0.01 (± 0.0004) |
| Standard Power | 0.124 (± 0.0049) | 0.009 (± 0.0003) | 0.103 (± 0.0045) | 0.007 (± 0.0003) | 0.091 (± 0.0043) | 0.006 (± 0.0003) |

(a) $\alpha = 0.01$

| | Contamination rate | | | | | |
| | 1% | | 3% | | 5% | |
| Method | Power | Type-I Error | Power | Type-I Error | Power | Type-I Error |
| --- | --- | --- | --- | --- | --- | --- |
| Standard | **1.0 (± 0.027)** | 0.019 (± 0.0005) | **1.0 (± 0.027)** | 0.014 (± 0.0004) | **1.0 (± 0.0329)** | 0.012 (± 0.0004) |
| Oracle (infeasible) | **1.057 (± 0.0281)** | 0.021 (± 0.0005) | **1.24 (± 0.0302)** | 0.02 (± 0.0005) | **1.507 (± 0.0364)** | 0.02 (± 0.0005) |
| Naive-Trim (invalid) | 1.305 (± 0.0289) | 0.028 (± 0.0006) | 1.957 (± 0.0336) | 0.038 (± 0.0006) | 2.747 (± 0.0395) | 0.05 (± 0.0008) |
| Small-Clean | 0.932 (± 0.0653) | 0.02 (± 0.0018) | 0.533 (± 0.0766) | 0.01 (± 0.0017) | 0.38 (± 0.0889) | 0.007 (± 0.0018) |
| Label-Trim | **1.046 (± 0.0276)** | 0.021 (± 0.0005) | **1.178 (± 0.0299)** | 0.018 (± 0.0005) | **1.344 (± 0.0349)** | 0.016 (± 0.0005) |
| Standard Power | 0.232 (± 0.0063) | 0.019 (± 0.0005) | 0.197 (± 0.0053) | 0.014 (± 0.0004) | 0.163 (± 0.0053) | 0.012 (± 0.0004) |

(b) $\alpha = 0.02$

| | Contamination rate | | | | | |
| | 1% | | 3% | | 5% | |
| Method | Power | Type-I Error | Power | Type-I Error | Power | Type-I Error |
| --- | --- | --- | --- | --- | --- | --- |
| Standard | **1.0 (± 0.0218)** | 0.028 (± 0.0006) | **1.0 (± 0.0244)** | 0.023 (± 0.0005) | **1.0 (± 0.0251)** | 0.018 (± 0.0005) |
| Oracle (infeasible) | **1.088 (± 0.0232)** | 0.031 (± 0.0006) | **1.231 (± 0.0255)** | 0.029 (± 0.0006) | **1.376 (± 0.0272)** | 0.03 (± 0.0006) |
| Naive-Trim (invalid) | 1.224 (± 0.0223) | 0.037 (± 0.0007) | 1.658 (± 0.0276) | 0.047 (± 0.0007) | 2.043 (± 0.0274) | 0.058 (± 0.0009) |
| Small-Clean | 0.758 (± 0.0461) | 0.021 (± 0.0017) | 0.897 (± 0.0659) | 0.023 (± 0.0024) | 0.97 (± 0.0695) | 0.021 (± 0.0022) |
| Label-Trim | **1.051 (± 0.0231)** | 0.03 (± 0.0006) | **1.11 (± 0.0244)** | 0.026 (± 0.0006) | **1.173 (± 0.0261)** | 0.024 (± 0.0005) |
| Standard Power | 0.304 (± 0.0066) | 0.028 (± 0.0006) | 0.261 (± 0.0064) | 0.023 (± 0.0005) | 0.237 (± 0.0059) | 0.018 (± 0.0005) |

(c) $\alpha = 0.03$

Table 23: Comparison of conformal outlier detection methods on TinyImageNet dataset (outliers) and CIFAR-10 dataset (inliers) for varying contamination rate $r$ and target type-I error level $\alpha$. All methods utilize the SCALE (Xu et al., 2024) method with a pretrained ResNet-18. The empirical power is presented relative to the `Standard` method (higher is better). Results are averaged across 100 random splits of the data, with standard errors presented in parentheses.

| | Contamination rate | | | | | |
| | 1% | | 3% | | 5% | |
| Method | Power | Type-I Error | Power | Type-I Error | Power | Type-I Error |
|---|---|---|---|---|---|---|
| Standard | **1.0 (± 0.0383)** | 0.009 (± 0.0003) | **1.0 (± 0.0394)** | 0.006 (± 0.0003) | **1.0 (± 0.0383)** | 0.005 (± 0.0002) |
| Oracle (infeasible) | **1.138 (± 0.0399)** | 0.01 (± 0.0003) | **1.436 (± 0.0534)** | 0.01 (± 0.0003) | **1.625 (± 0.0475)** | 0.01 (± 0.0004) |
| Naive-Trim (invalid) | 1.696 (± 0.0424) | 0.018 (± 0.0004) | 2.88 (± 0.0621) | 0.029 (± 0.0006) | 4.248 (± 0.0707) | 0.041 (± 0.0007) |
| Small-Clean | 0.0 (± 0.0) | 0.0 (± 0.0) | 0.0 (± 0.0) | 0.0 (± 0.0) | 0.0 (± 0.0) | 0.0 (± 0.0) |
| Label-Trim | **1.138 (± 0.0399)** | 0.01 (± 0.0003) | **1.433 (± 0.0535)** | 0.01 (± 0.0003) | **1.591 (± 0.0467)** | 0.009 (± 0.0004) |
| Standard Power | 0.159 (± 0.0061) | 0.009 (± 0.0003) | 0.122 (± 0.0048) | 0.006 (± 0.0003) | 0.102 (± 0.0039) | 0.005 (± 0.0002) |

(a) $\alpha = 0.01$

| | Contamination rate | | | | | |
| | 1% | | 3% | | 5% | |
| Method | Power | Type-I Error | Power | Type-I Error | Power | Type-I Error |
|---|---|---|---|---|---|---|
| Standard | **1.0 (± 0.0248)** | 0.018 (± 0.0004) | **1.0 (± 0.0317)** | 0.014 (± 0.0004) | **1.0 (± 0.0271)** | 0.011 (± 0.0004) |
| Oracle (infeasible) | **1.063 (± 0.0255)** | 0.021 (± 0.0005) | **1.254 (± 0.0309)** | 0.02 (± 0.0005) | **1.49 (± 0.0318)** | 0.02 (± 0.0005) |
| Naive-Trim (invalid) | 1.223 (± 0.0263) | 0.028 (± 0.0006) | 1.816 (± 0.0321) | 0.037 (± 0.0006) | 2.536 (± 0.0365) | 0.049 (± 0.0008) |
| Small-Clean | 0.803 (± 0.0609) | 0.017 (± 0.0018) | 0.625 (± 0.0816) | 0.013 (± 0.0022) | 0.263 (± 0.064) | 0.003 (± 0.001) |
| Label-Trim | **1.053 (± 0.0255)** | 0.02 (± 0.0005) | **1.183 (± 0.0325)** | 0.018 (± 0.0005) | **1.344 (± 0.0334)** | 0.016 (± 0.0004) |
| Standard Power | 0.273 (± 0.0068) | 0.018 (± 0.0004) | 0.227 (± 0.0072) | 0.014 (± 0.0004) | 0.187 (± 0.0051) | 0.011 (± 0.0004) |

(b) $\alpha = 0.02$

| | Contamination rate | | | | | |
| | 1% | | 3% | | 5% | |
| Method | Power | Type-I Error | Power | Type-I Error | Power | Type-I Error |
|---|---|---|---|---|---|---|
| Standard | **1.0 (± 0.0215)** | 0.028 (± 0.0006) | **1.0 (± 0.024)** | 0.022 (± 0.0005) | 1.0 (± 0.0221) | 0.017 (± 0.0005) |
| Oracle (infeasible) | **1.07 (± 0.0222)** | 0.031 (± 0.0006) | **1.188 (± 0.0264)** | 0.029 (± 0.0006) | **1.328 (± 0.0256)** | 0.03 (± 0.0006) |
| Naive-Trim (invalid) | 1.202 (± 0.0203) | 0.037 (± 0.0006) | 1.533 (± 0.0266) | 0.046 (± 0.0007) | 1.916 (± 0.0288) | 0.057 (± 0.0009) |
| Small-Clean | 0.678 (± 0.0476) | 0.017 (± 0.0018) | 0.932 (± 0.0576) | 0.025 (± 0.0025) | **1.005 (± 0.0573)** | 0.024 (± 0.0022) |
| Label-Trim | **1.046 (± 0.0222)** | 0.03 (± 0.0006) | **1.09 (± 0.0256)** | 0.026 (± 0.0005) | **1.14 (± 0.0234)** | 0.023 (± 0.0005) |
| Standard Power | 0.336 (± 0.0072) | 0.028 (± 0.0006) | 0.295 (± 0.0071) | 0.022 (± 0.0005) | 0.266 (± 0.0059) | 0.017 (± 0.0005) |

(c) $\alpha = 0.03$

