# OpenReview forum: "Robust Conformal Outlier Detection under Contaminated Reference Data"
_ICML.cc/2025/Conference — ICML 2025 poster_

### Official Review · Reviewer_6UJm · 2025-02-15

**Overall Recommendation:** 4

**Summary:**

The authors analyze the impact of contamination on the validity of conformal methods. They show that under realistic, non-adversarial
settings, calibration on contaminated data yields conservative type-I error control. This conservativeness, however, typically results
in a loss of power. To alleviate this limitation, they propose a novel, active data-cleaning framework that leverages a limited labeling budget and an outlier detection model to selectively annotate data points in the contaminated reference set that are suspected as outliers.

**Claims And Evidence:**

The claims made by the authors are supported by clear evidence, both theoretical and experimental in nature.

**Essential References Not Discussed:**

None; see also Question **Q1**.

**Experimental Designs Or Analyses:**

The experiments appear to be sound and valid, and support the theoretical claims.

**Methods And Evaluation Criteria:**

The proposed methods seem correct to show the validity of the proposed method.

**Other Comments Or Suggestions:**

See the Questions section.

**Other Strengths And Weaknesses:**

The approach seem novel, and intuitively appealing. The results proved are very interesting, and the impact of the assumptions (and more in general of the limitations) is discussed at length.

**Questions For Authors:**

**Q1** How does the concept of contaminated reference set relate to the ambiguous ground truth case? Refer e.g. to

https://openreview.net/forum?id=CAd6V2qXxc

https://openreview.net/forum?id=L7sQ8CW2FY

https://proceedings.neurips.cc/paper_files/paper/2024/hash/d42a8bf2f40555d4a5120300f98c88f6-Abstract-Conference.html)

I'd like the authors to comment on this (possible) relationship.

**Relation To Broader Scientific Literature:**

The related literature is discussed in sufficient depth. A suggestion for further related works is given in the Questions section.

**Theoretical Claims:**

There are two novel theoretical claims in the paper, whose proofs seem both largely correct.

---

> ### Author Rebuttal · Authors · 2025-04-01
>
> Thank you for the careful review and encouraging feedback. We answer your question below.
>
> > **R1:** How does the concept of contaminated reference set relate to the ambiguous ground truth case?
>
> To the best of our understanding, the papers you referenced study ambiguity in the labeling process within a multi-class classification setting, where some examples may plausibly belong to multiple classes and this ambiguity is explicitly reflected in the labels. In contrast, our work focuses on outlier detection rather than classification, and our setting does not involve any explicit label ambiguity: all calibration points are labeled as inliers, but some may in fact be outliers. The contamination is entirely latent: we have no indication of which points are mislabeled, nor any signal of uncertainty in the labels.
>
> That said, our results do offer some connection to the point of view of the papers you mentioned. As shown in Lemma 2.2 and supported by our experiments, when there is uncertainty about a point’s status, treating it as an inlier is a conservative strategy that preserves type-I error control. This highlights a potentially interesting connection between label ambiguity and reference set contamination, though the two settings at this point still appear quite fundamentally different in how they represent and handle possible labeling errors.
>
> In any case, we agree this connection is worth exploring further in the future and will cite the papers you mentioned while incorporating this discussion into Section 5 of the revised manuscript.

---

### Official Review · Reviewer_HWrV · 2025-03-11

**Overall Recommendation:** 3

**Summary:**

This manuscript, titled "Robust Conformal Outlier Detection under Contaminated Reference Data," focuses on the problem of conformal outlier detection in the presence of contaminated reference data. It discovers that in non-adversarial scenarios, data contamination makes conformal prediction methods conservative and reduces their detection power. To address this, the manuscript proposes the Label-Trim method, which utilizes a limited labeling budget and an outlier detection model to remove outliers from the contaminated data. Theoretical analysis demonstrates that this method can approximately and effectively control the error rate under certain conditions. Experiments comparing multiple methods on several datasets show that the Label-Trim method can significantly enhance the detection power while controlling the type-I error rate, and it performs particularly well in scenarios with low error rates and low contamination rates.

**Claims And Evidence:**

The claims in the manuscript are supported by clear and convincing evidence. Theoretically, the conservativeness of the conformal outlier detection method and the effectiveness of the Label-Trim method are proven through derivations. Experimentally, the comparison results on multiple real-world datasets support the claims of the article.

**Essential References Not Discussed:**

After a thorough review, there don't seem to be any essential related works that are not cited or discussed in the paper. The paper comprehensively references prior research on conformal inference under distribution shifts, robustness to data contamination, and outlier detection.

**Experimental Designs Or Analyses:**

I've examined the experimental designs and analyses in the paper, and they are generally sound and valid. They use a diverse set of benchmark datasets, including three tabular datasets (shuttle, credit card, KDDCup99) and six visual datasets. This variety helps capture different data characteristics and real-world scenarios, enhancing the generalizability of the results.

**Methods And Evaluation Criteria:**

In this manuscript, the proposed Label-Trim method and evaluation criteria are well-suited to the problem of outlier detection with contaminated reference data. The Label-Trim method effectively deals with contaminated data by using a limited labeling budget and an outlier detection model. The use of multiple benchmark datasets and comparison with multiple baseline methods comprehensively assess the method's performance.

**Other Comments Or Suggestions:**

N/A

**Other Strengths And Weaknesses:**

S1: The paper's focus on outlier - robust conformal outlier detection with contaminated reference data fills a significant gap in the existing literature. While many studies assume clean reference data in conformal prediction, this work directly addresses the practical issue of contamination, offering new insights into the behavior of conformal methods under such conditions.
S2: The Label - Trim method is a creative solution. By combining a pre - trained outlier detection model with a limited labeling budget to selectively clean the contaminated reference set, it presents a unique approach to enhancing the power of conformal outlier detection while maintaining type - I error control. This method offers a practical alternative to existing data - cleaning strategies in the context of conformal prediction.
S3: The theoretical analysis of the conservativeness of conformal methods in the presence of contaminated data and the validation of the Label - Trim method contribute to the theoretical understanding of conformal prediction. The results can serve as a foundation for further research in robust conformal inference and outlier detection.
S4: The paper is well - structured, with a clear introduction that motivates the research problem, followed by detailed sections on setup, methods, experiments, and discussion. Each section is logically connected, making it easy for readers to follow the flow of the research.

W1: The effectiveness of the Label - Trim method depends on the accuracy of the pre-trained outlier detection model. If the outlier detection model performs poorly, the performance of the Label - Trim method may be severely affected. The paper does not thoroughly explore how to select or improve the outlier detection model for better performance of the overall system.
W2: Although the paper uses visualizations to present experimental results, there is a lack of visual aids to help readers understand the working mechanism of the Label - Trim method.
W3: While the synthetic data experiments are useful, the authors could explore a wider range of outlier injection strategies.
W4: In real - world datasets, the preprocessing steps might be too simplistic.
W5: There could be more diverse baselines considered.
W6: While type - I error rate and power are important metrics, additional metrics could be considered.

**Questions For Authors:**

N/A

**Relation To Broader Scientific Literature:**

The paper's key contributions are closely related to the broader scientific literature in multiple ways. The Label-Trim method in this paper builds on the existing understanding of conformal prediction. By addressing the over-conservativeness problem in contaminated data scenarios, it fills a gap in the literature. It provides a new approach to enhance the power of conformal methods while maintaining type-I error control, which is an important addition to the body of knowledge on outlier detection and conformal prediction.

**Theoretical Claims:**

I have checked the correctness of the proofs for the theoretical claims in the paper. The proofs of Lemma 2.2 and Theorem 3.1 are particularly crucial. For Lemma 2.2, which analyzes the conservativeness of standard conformal p-values under contaminated data, the proof process is clear and the logic is rigorous. The proof of Theorem 3.1, which validates the Label-Trim method, is also well-structured. It constructs an imaginary "mirror" version of the method to analyze the relationship between different quantiles. It provide solid theoretical support for the claims in the paper.

---

> ### Author Rebuttal · Authors · 2025-04-01
>
> Thank you for your thoughtful and encouraging feedback. To fully address your comments, we have conducted additional experiments, available at https://tinyurl.com/rcod-exps. In our responses below, we refer to these as Supp-Figure X and Supp-Table Y.
>
> > **R1:** The effectiveness of the Label - Trim method depends on the accuracy of the pre-trained outlier detection model. The paper does not thoroughly explore how to select or improve the outlier detection model for better performance of the overall system.
>
> You're absolutely right that the performance of Label-Trim depends on the quality of the outlier detection model used to rank points for annotation. To explore this further, and in response to your concern (as well as R4 from Reviewer Zuz4), we conducted new experiments using outlier detection models with lower performance than Isolation Forest, specifically Local Outlier Factor and one-class SVM on the tabular datasets. As shown in Supp-Figures 2–3, when the underlying model struggles to distinguish inliers from outliers, Label-Trim does not show meaningful power gains over the standard conformal method. This is expected, as the model’s scores no longer reliably identify outliers.
>
> That said, as with any outlier detection task, we recommend using the best model available or fine-tuning a given model using a small set of labeled outliers, something that naturally improves performance. Importantly, Label-Trim is model-agnostic: it imposes no constraints on model complexity or architecture. This ensures that our approach remains compatible with future developments in outlier detection.
>
> > **R2:** Although the paper uses visualizations to present experimental results, there is a lack of visual aids to help readers understand the working mechanism of the Label - Trim method.
>
> Thank you for this excellent suggestion. We agree that a visual explanation would help communicate the logic and steps of the method more clearly. In the revised version, we will include a new schematic illustration that walks through the Label-Trim pipeline.
>
> > **R3:** While the synthetic data experiments are useful, the authors could explore a wider range of outlier injection strategies.
>
> We appreciate this point and want to clarify that all our experiments are conducted on real-world datasets (both inliers and outliers originate from real applications). That said, we agree that modeling more nuanced corruption is important. In our response to Reviewer 2tsw (R4), we include additional experiments with more strategic outlier injection strategies, which make the detection task more challenging.
>
> > **R4:** In real - world datasets, the preprocessing steps might be too simplistic.
>
> We’re not entirely sure what specific aspect of preprocessing you are referring to. In our work, we follow the preprocessing protocols established in prior studies (Bates et al., 2023; Zhang et al., 2024; Yang et al., 2022). Our main departure is the introduction of contamination into the training and calibration sets to reflect realistic scenarios more closely. If there are particular concerns about the preprocessing that we overlooked, we would be happy to address them.
>
> > **R5:** There could be more diverse baselines considered.
>
> We agree that comparisons with a broader set of baselines are useful. In addition to the four methods evaluated in the main manuscript, we have now added results for two additional outlier detection models on tabular data (Local Outlier Factor and one-class SVM), as well as two new models for visual data (ReAct with a VGG19 backbone and SCALE with a ResNet backbone). We’ve also included new experiments under higher contamination rates (up to 15%) and across a range of type-I error levels. As shown in the updated results, type-I error is consistently controlled, and power improves when the detection model separates inliers from outliers with reasonable accuracy.
>
> > **R6:** While type - I error rate and power are important metrics, additional metrics could be considered.
>
> We focused on type-I error and power because they are the standard evaluation metrics in the conformal prediction and outlier detection literature. These metrics directly capture the validity (false positive control) and utility (detection rate) of the method. To provide additional insight, we also report the number of trimmed outliers in the contaminated reference set as a measure of cleaning effectiveness.

---

### Official Review · Reviewer_Zuz4 · 2025-03-12

**Overall Recommendation:** 3

**Summary:**

This paper studies conformal outlier detection with contaminated reference sets. It theoretically shows that non-adversarial contamination induces conservative type-I error control, explaining empirical performance gaps. To address power loss, the authors propose Label-Trim: an active data-cleaning framework leveraging limited labeling budgets to annotate and remove suspected outliers from high-scoring regions. Theoretical analysis proves Label-Trim maintains approximate error control under practical conditions. Experiments on tabular and vision datasets validate that standard conformal methods become conservative under contamination, while Label-Trim recovers detection power without inflating errors, achieving near-oracle performance when contamination rates are low.

**Claims And Evidence:**

The paper’s key claims—conservative error control under contamination and Label-Trim’s power recovery—are supported by theoretical proofs (Lemma 2.2, Theorem 3.1) and experiments on tabular/vision datasets. Empirical validation includes score distribution analysis (Fig.1), type-I error trajectories (Fig.2,4), and detection rate comparisons (Fig.2-3, Table1). Near-oracle performance at low contamination is numerically confirmed.  Experiments exclude high contamination rates (>5%), limiting claims about robustness to realistic but challenging data shifts.

**Essential References Not Discussed:**

None

**Experimental Designs Or Analyses:**

Experiments are sound for core claims: controlled contamination (1%-5%) tests type-I error/power trade-offs. Tabular (Isolation Forest) and vision (ResNet) benchmarks are standard. However:

1. Contamination is simulated via random outlier injection, ignoring realistic contamination scenarios.
2. High contamination (>5%) and real-world drift (e.g., temporal shifts) are untested.
3. Only one detector per data type is used; architecture variations are unexplored.

**Methods And Evaluation Criteria:**

The proposed Label-Trim method aligns with practical constraints: it uses a pre-trained outlier detector to prioritize high-scoring calibration samples for limited manual labeling, then trims confirmed outliers. This leverages model confidence to focus labeling efforts, avoiding random or exhaustive cleaning. Evaluation employs standard conformal metrics (type-I error rate, detection power) on 3 tabular  and 6 vision datasets, with controlled contamination rates (1%-5%). Baseline comparisons include Oracle (clean reference), Standard (no cleaning), Naive-Trim (remove top scores without labels), and Small-Clean (random labeling).

While the datasets are established in outlier detection, the contamination simulation (random outlier injection) oversimplifies real-world scenarios where outliers may strategically mimic inliers. The 5% contamination cap excludes high-pollution cases common in practice (e.g., 10%-20%). The focus on low-error regimes (α=0.01-0.03) matches safety-critical applications but ignores moderate α settings. Isolation Forest (tabular) and ReAct (vision) are reasonable model choices, but using only one detector per data type (without testing alternatives) limits confidence in generalizability. The dependence on detector quality is not systematically tested, weakening claims about robustness across detector architectures.

**Other Comments Or Suggestions:**

1. **Dynamic Budget Allocation:** Explore adaptive labeling budgets (e.g., increasing *m* when contamination is suspected) instead of fixed *m=50*.
2. **Detector-Agnostic Analysis:** Systematically test Label-Trim with alternative architectures (e.g., autoencoders, One-Class SVM) to assess generalizability.

**Other Strengths And Weaknesses:**

**Strengths:**

1. **Originality:** This paper is the first to bridge conformal prediction with outlier detection under reference set contamination, bridging anomaly detection and robust statistics.

2. **Practicality:** Label-Trim’s simplicity suits real-world deployment.

3. **Clarity:** Figures (e.g., score distributions) intuitively explain conservative behavior.

**Weaknesses:**

1. **Moderate innovation:** The method’s simple pipeline (score-sort-label) lacks deeper algorithmic novelty compared to recent advances.

2. **Assumption-heavy theory:** Relies on independent and identically distributed inliers, limiting real-world applicability.

3. **Narrow scope:** Focus on low contamination (≤5%) excludes high-noise scenarios common in practice.

**Questions For Authors:**

1. How does Label-Trim perform on naturally contaminated datasets? Synthetic noise may not reflect real-world anomalies.
2. How does Label-Trim perform on high contamination rate (>10%) data?
3. Would the method fail catastrophically when the number of outliers exceeds the labeling budget *m*?

**Relation To Broader Scientific Literature:**

This paper is the first to bridge conformal prediction with outlier detection under reference set contamination, introducing a novel intersection of these fields. Prior conformal works focused on covariate/label shifts (Tibshirani et al., 2019) or label noise (Sesia et al., 2024), but none addressed **contaminated calibration sets** in outlier detection. Unlike semi-supervised anomaly detection (Jiang et al., 2022), which assumes partial labels, Label-Trim uses limited labels to clean reference data while preserving conformal guarantees—a unique hybrid approach. Theoretically, it extends conservative error bounds (Sesia et al., 2024) to **unknown outlier distributions**, avoiding explicit noise modeling. Compared to worst-case robustness analyses (Barber et al., 2023), this work identifies practical conservatism under non-adversarial contamination, aligning with empirical patterns. Experiments validate the framework on both tabular and vision data, broadening conformal methods beyond traditional single-domain applications.

- Tibshirani, R. J., Foygel Barber, R., Cande`s, E., and Ramdas, A. Conformal prediction under covariate shift. Advances in neural information processing systems, 32, 2019.
- Sesia, M., Wang, Y. R., and Tong, X. Adaptive conformal classification with noisy labels. J. R. Stat. Soc. Series B, pp. qkae114, 2024.
- Barber, R. F., Cand`es, E. J., Ramdas, A., and Tibshirani, R. J. Conformal prediction beyond exchangeability. Ann. Stat., 51(2):816–845, 2023.
- Jiang, X., Liu, J., Wang, J., Nie, Q., Wu, K., Liu, Y., Wang, C., and Zheng, F. Softpatch: Unsupervised anomaly detection with noisy data. Advances in Neural Information Processing Systems, 35:15433–15445, 2022.

**Theoretical Claims:**

The proofs for Lemma 2.2 and Theorem 3.1 are mathematically correct under their assumptions (e.g., i.i.d. inliers, fixed contamination).

---

> ### Author Rebuttal · Authors · 2025-04-01
>
> Thank you for your thoughtful and constructive feedback. We appreciate the opportunity to provide some clarifications. To address your questions and strengthen the empirical foundation of our claims, we have also conducted a range of new experiments available at https://tinyurl.com/rcod-exps, which we reference below as Supp-Figure X and Supp-Table Y.
>
> Our goal in this paper is to address a critical gap in conformal outlier detection: robustness to contamination in the reference set. To the best of our knowledge, no prior work offers a fully satisfactory solution to this problem. While our proposed method is deliberately simple, its theoretical guarantees are non-trivial, and our new experiments demonstrate its practical robustness.
>
> > **R1:** The method’s simple pipeline lacks deeper algorithmic novelty
>
> We understand this concern, though we believe the intuitive nature of our approach is a practical strength. Moreover, while the method is intentionally simple, establishing its validity is technically non-trivial. Our analysis required developing a novel proof technique, which we believe offers theoretical and methodological value beyond the specific application studied in this paper.
>
> > **R2:** Assumes i.i.d. inliers
>
> You're right that conformal prediction traditionally assumes i.i.d. inliers and a clean reference set. Our work relaxes the second of these assumptions: we allow the reference set to be contaminated with non-i.i.d. outliers. The assumption we retain is that the inliers themselves are i.i.d. We agree that relaxing the i.i.d. assumption on the inliers is an exciting direction for future work. However, doing so would likely require additional assumptions on the dependency structure, going outside the scope of this paper. We will clarify this point in the revision.
>
> > **R3:** Focus on low contamination ≤5%
>
> Thank you for pointing this out. As described in our response to Reviewer 2tsw, we’ve extended our experiments to include higher contamination levels, and continue to observe consistent trends.
>
> > **R4:** Systematically test Label-Trim with alternative models
>
> We appreciate the suggestion. In addition to Isolation Forest (tabular) and ReAct with a ResNet backbone (visual) used in the main manuscript, we’ve now added results for:
> Tabular data: Local Outlier Factor (LOF) and One-Class SVM (OCSVM)
> Visual data: ReAct with a VGG19 backbone and SCALE with a ResNet backbone
> As shown in Supp-Figures 2–3, Label-Trim consistently controls the type-I error across all models. As expected, when the outlier detection model is less effective (e.g., LOF, OCSVM), the candidate set becomes noisier and power decreases. Still, our method improves over the baseline. For the visual data, Supp-Tables 1–2 show results that closely mirror the trends in the main manuscript. We’ll include these in the revision.
>
> > **R5:** Random outlier injection oversimplifies real-world scenarios where outliers may strategically mimic inliers
>
> We completely agree and have addressed this in our response to Reviewer 2tsw, where we design and test more challenging outlier scenarios.
>
> > **R6:** Real-world drift are untested
>
> We share your interest in this important issue. In our response to Reviewer 2tsw, we present new experiments simulating drift in the outlier distribution over time, and find that Label-Trim remains robust.
>
> > **R7:** Explore adaptive labeling budgets instead of fixed m=50.
>
> We refer you to our response to Reviewer 2tsw, where we discuss the role of the annotation budget m, and show that Label-Trim performs well across a range of values. We also highlight that its performance degrades gracefully as the budget decreases.
>
> > **R8:** How does Label-Trim perform on naturally contaminated datasets?
>
> Please see our response to Reviewer 2tsw, where we explain why controlled contamination is necessary for rigorous evaluation and provide additional experiments modeling more realistic contamination.
>
> > **R9:** Would the method fail catastrophically when the number of outliers exceeds the labeling budget m?
>
> No. Both our theoretical results and experiments show that type-I error control is preserved regardless of how small m is relative to the number of outliers. For instance, in Figure 3 of the main manuscript and Supp-Figure 1 (right panel), the number of outliers significantly exceeds the labeling budget, yet Label-Trim still controls the error and achieves meaningful power gains.
>
> > **R10:** The focus on low-error regimes (α=0.01-0.03) matches safety-critical applications but ignores moderate α
>
> We’ve added experiments evaluating performance for higher type-I error thresholds. As shown in Supp-Figure 9, Label-Trim continues to perform well across all tested α levels. We'll incorporate these results into the revised version.

---

### Official Review · Reviewer_2tsw · 2025-03-18

**Overall Recommendation:** 3

**Summary:**

This paper analyzes the impact of such contamination on the validity of conformal methods. The paper proves that under realistic, non-adversarial settings, calibration on contaminated data yields conservative type-I error control.

**Claims And Evidence:**

This paper focuses on detecting outliers with conformal prediction in the context of contaminated reference data, which is both an interesting and important problem.

**Essential References Not Discussed:**

No.

**Experimental Designs Or Analyses:**

Extensive experiments on real data validate that standard conformal outlier detection methods are conservative under contamination and show that the proposed method improves power without sacrificing validity in practice.

**Methods And Evaluation Criteria:**

Yes.

**Other Comments Or Suggestions:**

No.

**Other Strengths And Weaknesses:**

Weaknesses:
1. In most experiments, m is fixed to 50. Generally, how should $m$ be set to achieve good model performance?
2. This paper primarily considers injected outliers within the contaminated calibration set. How does the proposed method perform when applied to real data that is inherently contaminated?

**Questions For Authors:**

See the weakness.

**Relation To Broader Scientific Literature:**

Yes.

**Theoretical Claims:**

The authors provide a solid and new theoretical analysis of the proposed method in terms of the type-I error rate.

---

> ### Author Rebuttal · Authors · 2025-04-01
>
> Thank you for your thoughtful review. We’ve conducted additional experiments to answer your questions and further support and clarify our contributions. These results are available at https://tinyurl.com/rcod-exps, and we refer to them as *Supp-Figure X* and *Supp-Table Y* in our responses below.
>
> > **R1:** In most experiments, m is fixed to 50. Generally, how should it be set to achieve good model performance?
>
> We view the annotation budget m as primarily determined by resource constraints (how much it costs to label the data points) rather than as a tuning parameter that can be optimized by a general mathematical argument. This is because the power of our method depends not just on m, but also on the performance of the outlier detection model and the underlying contamination level, both of which are hard to characterize theoretically without much stronger assumptions than we make in this work.
>
> That said, Figure 3 in the main manuscript provides interesting practical insight. In that experiment, the calibration set contains approximately 80 outliers. As shown in the figure, even a small annotation budget (e.g., m ≈ 20) is enough for Label-Trim to achieve power nearly matching that of the oracle conformal method that uses clean reference data. This suggests that even a modest budget (on the order of a quarter of the expected number of outliers) can already yield strong performance. Even more encouragingly, the power of Label-Trim appears to vary quite smoothly with the labeling budget in these experiments, indicating graceful performance degradation under tighter annotation constraints.
>
> Thank you for giving us the opportunity to further highlight this important aspect of Figure 3. We will clarify this message and the associated practical guidance in the revised version.
>
>
> > **R2:** How does the proposed method perform when applied to real data that is inherently contaminated?
>
> To answer this question, which also relates to Reviewer Zuz4’s comments, we conducted additional experiments simulating various realistic contamination scenarios. We summarize these results at the end of this response.
>
> We also wish to clarify that all our existing experiments are already based on real-world datasets, meaning both the inliers and outliers originate from actual applications. Since our method relies on selective annotation under a limited budget, we simulate contamination in the reference set by injecting outliers. This setup reflects a realistic scenario where labeling resources are constrained and full manual annotation is impractical due to the need for domain expertise.
>
> Moreover, a controlled contamination process is essential for evaluating our method because computing the performance metrics requires knowing the ground-truth labels. Specifically, to estimate type-I error, we need to know which test points are inliers, and to measure detection power, we need to know which test points are outliers.
> Summary of additional experiments:
>
> - **Varying contamination rate.** We extend Figure 2 by increasing the contamination rate up to 15%. As shown in Supp-Figure 1, Label-Trim maintains type-I error control while continuing to outperform standard conformal and the “small clean” baseline in terms of power.
> - **Strategic outlier injection.** Instead of injecting outliers at random, we selected outliers that resemble inliers—those falling below a given score percentile. Supp-Figure 5 (shuttle dataset) shows how these lower-percentile outliers increasingly resemble inliers, while Supp-Figure 4 demonstrates that Label-Trim still controls the error and improves power. Similar trends hold on the credit-card and KDDCup99 datasets (Supp-Figures 6–7).
> - **Test-time distribution drift.** On the shuttle dataset, we simulate drift in the outlier distribution by contaminating the calibration set with high-percentile outliers and gradually shifting to harder, low-percentile outliers at test time. As shown in Supp-Figure 8, Label-Trim remains robust, maintaining error control and power throughout the distribution shift.
>
> We will revise the paper to clarify how our experimental design reflects real-world constraints and to incorporate discussion of these new results demonstrating robustness to a range of contamination scenarios.

---

### Decision · Program_Chairs · 2025-05-01

**Decision:**

Accept (poster)

**Comment:**

This paper addresses conformal outlier detection in the presence of contaminated reference data. It is shown that under realistic, non-adversarial settings, calibration on contaminated data yields conservative type-I error, which leads to a loss of power. To alleviate this limitation, a method referred to as Label-Trim is proposed to remove as many outliers as possible from the calibration set without altering the inlier score distribution. The upper bound on the type-I error rate is provided. During the discussion period after the rebuttal, all of reviewers agree that the paper  has valuable contributions.